# Interneuron-specific gamma synchronization indexes cue uncertainty and prediction errors in lateral prefrontal and anterior cingulate cortex

Kianoush Banaie Boroujeni[1]*, Paul Tiesinga[2], Thilo Womelsdorf[1,3]*

[1]Department of Psychology, Vanderbilt University, Nashville, United States; [2]Donders Institute for Brain, Cognition and Behaviour, Radboud University Nijmegen, Nijmegen, Netherlands; [3]Department of Biology, Centre for Vision Research, York University, Toronto, Canada

**Abstract** Inhibitory interneurons are believed to realize critical gating functions in cortical circuits, but it has been difficult to ascertain the content of gated information for well-characterized interneurons in primate cortex. Here, we address this question by characterizing putative interneurons in primate prefrontal and anterior cingulate cortex while monkeys engaged in attention demanding reversal learning. We find that subclasses of narrow spiking neurons have a relative suppressive effect on the local circuit indicating they are inhibitory interneurons. One of these interneuron subclasses showed prominent firing rate modulations and (35–45 Hz) gamma synchronous spiking during periods of uncertainty in both, lateral prefrontal cortex (LPFC) and anterior cingulate cortex (ACC). In LPFC, this interneuron subclass activated when the uncertainty of attention cues was resolved during flexible learning, whereas in ACC it fired and gamma-synchronized when outcomes were uncertain and prediction errors were high during learning. Computational modeling of this interneuron-specific gamma band activity in simple circuit motifs suggests it could reflect a soft winner-take-all gating of information having high degree of uncertainty. Together, these findings elucidate an electrophysiologically characterized interneuron subclass in the primate, that forms gamma synchronous networks in two different areas when resolving uncertainty during adaptive goal-directed behavior.

*For correspondence:
kianoush.banaie.boroujeni@
vanderbilt.edu (KBB);
thilo.womelsdorf@vanderbilt.edu
(TW)

**Competing interests:** The authors declare that no competing interests exist.

## Introduction

Inhibitory interneurons in prefrontal cortex are frequently reported to be altered in neuropsychiatric diseases with debilitating consequences for cognitive functioning. Groups of fast spiking interneurons with basket cell or chandelier morphologies have consistently been found to be abnormal in individuals with schizophrenia and linked to dysfunctional working memory and reduced control of attention (*Dienel and Lewis, 2019*). Altered functioning of a non-fast spiking interneuron class is linked to reduced GABAergic tone in individuals with severe major depression (*Levinson et al., 2010*; *Fee et al., 2017*). These findings suggest that the circuit functions of different subtypes of interneurons in prefrontal cortices are important to regulate specific aspects of cognitive and affective functioning.

But it has remained a challenge to identify how individual interneuron subtypes support specific cognitive or affective functions in the nonhuman primate. For rodent prefrontal and anterior cingulate cortices, cells with distinguishable functions express differentially cholecystokinin (CCK), parvalbumin (PV), or somatostatin (SOM), amongst others (*Roux and Buzsáki, 2015*; *Cardin, 2018*). Prefrontal CCK expressing basket cells have been shown to impose inhibition that is required during

the choice epoch, but not during the delay epoch of a working memory task (*Nguyen et al., 2020*). In contrast, retention of visual information during working memory delays has been shown to require activation specifically of PV+ expressing fast spiking interneurons (*Lagler et al., 2016*; *Kamigaki and Dan, 2017*; *Nguyen et al., 2020*). In the same prefrontal circuits, the PV+ neurons have also been associated with attentional orienting (*Kim et al., 2016*), shifting of attentional sets and response strategies during reward learning (*Cho et al., 2015*; *Canetta et al., 2016*; *Cho et al., 2020*), and with spatial reward choices (*Lagler et al., 2016*), among other functions (*Pinto and Dan, 2015*). Distinct from PV+, the group of somatostatin expressing neurons (SOM+) have been shown to be necessary during the initial encoding phase of a working memory task but not during the delay (*Abbas et al., 2018*), and in anterior cingulate cortex they activate specifically during the approach of reward sites (*Kvitsiani et al., 2013*; *Urban-Ciecko and Barth, 2016*). Taken together, these findings illustrate that rodent prefrontal cortex interneurons expressing PV, SOM, or CCK fulfill separable, unique roles at different processing stages during goal-directed task performance (*Pinto and Dan, 2015*; *Lagler et al., 2016*).

The rich insights into cell-specific circuit functions in rodent prefrontal cortices stand in stark contrast to the limited empirical data from primate prefrontal cortex. While there are recent advances using optogenetic tools for use in primates (*Acker et al., 2016*; *Dimidschstein et al., 2016*; *Gong et al., 2020*), most existing knowledge about cell-specific circuit functions are indirectly inferred from studies that distinguish only one group of putative interneurons that show narrow action potential spike width. Compared to broad spiking neurons the group of narrow spiking, putative interneurons in lateral prefrontal cortex have been found to more likely encode categorical information during working memory delays (*Diester and Nieder, 2008*), show stronger stimulus onset responses during cognitive control tasks (*Johnston et al., 2009*), stronger attentional modulation (*Thiele et al., 2016*), more location-specific encoding of task rules (*Johnston et al., 2009*), stronger reduction of firing selectivity for task irrelevant stimulus features (*Hussar and Pasternak, 2009*), stronger encoding of errors and loss (*Shen et al., 2015*; *Sajad et al., 2019*), more likely encoding of outcome history (*Kawai et al., 2019*), and stronger encoding of feature-specific reward prediction errors (*Oemisch et al., 2019*), amongst other unique firing characteristics (*Constantinidis and Goldman-Rakic, 2002*; *Ardid et al., 2015*; *Rich and Wallis, 2017*; *Voloh and Womelsdorf, 2018*; *Torres-Gomez et al., 2020*).

These summarized findings suggest that there are subtypes of narrow spiking neurons that are particularly important to regulate prefrontal circuit functions. But it is unclear whether these narrow spiking neurons are inhibitory interneurons and to which interneuron subclass they belong. Comparisons of protein expression with action potential spike width have shown for prefrontal cortex that > 95% of all PV+ and ~ 87% of all SOM + interneurons show narrow spike width (*Ghaderi et al., 2018*; *Torres-Gomez et al., 2020*), while narrow spikes are also known to occur in ~20% of VIP interneurons (*Torres-Gomez et al., 2020*) among other GABAergic neurons (*Krimer et al., 2005*; *Zaitsev et al., 2009*), and (at least in primate motor cortex) in a subgroup of pyramidal cells (*Soares et al., 2017*). In addition, electrophysiological characterization has shown at least three different types of firing patterns in narrow spiking neurons of monkeys during attention demanding tasks (*Ardid et al., 2015*; *Dasilva et al., 2019*; *Trainito et al., 2019*). Taken together, these insights raise the possibility that spike width and electrophysiology will allow identifying the interneuron subtypes that are particularly important for prefrontal cortex functions.

Here, we investigated this possibility by recording narrow spiking cells in nonhuman primate prefrontal and cingulate cortex during an attention demanding reversal learning task. We found that in both areas three narrow spiking neuron classes are well distinguished and show a suppressive influence on the local circuit activity compared to broad spiking neurons, supporting labeling them as inhibitory interneurons. Among these interneurons the same sub-type showed significant functional correlations in both ACC and LPFC, firing stronger to reward predictive cues when their predictability is still learned during the reversal (in LPFC), and firing stronger to outcomes when they are most unexpected during reversal (in ACC). Notably, in both, ACC and LPFC, these functions were evident in 35–45 Hz gamma rhythmic synchronization to the local field potential in the same interneuron subclass.

# Results

We used a color-based reversal paradigm that required subjects to learn which of two colors were rewarded as described previously (*Oemisch et al., 2019*). The rewarded color reversed every ~30–40 trials. Two different colors were assigned to stimuli appearing randomly left and right to a central fixation point (*Figure 1A*). During the task the color information was presented independently from the up-/downward- direction of motion of the stimuli. The up-/downward direction instructed the saccade direction that animals had to show to a Go event in order to receive reward. Motion was thus the cue for an overt choice (with saccadic eye movements), while color was the cue for covert selective attention. Color was shown either before (as Feature-1) or after the motion onset (as Feature-2) (*Figure 1B*). Both animals took on average 7/7 (monkey H/K) trials to reach criterion performance, that is, they learned which color was rewarded within seven trials (*Figure 1C*). The asymptotic performance accuracy was 83/86% for monkey's H/K (see Materials and methods).

## Characterizing narrow spiking neurons as inhibitory interneurons

During reversal performance, we recorded the activity of 329 single neurons in LPFC areas 46/9 and anterior area 8 (monkey H/K: 172/157) and 397 single neurons in dorsal ACC area 24 (monkey H/K: 213/184) (*Figure 1D*, *Figure 1—figure supplement 1*). The average action potential waveform shape of recorded neurons distinguished neurons with broad and narrow spikes similar to previous studies in LPFC and ACC (*Gregoriou et al., 2012*; *Ardid et al., 2015*; *Westendorff et al., 2016*; *Dasilva et al., 2019*; *Oemisch et al., 2019*; *Figure 1E*). Prior biophysical modeling has shown that the extracellular action potential waveform shape, including its duration, is directly related to transmembrane currents and the intracellularly measurable action potential shape and duration (*Gold et al., 2006*; *Bean, 2007*; *Gold et al., 2007*; *Buzsáki et al., 2012*). Based on this knowledge we quantified the extracellularly recorded spike duration of the inferred hyperpolarization rates and their inferred time-of-repolarizations (*see* Materials and methods, *Figure 1—figure supplement 2A, B*). These measures split narrow and broad spiking neurons into a bimodal distribution (calibrated Hartigan's dip test for bimodality, p<0.001), which was better fit with two than one gaussian (*Figure 1E*, Bayesian information criterion for two and one gaussian fit: 4.0450, 4.8784, where a lower value indicates a better model). We found in LPFC 21% neurons had narrow spikes (n = 259 broad, n = 70 narrow cells) and in ACC 17% of neurons had narrow action potentials (n = 331 broad, n = 66 narrow cells).

To assess the excitatory or inhibitory identity of the broad and narrow spiking neuron classes (*B-* and *N-type* neurons), we estimated the power of multi-unit activity (MUA) in its vicinity (at different electrodes than the spiking neuron) around the time of spiking for each cell and tested how this spike-triggered MUA-power changed before versus after the cell fired a spike (see Materials and methods). This approach expects for an excitatory neuron to spike concomitant with neurons in the local population reflected in a symmetric rise and fall of MUA before and after its spike. In contrast, inhibitory neurons are expected to spike when MUA rises, but when the spike occurs, the spike should contribute to suppress the local MUA activity, which should be reflected in a faster drop in MUA activity after the spike occurred (*Oemisch et al., 2015*). We found that B-*type* cells showed on average a symmetric pre- to post- spike triggered MUA activity modulation indicative of excitatory participation with local activity (*Figure 1F*). In contrast, spikes of *N-type* cells were followed by a faster drop of MUA activity indicating an inhibitory influence on MUA (*Figure 1F*). The excitatory and inhibitory effects on local MUA activity were consistent across the population and significantly distinguished *B-* and *N-type* neurons (*Figure 1G*; MUA modulation index: [(post MUA$_{spike}$ - pre MUA$_{spike}$)/pre MUA$_{spike}$] for *B-* vs *N-type* cells, Wilcoxon test, p=0.001). This distinction was evident in ACC and in LPFC (*Figure 1H*; for the *N-type* the MUA modulation index was different from zero, Wilcoxon test, in ACC, p<0.001, and in LPFC, p=0.03; for B-type cells the difference was not sign.). These findings suggest narrow spiking cells contain mostly inhibitory interneurons (see Discussion).

## Putative interneurons in prefrontal cortex index choices when choice probability is low

To discern how *B-* and *N- type* neurons encoded the learning of the rewarded color during reversal, we analyzed neuronal response modulation around color onset, which instructed animals to covertly shift attention to the stimulus with the reward predicting color. In addition to this *color cue* (acting

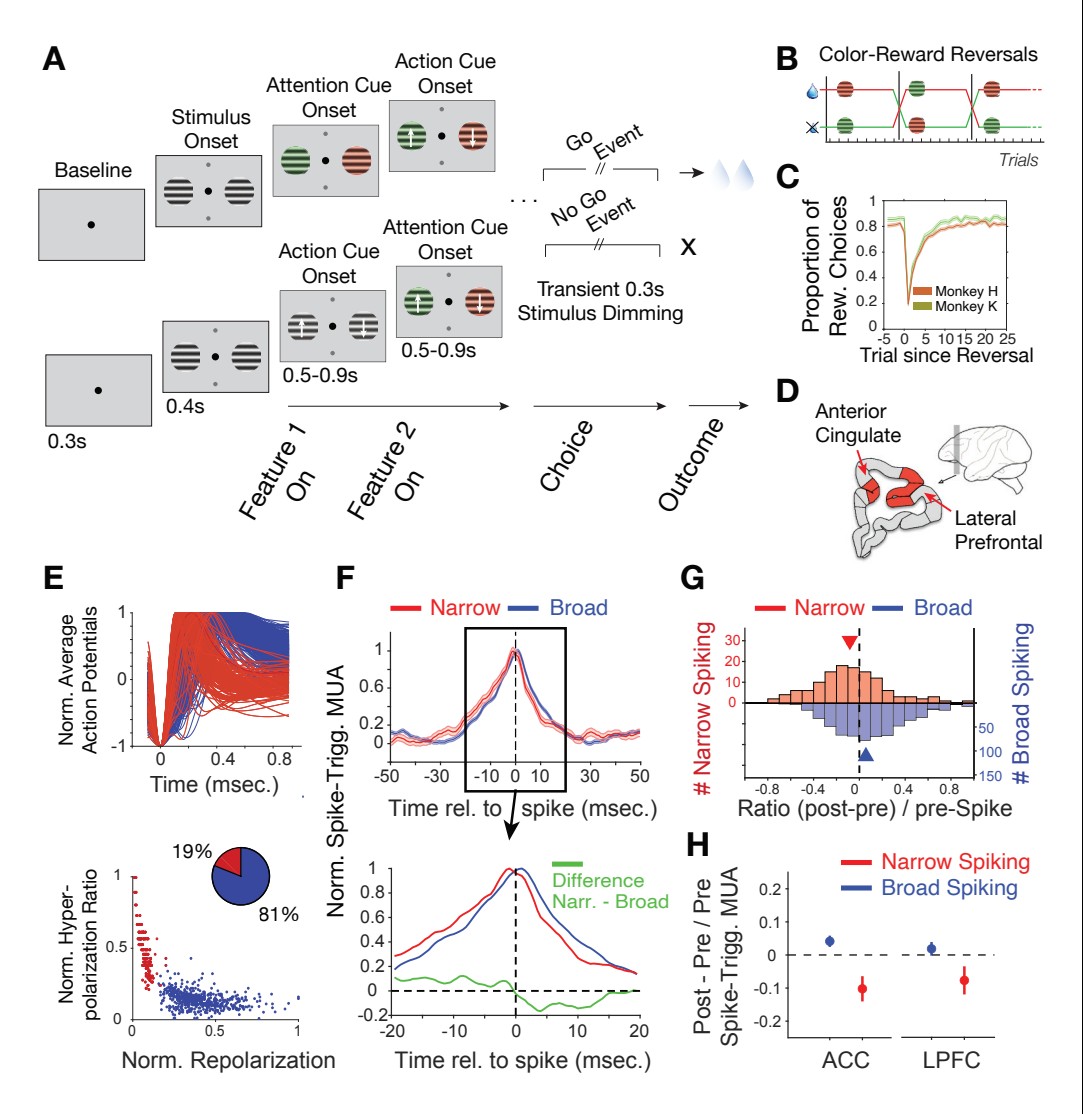

**Figure 1.** Task paradigm and cell classification. (**A**) Trials required animals to covertly attend one of two peripheral stimuli until a dimming (Go-event) instructed to make a saccade in the direction of the motion of the attended stimulus. During the trial, the two stimuli were initially static black/white and then either were colored first or started motion first. Following this feature 1 Onset the other feature (Feature two on) was added 0.5–0.9 s later. (**B**) The task reversed the color (red or green) that was rewarded over at least 30 trials. (**C**) Two monkeys learned through trial-and-error the reward-associated color as evident in increased accuracy choosing the rewarded stimulus (*y-axis*) over trials since reversal (*x-axis*). (**D**) Recorded areas (details in *Figure 1—figure supplement 1*). (**E**) *Top*: Average normalized action potential waveforms of recorded neurons were narrow (*red*) or broad (*blue*). *Bottom*: Inferred hyperpolarization ratio and repolarization duration distinguishes neurons. (**F**) Average spike-triggered multiunit modulation for narrow and broad spiking neurons (Errors are SE's). Spiking neuron and MUA were from different electrodes. The bottom panel zooms into the ±20 ms around the spike time and shows the difference between neuron classes (in green). (**G**) The histogram of post-to-pre spike AUC ratios for narrow (*red*) and broad (*blue*) spiking neurons. (**H**) Average ratio of post- to pre-spike triggered MUA for narrow and broad cell classes in ACC (*left*) and in LPFC (*right*). Values < 0 indicate reduced post- versus pre-spike MUA modulation. Error bars are SE.

The online version of this article includes the following figure supplement(s) for figure 1:

**Figure supplement 1.** Anatomical locations of recording sites.

**Figure supplement 2.** Action potential waveform parameters and spike variability measures used for clustering cells.

as attention cue), we also analyzed activity around the motion onset that served as *action cue*. Its direction of motion indicated the saccade direction the animal had to elicit for receiving reward. This *action cue* could happen either 0.5–0.9 s. before or 0.5–0.9 s. after the *color cue*. Many neurons in LPFC selectively increased their firing to the *color attention cue* with no apparent modulation to the *motion action cue* (n = 71 cells with firing increases to the color but not motion cue) (for examples: *Figure 2A,B*). These neurons increased firing to the color onset when it was the first, or the second feature that was presented, but did not respond to the motion onset when it was shown as first or second feature (for more examples, *Figure 2—figure supplement 1*).

We found that N-type neurons in LPFC change transiently their firing to the attention cue when it occurred either early or late relative to the action cue (significant increase within 25–275 ms post-cue for Feature 1 and within 50–250 ms post-cue for Feature 2, p<0.05 randomization statistics, n = 21 *N-type* cells with increases and seven with decreases to the color cue, *Figure 2C*). This attention cue-specific increase was absent in *B-type* neurons in LPFC (n.s., randomization statistics, n = 44 *B-type* cells with increases and n = 35 with decreases to the color cue, *Figure 2C*). In contrast to LPFC, ACC *N-* and *B-type* neurons did not show an on-response to the color cue (n = 36/6 *B- and N- type* cells with increases, respectively, and n = 31/12 *B- and N- type* cells with decreased firing, respectively, to the color cue, the total cell number included in this analysis for the *B- and N- type* was n = 216/50, respectively) (*Figure 2D*).

The *N-type*-specific response to the attention cue might carry information about the rewarded stimulus color or the rewarded stimulus location. We found that the proportion of neurons whose firing rate significantly distinguished rewarded and nonrewarded colors sharply increased for *N-type* cells after the onset of the color cue in LPFC proportion of color selective responses within 0–0.5 s. after cue, 18%; n = 10 of 54 N-type cells, randomization test p<0.05 within [175 575] ms after cue onset, but not in ACC (cells with significant information: 6%; n = 3 of 50 N-type cells, ns., randomization test within [300 700] ms after cue onset) (*Figure 2—figure supplement 2A,B*). Similar to the selectivity for the rewarded stimulus color *N-type* cells in LPFC (but not in ACC) showed significant encoding of the right versus left location of the rewarded stimulus (in LPFC: 22% with reward location information; n = 12 of 54 N-type cells, randomization test p<0.05 within [200 500] ms after cue onset; in ACC: 10% with reward location information; n = 5 of 50 N-type cells, n.s. randomization test) (*Figure 2—figure supplement 2C,D*).

The color-specific firing increase and the encoding of the rewarded color by *N-type* neurons in LPFC suggest they support reversal learning performance. We tested this by correlating their firing rates around the color cue onset with the trial-by-trial variation of the choice probability for choosing the stimulus with the rewarded color. Choice probability, *p(choice)*, was calculated with a reinforcement learning model that learned to optimize choices based on reward prediction errors (see *Equation 3* in Materials and methods and *Oemisch et al., 2019*). Choice probability was low (near ~0.5) early during learning and rose after each reversal to reach a plateau after around ~10 trials (*Figure 1C*, for example blocks, *Figure 2—figure supplement 3A*). We found that during the post-color onset time period 17% (n = 20 of 120) of *B-type* cells and 27% (n = 11 of 41) of *N-type* cells in LPFC significantly correlated their firing with p(choice), which was larger than expected by chance (binomial test B-type cells: p<0.001; *N-type* cells: p<0.001). On average, *N-type* cells in LPFC showed positive correlations (Pearson r = 0.068, Wilcoxon rank test, p=0.011), while *B-type* neurons showed on average no correlation (Wilcoxon rank test, p=0.20) (*Figure 2E*). The positive p(choice) correlations of *N-type* neurons in LPFC grew following color onset and remained significant for 0.7 s following color onset (N = 41 *N-type* neurons, randomization test, p<0.05 from 0 to 0.7 s post-cue, *Figure 2E*). *N-type* neurons in LPFC of both monkeys showed a similar pattern of response to the attention cue and positive correlation of firing rate with p(choice) (*Figure 2—figure supplement 4A–C*). Compared to LPFC, significantly less *N-type* cells in ACC correlated their firing with choice probability (6%, n = 2 of 33 in ACC, versus 27% in LPFC, $X^2$-test for prop. difference, $X^2$-stat = 5.45, p=0.019) and showed no *p(choice)* correlations over time (Wilcoxon rank test, p=0.49, n.s., *Figure 2F*).

## Putative interneurons in anterior cingulate cortex index high reward prediction errors

Choice probabilities (p(choice)) increase during reversal learning when reward prediction errors (RPEs) of outcomes decrease, which was evident in an anticorrelation of (p(choice)) and RPE of

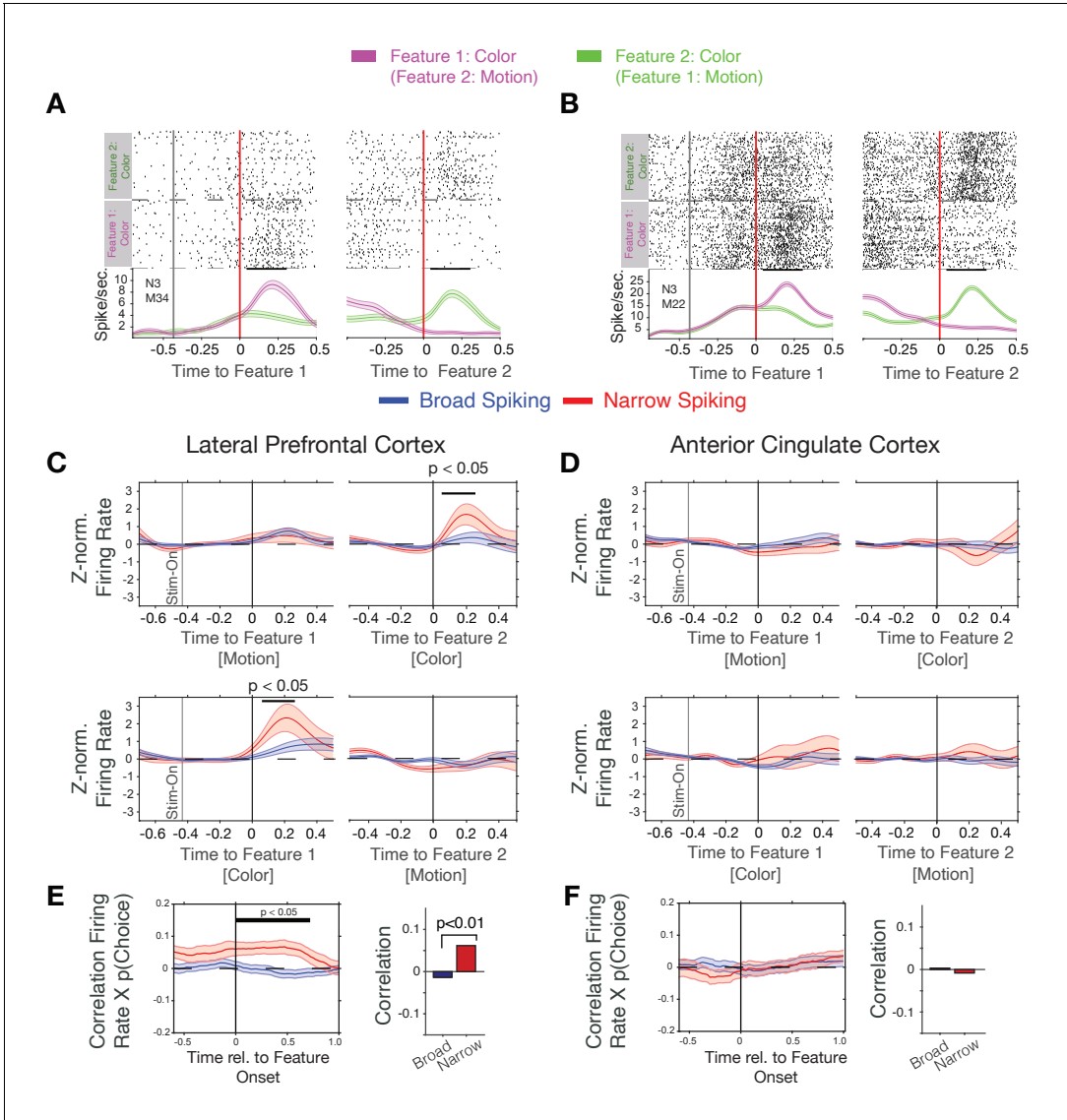

**Figure 2.** Firing rate modulation of narrow and broad spiking neurons to the color cue correlate with choice probability. (A, B) Spike rasters for example neurons around the onset of feature-1 and feature-2 when feature-1 was color (magenta) or motion (green). Both neurons responded stronger to the color than the motion onset irrespective of whether it was shown as first or as second feature during a trial. (C) Narrow spiking neurons (red) in LPFC respond to the color onset when it occurred as feature-2 (*upper panel*), or as feature-1 (*bottom panel*). (D) Same as c for the ACC shows no or weak feature onset responses. (E) Firing rates of narrow spiking neurons (red) in LPFC correlate with the choice probability of the to be chosen stimulus (*left*). The average Rate x Choice Probability correlation in LPFC was significantly larger in narrow than in broad spiking neurons (*right*). (F) Same as e for ACC shows no significant correlations with choice probability. *Source data 1* Correlation data and script for ploting panels E, and F.

The online version of this article includes the following figure supplement(s) for figure 2:

**Figure supplement 1.** Narrow spiking neuron examples responding to the color but not motion cue.

**Figure supplement 2.** Proportion of neurons encoding the rewarded color and location.

**Figure supplement 3.** Distribution of choice probabilities (p(Choice)) and reward prediction errors (RPEs) estimated by the reinforcement learning model (*see* Materials and methods).

**Figure supplement 4.** Cell-type-specific responses and correlations with p(choice) and reward prediction errors for each monkey separately.

r = −0.928 in our task (*Figure 2—figure supplement 3A,B*) with lower p(choice) (near ~0.5) and high RPE over multiple trials early in the reversal learning blocks when the animals adjusted to the newly rewarded color (*Figure 2—figure supplement 3E,F*). Prior studies have shown that RPEs are prevalently encoded in the ACC (*Kennerley et al., 2011*; *Oemisch et al., 2019*). We therefore reasoned that RPEs might preferentially be encoded by narrow spiking putative interneurons. First, we

analyzed *N-* and *B-type* cell responses to the reward. In both, LPFC and ACC, *N-* and *B-type* cells on average increased firing after the reward onset (p<0.05, randomization test, n = 26 of 54 and 18 of 188 *B- type* cells with increases, respectively, and n = 14 of 54 *N- type* and 5 of 188 *B-type* cells with decreased firing in LPFC, and n = 30 of 50 *N-type* and 13 of 216 *B- type* cells with increases, respectively, and n = 19 of 50 and 8 of 216 *B-type* cells with decreased firing in ACC). However, the *N-* and *B-type* responses to the reward were not significantly different in ACC or LPFC (ns., randomization test, *Figure 3A,B*). We estimated trial-by-trial RPEs with the same reinforcement learning model that also provided p(choice) for the previous analysis. RPE is calculated as the difference of received outcomes R and expected value V of the chosen stimulus (*see Materials and methods*). We found that on average 23% of LPFC and 35% of ACC neurons showed significant firing rate correlations with RPE in the post-outcome epoch with only moderately and non-significantly more *N-type* than *B-type* neurons having significant rate-RPE correlations (n = 9 *N-type* neurons, n = 31 *B-type* neurons, $X^2$-test; p=0.64 for LPFC; n = 15 *N-type* neurons, n = 47 *B-type* neurons, $X^2$-test; p=0.83 for ACC; *Figure 3C,D*). However, time-resolved analysis of the strength of the average correlations revealed a significant positive firing x RPE correlation in the 0.2–0.6 s after reward onset for ACC *N-type*

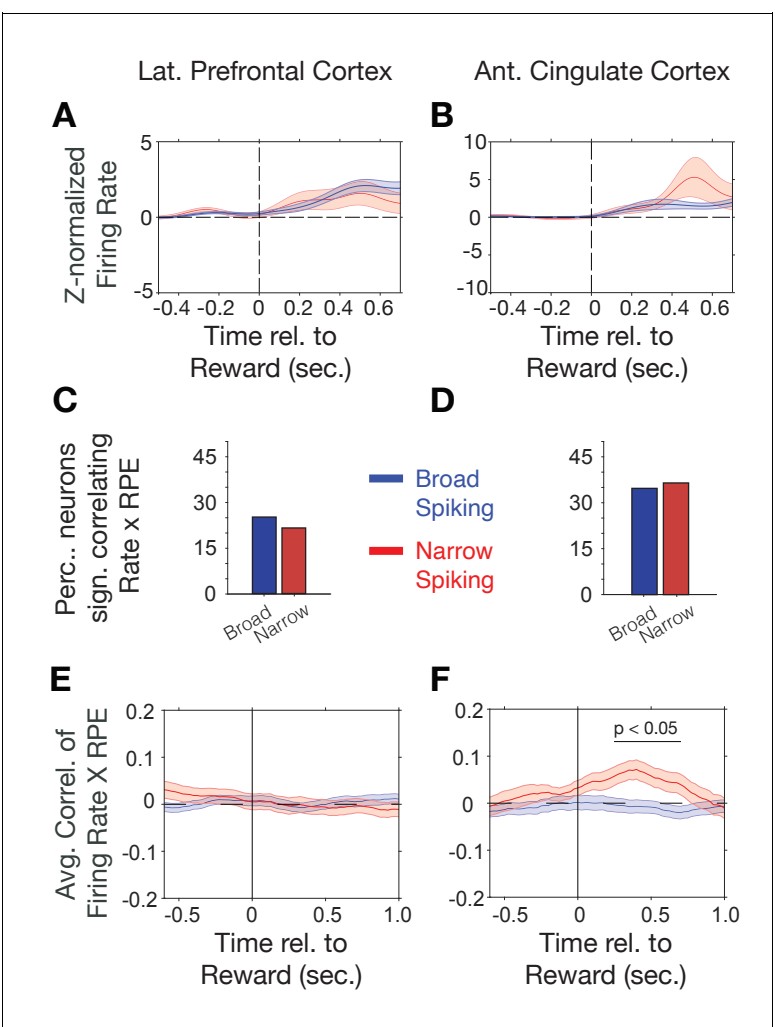

**Figure 3.** Firing rate modulation to trial outcomes correlate with reward prediction errors. (**A, B**) Narrow (red) and broad spiking neurons (blue) in LPFC (**A**) and ACC (**B**) on average activate to the reward outcome. (**C, D**) Proportion of narrow and broad spiking neurons in LPFC (**C**) and ACC (**D**) with significant firing rate X reward prediction error correlations in the [0 0.75] s after trial outcomes were received. (**E, F**) Time course of firing rate X reward prediction error correlations for narrow and broad spiking neurons in LPFC (*E*) and ACC (*F*) around the time of reward onset. Horizontal bar denotes time with significant correlations. *Source data 1* Correlation data and script for ploting panels E, and F.

neurons, which was absent in LPFC (ACC, n = 43 *N-type* neurons, randomization test p<0.05; LPFC: n = 31 *N-type* neurons, no time bin with sign.; *Figure 3E,F*). In ACC, the positive correlation of N-type neurons firing rate and RPE was evident in both monkeys (*Figure 2—figure supplement 4D*).

## Classification of neural subtypes of putative interneurons

We next asked whether the narrow spiking, putative interneurons whose firing indexed relatively lower p(choice) in LPFC and relatively higher RPE in ACC are from the same *electrophysiological* cell type, or *e-type* (*Markram et al., 2015*; *Gouwens et al., 2019*). Prior studies have distinguished different narrow spiking *e-types* using the cells' spike train pattern and spike waveform duration (*Ardid et al., 2015*; *Dasilva et al., 2019*; *Trainito et al., 2019*; *Banaie Boroujeni et al., 2020b*). We followed this approach using a cluster analysis to distinguish *e-types* based on spike waveform duration parameters (inferred hyperpolarization rate and time to 25% repolarization, *Figure 1—figure supplement 2A,B*), on whether their spike trains showed regular or variable interspike intervals (local variability '*LV*', *Figure 1—figure supplement 2D*), or more or less variable firing relative to their mean interspike interval (coefficient of variation '*CV*', *Figure 1—figure supplement 2C*). LV and CV are moderately correlated (r = 0.26, *Figure 1—figure supplement 2E*), with LV indexing the local similarity of adjacent interspike intervals, while CV is more reflective of the global variance of higher and lower firing periods (*Shinomoto et al., 2009*). We ran the k-means clustering algorithm on neurons in ACC and LPFC using variables mentioned above and their firing rate (details in Materials and methods). Clustering resulted in eight *e-types* (*Figure 4A–C*). Cluster boundaries were highly reliable (*Figure 4—figure supplement 1*). Moreover, the assignment of a cell to its class was statistically consistent, and reliably evident for cells from each monkey independently (*Figure 4—figure supplement 2*). Narrow spiking neurons fell into three *e-types*. The first narrow spiking *N1 e-type* (n = 18, 13% of narrow spiking neurons) showed high firing rates and highly regular spike trains (low LVs, mean LV 0.47, SE 0.05). The second *N2 e-type* (n = 27, 20% of narrow spiking neurons) showed on average Poisson spike train variability (LVs around 1) and the narrowest waveforms, and the *N3 e-type* (n = 91, 67% of all narrow spiking neurons) showed intermediate narrow waveform duration and regular firing (LV's < 1, mean LV 0.84, SE 0.02) (*Figure 4C*). Neurons within an *e-type* showed similar feature characteristics irrespective of whether they were from ACC or LPFC. For example, N3 *e-type* neurons from ACC and in LPFC were indistinguishable in their firing and action potential characteristics (LV$_{ACC / LPFC}$ = 0.79/0.88, ranksum-test, p=0.06; CV$_{ACC / LPFC}$ = 1.19/1.31, ranksum-test, p=0.07; Firing Rate$_{ACC/LPFC}$ = 4.41/4.29, ranksum-test p=0.71; action potential repolarization time (hyperpolarization rate)$_{ACC / PFC}$ = 0.18 sec. (97 s.$^{-1}$)/0.17 s. (93 s.$^{-1}$)).

Beyond the narrow spiking classes, spiketrains and LV distributions showed five broad spiking neuron *e-types*. The B1-B5 *e-types* varied from irregular burst firing in *e-types* B2, B3 and B4 (LV >1, class B2 mean LV 1.20, SE 0.02, class B3 mean LV 0.93, SE 0.02, class B4 mean 1.24, SE 0.03), regular firing in B1 (LV <1, class B1 mean LV 0.75, SE 0.02) to regular non-Poisson firing in B5 (LV >1, class B5 mean LV 1.68, SE 0.02) (number and % of broad spiking cells: B1: 109 (18%), B2: 103 (17%), B3: 94 (16%), B4: 146 (25%), B5: 138 (23%)) (*Figure 4B,C*). LV values > 1 indicate bursty firing patterns which is supported by a positive correlation of the LV of neurons with their probability to fire bursts defined as spikes occurring ≤5 ms apart (r = 0.44, p<0.001, *Figure 1—figure supplement 2F*). We next calculated the post- to pre- spike-triggered MUA modulation ratio for each of the *e-types*. Across all *e-types* only the spike-triggered MUA modulation ratio for the N3 *e-type* was different from zero (p<0.05, FDR-corrected) (*Figure 4D*). Comparison between cell classes showed that the spike-triggered MUA modulation ratio for the N3 *e-type* differed significantly from the B4 (p=0.02) and B5 (p=0.03) e-types.

## The same interneuron subclass indexes P(choice) in LPFC and RPE in ACC

The distinct *e-types* allowed testing how they correlated their firing with choice probability and with RPE. We found that the only *e-type* with a significant average correlation of firing and choice probability during the cue period was the *N3 e-type* in LPFC (r = 0.08, Kruskal Wallis test, p=0.04; randomization test difference to zero, Tukey-Kramer multiple comparison corrected, p<0.05; *Figure 5A,B*). Consistent with this correlation, neurons of the *N3 e-type* in LPFC also significantly increased firing

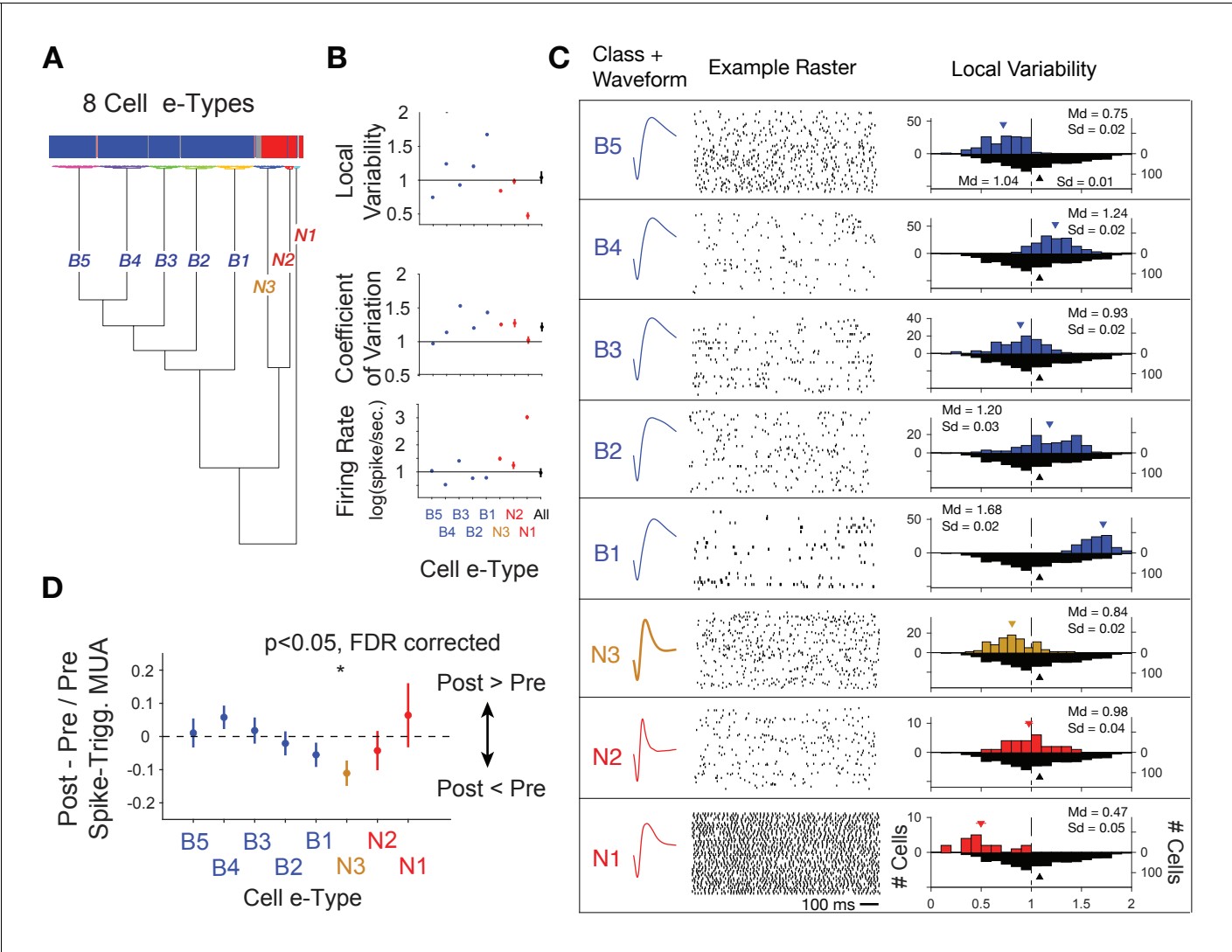

**Figure 4.** Clustering of *e-type* sub-classes of cells using their spike width, firing variability and rate. (A) Dendrogram of cluster distances for neuron classes with broad spikes (five subclasses, *blue*), and narrower spikes (three subclasses, *orange* and *red*). (B) For each e-type (*x-axis*) the average LV, CV and firing rate. The rightmost point shows the average for all *e-types* combined. (C) Illustration of the average spike waveform, spiketrain raster example, and Local Variability (LV, *upper* histograms) for each clustered *e-type*. The bottom grey LV histogram includes all recorded cells to allow comparison of *e-type* specific distribution. (D) The average post- to pre- spike MUA modulation (*y-axis*) for neurons of the different *e-types*. Values below 0 reflect reduced multiunit firing after the neuron fires a spike compared to before the spike, indicating a relative suppressive relationship. Only the N3 *etype* showed a systematically reduced post-spike MUA modulation. MUA were always recorded from other electrodes nearby the spiking neuron. *Source data 2* Data and script used for clustering (panel A) and data used for plotting panels B, and C.

The online version of this article includes the following figure supplement(s) for figure 4:

**Figure supplement 1.** Determining number of clusters.

**Figure supplement 2.** Clustering of neurons.

to the color cue, irrespective of whether the color cue appeared early or later in the trial (p<0.05 during 0.04–0.2 s after feature two onset, and p<0.05 during 0.175–0.225 s after feature one onset, *Figure 5—figure supplement 1*). The on-average positive correlation of firing rate and p(choice) was also evident in an example *N3 e-type* cell (*Figure 5—figure supplement 2A–C*). There was no other *e-type* in LPFC and in ACC showing significant correlations with choice probability. In LPFC, a linear classifier trained on multiclass p(choice) values was able to label N3 *e-type* neurons based on their p(choice) values with an accuracy of 31% (*Figure 5—figure supplement 3A*).

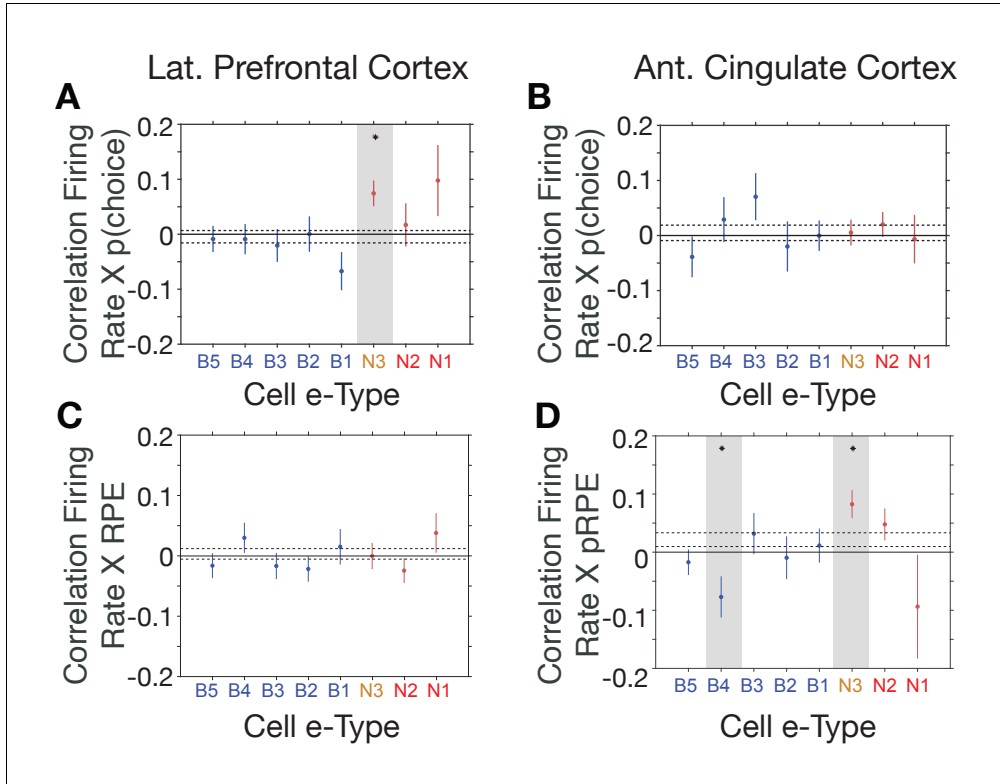

**Figure 5.** *E-type*-specific correlations with choice probability and reward prediction error in LPFC and ACC. (**A**, **B**) Firing Rate X Choice Probability correlations for neurons of each *e-type* subclass in LPFC (**A**) and ACC (**B**). Only the N3 *e-type* neurons in LPFC show significant correlations. (**C**, **D**) Firing Rate X Reward Prediction Error correlations for neurons of each *e-type* subclass in LPFC (**C**) and ACC (**D**). The N3 *e-type* neurons in ACC show significant positive correlations, and the B3 *e-type* shows negative firing rate x RPE correlations. Grey shading denotes significance at p<0.05 (multiple comparison corrected). Error bars are SE's. *Source data 1* Correlation data and script for ploting panels A-D.

The online version of this article includes the following figure supplement(s) for figure 5:

**Figure supplement 1.** Color-, motion-, and reward- onset firing rate modulation for each cell e-type in LPFC and ACC.

**Figure supplement 2.** N3 e-type single cell example of firing rate and p(choice) in LPFC, and of firing rate and RPE in ACC.

**Figure supplement 3.** Predicting cluster label of cells from their functional correlation values.

Similar to the N3 *e-type* in LPFC, in ACC it was the *N3 e-type* that was the only narrow spiking subclass with a significant functional firing rate correlation with reward prediction errors (RPE) (n = 30 neurons; r = 0.09, Kruskal Wallis test, p=0.01, randomization test for sign. difference to zero, Tukey-Kramer multiple comparison corrected p<0.05, *Figure 5C,D*). The only other *e-type* with a significant firing rate x RPE correlation was the B4 class which fired stronger with lower RPE's (n = 18 neurons; r = −0.08, Kruskal Wallis test, p=0.01, randomization test for sign. difference to zero, multiple comparison corrected p<0.05). There was no subtype-specific RPE correlation in LPFC (*Figure 5C,D*). The average positive correlation of firing rate and RPE was also evident in example ACC *N3 e-type* cells (*Figure 5—figure supplement 2D–F*). In ACC, a linear classifier trained on multiclass RPE values was able to label N3 *e-type* neurons from their RPE value with an accuracy of 34% (*Figure 5—figure supplement 3B*).

## Narrow spiking neurons synchronize to theta, beta, and gamma band network rhythms

Prior experimental studies have suggested that interneurons have unique relationships to oscillatory activity (*Puig et al., 2008*; *Cardin et al., 2009*; *Sohal et al., 2009*; *Vinck et al., 2013*;

*Womelsdorf et al., 2014a*; *Chen et al., 2017*; *Voloh and Womelsdorf, 2018*; *Shin and Moore, 2019*; *Banaie Boroujeni et al., 2020b*; *Onorato et al., 2020*), raising the possibility that the N3 *e-type* neurons realize their functional contributions to p(choice) and RPE processing also through neuronal synchronization. To discern this, we first inspected the spike-triggered LFP averages (STAs) of neurons and found that STAs of many N3 *e-type* neurons showed oscillatory sidelobes in the 10–30 Hz range (*Figure 6A*). We quantified this phase synchrony by calculating the spike-LFP pairwise phase consistency (PPC) and extracting statistically significant peaks in the PPC spectrum (*Vinck et al., 2012*; *Banaie Boroujeni et al., 2020a*), which confirmed the presence of significant synchrony peaks across theta/alpha, beta and low gamma frequency ranges (*Figure 6B*). The density of spike-LFP synchrony peaks, measured as the proportion of neurons that show reliable PPC peaks (see Materials and methods), showed a high prevalence of 15–30 Hz beta synchrony for broad spiking neurons in both, ACC and LPFC, a peak of ~5–12 Hz synchrony that was unique to ACC, and a high prevalence of 35–45 Hz gamma synchronization in narrow spiking cells (but not in broad spiking cells) in both areas (*Figure 6C*; *Voloh et al., 2020*). The synchrony peak densities of the N3 *e-type* neurons mimicked this overall pattern by showing beta to gamma band synchrony peak densities in LPFC and a 5–12 Hz theta/alpha and a gamma synchrony in ACC (*Figure 6C*) (for peak densities of other *e-types*, see *Figure 6—figure supplement 1*).

## Interneuron-specific gamma synchronization following cues in LPFC and outcomes in ACC

The overall synchrony patterns leave open whether the synchrony is task modulated or conveys information about choices and prediction errors. We addressed these questions by calculating spike-LFP

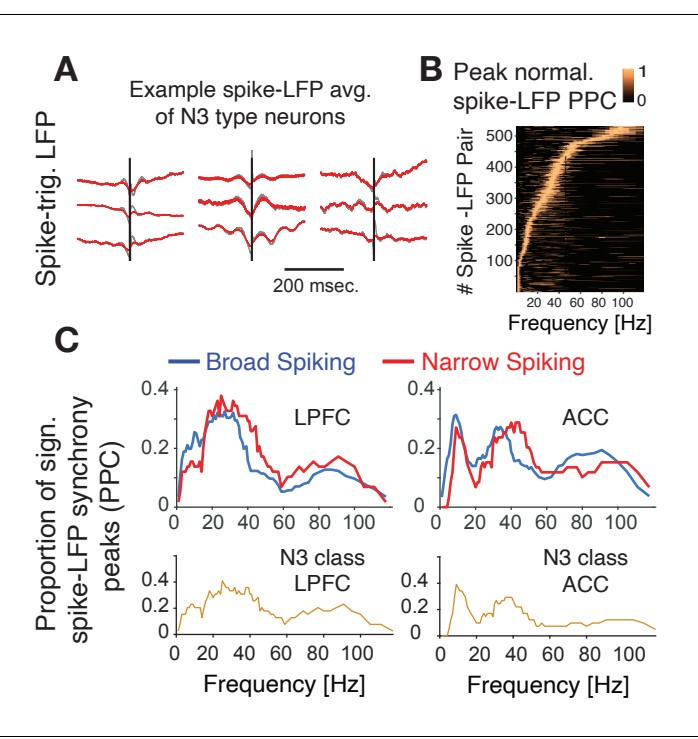

**Figure 6.** Spike-LFP phase synchronization. (**A**) Average spike-triggered local field potential fluctuations of nine N3 *e-type* neurons showing a transient LFP oscillations from 5 Hz up to ~30 Hz. Black vertical line is the time of the spike. The red lines denote the LFP after adaptive spike artifact removal (raw traces in gray). (**B**) Peak normalized pairwise phase consistency for each spike-LFP pair (*y-axis*) rank ordered according to the frequency (*x-axis*) with peak PPC. (**C**) Proportion of sign. peaks of spike-LFP synchronization for neurons in LPFC (*left*) and ACC (*right*) for narrow and broad spiking neurons (*upper rows*) and for the N3 *e-type* neurons (*bottom row*).

The online version of this article includes the following figure supplement(s) for figure 6:

**Figure supplement 1.** Spike-LFP synchronization for cell e-types in LPFC (A) and ACC (B).

phase synchronization time-resolved around the color cue onset (for LPFC) and around reward onset (for ACC) separately for trials with high and low choice probabilities (for LPFC) and high and low reward prediction errors (for ACC). We found in LPFC that the N3 *e-type* neurons showed a sharp increase in 35–45 Hz gamma band synchrony shortly after the color cue is presented and choice probabilities were low (i.e. when the animals were uncertain which stimulus is rewarded), while broad spiking neurons did not show gamma synchrony (*Figure 7A–C*) (N3 *e-type* vs broad spiking cell difference in gamma synchrony in the 0–700 ms after color cue onset: p<0.05, randomization test, multiple comparison corrected). When choice probabilities are high, N3 *e-type* neurons and broad spiking neurons in LPFC showed significant increases of 20–35 Hz beta-band synchronization (*Figure 7D,E*) with N3 *e-type* neurons synchronizing significantly stronger to beta than broad spiking neuron types (*Figure 7F*) (p<0.05 randomization test, multiple comparison corrected). These effects were restricted to the color cue period. LPFC broad spiking neurons and N3 *e-type* neurons did not show spike-LFP synchronization after the reward onset in low or high RPE trials (*Figure 7—figure supplement 1A–D*). Moreover, the gamma synchrony when p(choice) was low was not found in other narrow spiking or broad spiking *e-types* with the LPFC N3 *e-type* showing stronger gamma synchrony than broad spiking classes in the low p(choice) trials (p=0.02, Tukey-Kramer multiple comparison corrected) (*Figure 7—figure supplement 1E–F*). There was no difference in 35–45 Hz gamma synchrony of other cell classes in LPFC in the 0–0.7 s after reward onset in the high or low RPE trials, or around the (0.7 s) color onset in the high p(choice) trials (*Figure 7—figure supplement 1E–H*, see *Figure 7—figure supplement 2A* for time-frequency maps for all cell classes around cue onset).

In ACC, the N3 *e-type* neurons synchronized in a 35–42 Hz gamma band following the reward onset when RPE's were high (i.e. when outcomes were unexpected), which was weaker and emerged later when RPEs were low, and which was absent in broad spiking neurons (*Figure 8*). In contrast to

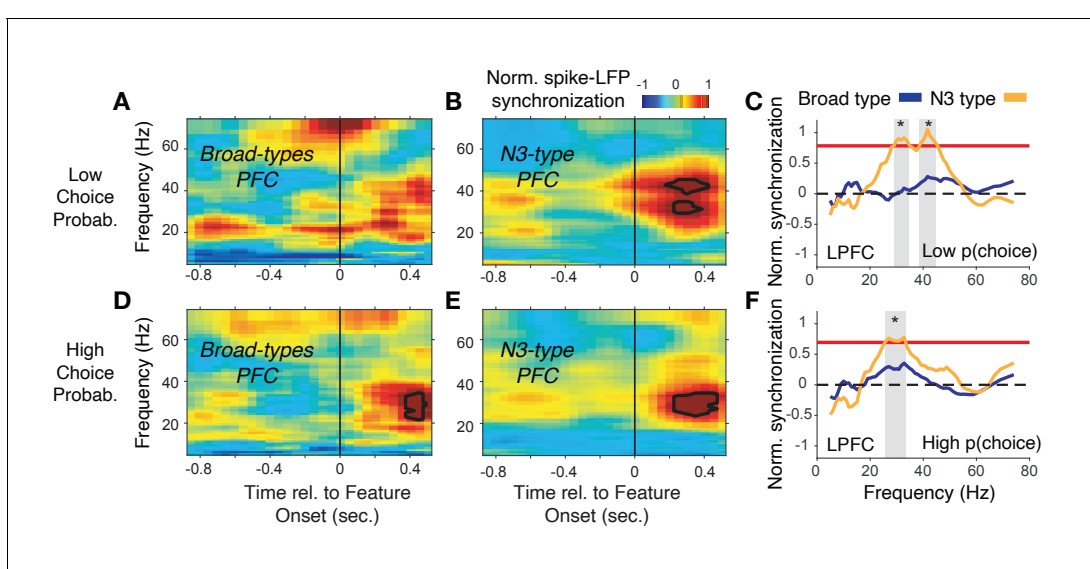

**Figure 7.** Spike-LFP phase synchronization in LPFC around the color onset for trials with low and high choice probability. (**A**) Spike-LFP pairwise phase consistency for broad spiking neurons in LPFC around the time of the color onset (*x-axis*) for trials with the 50% lowest choice probabilities. (**B**) Same as (*A*) for neurons of the N3 *e-type*. Black contour line denotes statistically significant increased phase synchrony relative to the pre-color onset period. (**C**) Statistical comparison of spike-LFP synchrony for N3 *e-type* neurons (orange) versus broad spiking neurons (blue) for low choice probability trials in LPFC. Synchrony is normalized by the pre-color onset synchrony. Gray shading denotes p<0.05 significant differences of broad and N3 type neurons. (**D,E,F**) Same format as (*A,B,C*) but for the 50% of trials with the highest choice probability. *Source data 3* Coherence data and script for ploting panels A-F.

The online version of this article includes the following figure supplement(s) for figure 7:

**Figure supplement 1.** Spike-LFP phase synchronization in LPFC around the reward onset for trials with low and high reward prediction error.
**Figure supplement 2.** Spike-LFP phase synchronization of *e-types* in LPFC and ACC during outcome processing for trials with low and high choice probabilities and reward prediction error.
**Figure supplement 3.** Spike-LFP phase synchronization in ACC during outcome processing for trials with low and high choice probabilities.
**Figure supplement 4.** Spike-LFP synchronization of broad spiking neurons and the N3 *e-type* in LPFC and ACC after subtracting event evoked LFP.

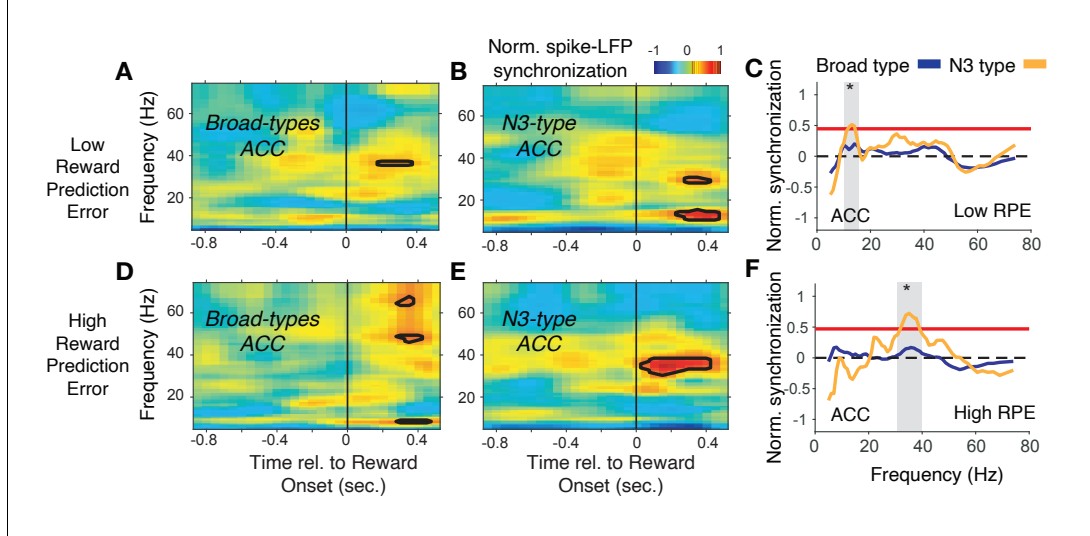

**Figure 8.** Spike-LFP phase synchronization in ACC during outcome processing for trials with low and high reward prediction errors. (A) Spike-LFP pairwise phase consistency for broad spiking neurons in ACC around reward onset (*x-axis*) for trials with the 50% lowest reward prediction errors. (B) Same as (A) for neurons of the N3 *e-type*. Black contour line denotes statistically significant increased phase synchrony relative to the pre-reward period. (C) Statistical comparison of the spike-LFP synchrony (normalized by the pre-reward synchrony) for N3 *e-type* neurons (orange) versus broad spiking neurons (blue) in ACC for trials ending in low reward prediction errors. Gray shading denotes frequencies with p<0.05 significant differences of broad spiking versus N3 *e-type* neurons. (D,E,F) Same format as (A,B,C) but for the 50% of trials with the highest high reward prediction error outcomes. *Source data 3* Coherence data and script for ploting panels A-F.

this gamma synchronization at high RPE, low RPE trials triggered increased spike-LFP synchronization at a ~ 6–14 Hz theta/alpha frequency in the N3 *e-type* neurons (*Figure 8C*). The increase of 6–14 Hz synchrony was significantly stronger in the N3 *e-type* than in broad spiking neurons in the 0 to 0.7 s post reward onset period (*Figure 8F*). These gamma and theta band effects of the N3 *e-type* neurons in ACC were restricted to the reward period, that is, they were absent in the color cue period for trials with high or low p(choice) (*Figure 7—figure supplement 3A–D*). Comparison to the other e-types showed that the N3 *e-type* significantly stronger gamma synchronized in the reward period when RPEs were high (p=0.04, Tukey-Kramer, multiple comparison corrected) (*Figure 7—figure supplement 3E*). Other *e-type* classes did not differ in their spike-LFP synchronization in this 35–45 Hz gamma band in low or high RPE trials with the exception of the B2 class in ACC that synchronized in high RPE trials at a higher >50 Hz gamma band (*Figure 7—figure supplement 3E–H*, see *Figure 7—figure supplement 2B* for time-frequency maps for all cell classes around reward onset).

The spike-LFP synchronization results in PFC and in ACC were unchanged when the average reward onset aligned LFP, or the average color-cue aligned LFP was subtracted prior to the analysis, which controls for a possible influence of lower frequency evoked potentials (*Figure 7—figure supplement 4*).

## Circuits model of interneuron-specific switches between gamma and beta or theta synchronization

The previous results showed that neurons of the N3 *e-type* engaged in a transient ~35–45 Hz gamma band synchronization during trials that were characterized by uncertainty. In LPFC gamma synchronization was evident when expected stimulus values were uncertain (reflected in low p(choice)), and in ACC gamma synchronization emerged when reward outcomes were uncertain (reflected in high RPE). In contrast, there was no gamma-band synchrony when choice probabilities were certain and reward outcomes predictable. In these trials, N3 *e-type* neurons rather showed beta synchronization to the cue (in LPFC), or theta band synchronization to the reward onset (in ACC). These findings indicate that oscillatory activity signatures inform us about the possible circuit motifs underlying uncertainty-related related computations. These computations are formally described in the reinforcement learning framework allowing us to propose a linkage of specific computations to oscillatory activity

signatures and their putative circuits as proposed in the *Dynamic Circuits Motif* framework (*Womelsdorf et al., 2014b*).

To show the feasibility of this approach we devised two circuit models that reproduces the gamma band activity signatures in LPFC and ACC using populations of inhibitory cells modeled to correspond to N3 e-type cells (for modeling details, see Appendix 1). First, we modeled a putative LPFC circuit. Here, N3 *e-type* neurons showed gamma synchronization when p(choice) was low which happens in trials in which the values of the two available objects are similar and the choice among them is difficult (*see Equation 3* in Materials and methods). We predicted in this situation gamma synchronization of the N3 *e-type* reflects resolving competition among inputs from similarly active, pyramidal cell populations encoding the expected values of the two objects. To test whether this scenario is plausible, we conceptualized and then simulated a circuit which modelled the activity of an N3 *e-type* neuron population that we presumed to be PV+ fast spiking basket cells (see Discussion) activated by two excitatory pyramidal cell populations (*Es*) whose activity scales with the value of the stimuli (*Figure 9A*). Such an *E-I* network can synchronize by way of mutual inhibition at beta or gamma frequencies depending on the total amount of drive the network receives (*Wang and Buzsáki, 1996*; *White et al., 1998*; *Tiesinga and José, 2000*). When both stimuli have similar values and the choice probability is relatively low, the drive to the network is high and it synchronizes in the gamma band. In contrast, when one of the objects has a value that is much larger than the other which results in high choice probabilities for that stimulus, it results in a net level of drive that makes the network synchronize in the beta band. We observed such a switch from gamma to beta frequencies in N3 *e-type* interneurons in LPFC when the choice probabilities changed from low to high

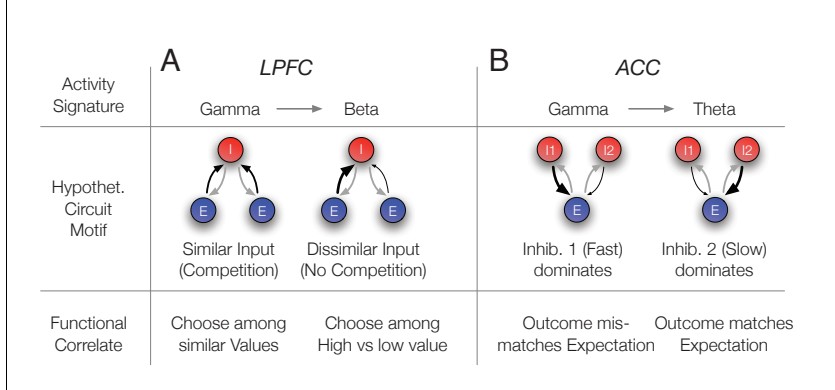

**Figure 9.** Hypothetical link of the observed gamma band synchronization of the N3 e-type to circuit motifs and their putative functional correlate. (**A**) The N3 e-type in LPFC synchronized at gamma when p(choice) was relatively low and at beta frequencies otherwise. The switch from gamma to beta synchronization can be parsimoniously reproduced in a circuit model with an interneuron (I) population receiving inputs from two excitatory (E) populations. When the input is diverse (similar p(choice)) a simulated circuit shows gamma activity (left) while when one excitatory population dominates it engages in beta synchronization (simulation details in Appendix 1). This activity signature could correspond at the functional level to choosing among similar valued stimuli (left) versus choosing stimuli with different values (bottom row). (**B**) In ACC the N3 e-type synchronized at gamma when the prediction error was large and at theta frequencies otherwise. The switch from gamma to theta synchronization can parsimoniously be reproduced in a circuit model with two I populations having different time constants and reciprocally connected to an E population. When the faster spiking I1 population is activated stronger, either directly from an external source, putatively by disinhibition of another interneuron population, the network synchronizes at gamma while otherwise the I2 neurons population imposes slower theta rhythmic synchrony to the network (simulation details in Appendix 1). Bottom: The activity states were functionally linked to those trials when outcomes mismatched expectations (high RPE) or matched the expected outcomes (low RPE).

The online version of this article includes the following figure supplement(s) for figure 9:

**Figure supplement 1.** E-E-I circuit simulation results: Gamma oscillations index similar excitatory input strength from two excitatory neuron populations to a fast spiking inhibitory neuron, whereas beta oscillations index diverse input strength (see also *Figure 9A*).

**Figure supplement 2.** E-I-I circuit simulation results: The circuit synchronizes at low or high frequencies depending on whether I1 interneurons are inhibited or released from inhibition (see also *Figure 9B*).

(*Figure 7*). In order to show that such gamma-to-beta switch can indeed follow from such a E-I network as a function of the diversity of inputs we ran simulations in a firing rate E-I model (*Keeley et al., 2017*), described in detail in Appendix 1, which reproduces the gamma-beta switch (*Figure 9—figure supplement 1*). The network model simulations suggest that the N3 *e-type* inhibition in LPFC after color-cue onset might accomplish two functions. It leads to a normalization that transforms the object value into a choice probability (a soft winner-take-all gating of values, see *Equation 3* in Materials and methods) and its gamma synchrony indexes resolving strong competition when similar excitatory drive originates from different sources (*Figure 9A*).

Secondly, we conceptualized and simulated a circuit model that reproduces the oscillatory findings in ACC where the N3 *e-type* neurons gamma-synchronized when outcomes were unexpected (high RPE) but synchronized in the theta band otherwise (low RPE). Such a gamma/theta switch is different to the gamma/beta switch seen in LPFC (*see* above). A parsimonious circuit realizing such a switch uses two separate interneuron populations (*Is*) that inhibit a common group of pyramidal cells (*Es*): A fast interneuron (*I1*) presumed to be PV+, corresponding to the N3 *e-type* (*see* Discussion), and a slower interneuron population (*I2*) (*Figure 9B*). When both are reciprocally connected with an excitatory population (*E*), an oscillatory regime emerges whose frequency varies depending on which interneuron population receives more excitatory drive (details in Appendix 1). When the *I1* population receives stronger drive, gamma frequency synchronization dominates the network, while a relatively stronger drive to the *I2* population causes neurons in the network to switch to slower, theta band synchronization. We documented this gamma/theta switching result in simulations of firing rate neurons in detail in the Appendix 1. The activity signatures of this E-I-I model resembles the empirical activity signatures. The theta synchronous activity that reflects the activity of *I2* neurons corresponds to low RPE trials, in which a reward *R* is received and the value *V* of the chosen stimulus was relatively high (a high *V* and a large *R*, the RPE is computed as = *R*-*V*) (see *Equation 1* in Materials and methods) (*Watabe-Uchida et al., 2017*). In contrast, the gamma synchronous state that emerged with larger drive to the *I1* neurons in the model corresponds to high RPE trials, in which a reward *R* is received, but the value *V* of the chosen stimulus was relatively low. This circuit motif is plausible when one assumes that the *I1* neuron population is disinhibited when the chosen stimulus value is low. Such a disinhibition can be achieved by lowering the drive to *I2* cells (which may require high values to be activated), or by assuming a separate disinhibitory circuit (for details see Appendix 1). In summary, the E-I-I motif reproduces the switch of gamma to theta synchronization we observed in ACC N3 *e-type* neurons. At the functional level, the circuit suggests that the emergence of gamma activity in this network indexes the detection of a mismatch between the received reward (as one source of excitation) and the chosen stimulus value (as another source of excitation) (*Figure 9B*).

The described circuits provide proofs-of-concept that the synchronization patterns we observed in the N3 *e-type* interneurons in ACC and LPFC during periods of uncertain values and outcomes can originate from biologically realistic circuits. The results justify future studies generating and testing quantitative predictions that can be derived from these circuit motifs.

## Discussion

We found that narrow spiking neurons in the medial and lateral prefrontal cortex of macaques cause a fast drop of local multiunit activity indicative of inhibitory interneurons. These putative interneurons in LPFC showed increased firing rates to the color-cue onset, encoded the rewarded color and correlated their rates with the choice probabilities, while in ACC their firing correlated with reward prediction errors during the processing of the reward outcome. These functional signatures were specifically linked to a putative interneuron subtype (*N3*) that showed intermediate narrow action potential waveforms and more regular firing patterns than expected from a Poisson process (LVs of N3 *e-type* neurons: 0.84). Moreover, this putative interneuron (N3) *e-type* engaged in prominent event-triggered 35–45 Hz gamma band synchronization in each of the recorded brain areas. In LPFC, the N3 *e-type* synchronized at gamma to the cue when choice probabilities were low and uncertain, and in ACC the N3 *e-type* synchronized at gamma to the reward onset when the RPE was high and the reward outcome was unexpected. Thus, the same *e-type* showed functional firing correlations and gamma synchrony in LPFC and in ACC during periods of uncertainty about cues and outcomes, respectively. Taken together, these findings point to a special role of the same type of

interneuron in LPFC and in ACC to realize their area specific functional contribution to the color-based reversal learning task. This interpretation highlights several aspects of interneuron specific circuit functions.

## Characterizing narrow spiking interneurons in vivo

The first implication of our findings is that narrow spiking neurons can be reliably subdivided in three subtypes based on their electrophysiological firing profiles. Distinguishing three narrow spiking neurons in vivo during complex task performance is a significant step forward to complement previous electrophysiological distinctions of three interneuron types in-vitro (*Zaitsev et al., 2009*; *Torres-Gomez et al., 2020*) or in vivo (*Ardid et al., 2015*; *Dasilva et al., 2019*; *Shin and Moore, 2019*; *Banaie Boroujeni et al., 2020b*), and complementing the finer-grained electrophysiological characterization of 'e-types' in-vitro that has been achieved with a rich battery of current injection patterns that are difficult to apply in the awake and behaving primate (*Markram et al., 2004*; *Monyer and Markram, 2004*; *Medalla et al., 2017*; *Gouwens et al., 2019*). This in vitro 'e-typing' has distinguished eleven (*Markram et al., 2015*) or thirteen (*Gouwens et al., 2019*) distinct interneuron e-types in rodent somatosensory and mouse visual cortex, respectively. In the visual cortex, these classes entailed six fast spiking subclasses showing variably transient, sustained or pause-delay response patterns (*Gouwens et al., 2019*). Notably, the fast spiking interneuron classes in that study were characterized by a low coefficient of variation (CV), low bursting reflective of a low Local Variability (LV), and a feature-importance analysis showed that the narrow action potential width and firing rate of these neurons were most diagnostic for separating the fast spiking from other neuron classes (Figure 2i, S9, and S14 in *Gouwens et al., 2019*). Our study used these diagnostic metrics (LV, CV, AP width and rate) directly for the clustering because we do not have the current injection responses available and distinguished three interneurons in the monkey compared to six fast spiking interneuron e-types in the mouse study. These results illustrate that our three interneuron e-types will encompass further subclasses that future studies should aim to distinguish in order to narrow the gap between the in vivo e-types that we and others report in the monkey, and the in-vitro e-types in the rodents that are more easily mapped onto specific molecular, morphological and genetic make-ups (*Markram et al., 2015*; *Gouwens et al., 2019*). As a caveat, this mapping of cell types between species might also reveal cell classes and unique cell class characteristics in nonhuman primate cortices that are not similarly evident in rodents as recently demonstrated in a cross-species study of non-fast spiking gamma rhythmic neurons in early visual cortex that were exclusively evident in the primate and not in mice (*Onorato et al., 2020*).

With regard to the specific interneuron e-types we believe that the N3 e-type that showed functional correlations in two areas encompasses mostly parvalbumin PV+ expressing neurons, because of their narrow spikes, regular inter-spike intervals and their propensity to synchronize at gamma, which resemble the regular firing and gamma synchrony described for PV+ cells in the rodent (*Cardin et al., 2009*; *Tiesinga, 2012*; *Stark et al., 2013*; *Amilhon et al., 2015*; *Chen et al., 2017*; *Gouwens et al., 2019*). Moreover, similar to the N3 e-type responses to the attention cue, rodent dorsomedial frontal PV+ neurons systematically activate to preparatory cues while somatostatin neurons respond significantly less (*Pinto and Dan, 2015*). However, PV+ neurons are heterogeneous and entail Chandelier cells and variably sized basket cells (*Markram et al., 2004*; *Markram et al., 2015*; *Gouwens et al., 2019*). It might therefore be an important observation that the N3 e-type was distinguished from other narrow spiking neurons by having a lower firing rate and an intermediate-narrow action potential shape as opposed to the narrowest waveform and highest firing rates that N1 e-types showed. The proposed tentative suggestion that N3 e-type neurons will be mostly PV+ cells also entails for the primate brain that they would not be part of calretinin (CR+) or calbindin (CB+) expressing cells as their expression profiles do not apparently overlap (*Dombrowski et al., 2001*; *Medalla and Barbas, 2009*; *Raghanti et al., 2010*; *Torres-Gomez et al., 2020*).

## What is the circuit role of the N3 interneuron e-type?

Assuming that N3 e-type neurons are partly PV+ neurons, we speculate that this translates into gamma rhythmic inhibition of local circuit pyramidal cells close to their soma where they impose output gain control (*Tiesinga et al., 2004*; *Bartos et al., 2007*; *Womelsdorf et al., 2014b*;

*Tremblay et al., 2016*). In our task, such local inhibition was linked to how uncertain the expected values of stimuli were (reflected in low choice probabilities) or how unexpected reward outcomes were (reflected in high RPE's). These conditions are periods that require a behavioral adaptation for which N3 *e-type* mediated inhibition could be instrumental. For example, in LPFC pyramidal cells that encoded the rewarded color in trials prior to the un-cued reversal become irrelevant when the reversal links reward to the alternative color and hence need to be suppressed during the reversal. This suppression of neurons encoding the previously relevant but now irrelevant color might be realized through activation of the N3 *e-type* neuron. Similarly, the N3 *e-type* activation in ACC reflects a rise in inhibition when an unexpected outcome (high RPE) is detected. This activation might therefore facilitate the updating of value expectations to reduce future prediction errors (*Sutton and Barto, 2018*; *Oemisch et al., 2019*).

The described, putative functions of N3 e-type activity provide direct suggestions on how they might contribute to transform inputs to outputs in a neural circuit. To understand this process, we devised and simulated circuit models of the activity signatures of inhibitory cells for the LPFC and the ACC (*Figure 9*, Appendix 1). For LPFC, we devised an E-E-I circuit where the interneuron (I) population synchronized at gamma when the excitatory drive of two E-cell populations was similar (Appendix 1, *Figure 9—figure supplement 1*). This situation mimics the situation when the values of two objects are similar, resulting in a low choice probability. According to this circuit, the function of I cells that putatively correspond to the N3 *e-type* neurons in LPFC is twofold. They normalize the activity of the excitatory cells, and they are instrumental in gating the activity of one over the other excitatory cell population when there is competition among them. Such competition arises specifically when choice probabilities are low because the low p(choice) indicates that the expected values of the stimuli to choose from are similar which makes a choice difficult. We therefore speculate that the putative circuit function of the N3 *e-type* cells in LPFC is the gating of competing excitatory inputs (*Figure 9A*).

For ACC, we devised an E-I-I circuit where the population of the N3 *e-type* putatively corresponded to one population of fast spiking inhibitory neurons (*I1*) that synchronized to gamma when receiving stronger excitatory drive than another population of slower inhibitory neurons (*I2*) (*Figure 9—figure supplement 1B*). The enhanced excitation of the *I1* over the *I2* population was modeled to correspond to trials with high RPE, which occurred when a reward (*R*) was received but the expected value (*V*) of the chosen stimulus was relatively low (a large RPE defined as the difference of *R-V*). In this situation, a stronger excitatory drive and consequently a gamma synchronous activity, could follow from disinhibiting the *I1* population. Such a disinhibition could originate from reduced inhibition from the *I2* cells in trials with low stimulus value, or it could originate from disinhibition from other neurons. These scenarios deserve explicit testing in future studies (for further discussion, see Appendix 1). They gain plausibility from anatomical studies that report that a large proportion of connections to interneurons go to disinhibitory interneurons that express calretinin and are distinct from the fast-spiking PV+ neurons that more likely entail the N3 *e-type* neurons (*Medalla and Barbas, 2009*; *Medalla and Barbas, 2010*). In summary, the proposed circuit model for the ACC suggests that the N3 *e-type* neurons activate when there is a mismatch of reward and chosen value. Activation of the N3 *e-type* neurons may thus be a (bio-) marker that predictions need to be updated to improve future performance.

We acknowledge that the proposed circuit models represent merely a proof-of-concept that says that the neuronal activities can originate in reasonable and previously described E-I motifs. They are not full biophysical implementations of the actual reversal learning task and entail finer predictions that await quantitative testing in future studies. They motivate combined electrophysiological and optogenetic studies in the primate to clarify cell-type-specific circuit functions during higher cognitive operations.

## Interneuron-specific gamma synchronization: Comparison to previous studies

Two major findings of our study pertain to spike-LFP gamma band synchronization. First, we found that N3 *e-type* neurons showed an event-triggered synchrony increase in the same 35–45 Hz gamma frequency band in both LPFC and ACC when there was uncertainty about the correct choice (low p(choice) or about the outcomes (high RPE) [see *Figures 7C* and *8F*]). Synchronization of the N3 *e-type* switched from a gamma frequency to the beta frequency in LPFC when the choices became

more certain, and to the theta frequency in ACC when outcomes became more certain. An intrinsic propensity for generating gamma rhythmic activity through, for example GABA$_a$ergic time constant, is well described for PV+ interneurons (*Wang and Buzsáki, 1996*; *Bartos et al., 2007*; *Womelsdorf et al., 2014b*; *Chen et al., 2017*) and is a documented activity signature even at moderate excitatory feedforward drive that might be more typical for prefrontal cortices than earlier visual cortices (*Cardin et al., 2009*; *Vinck et al., 2013*; *Shin and Moore, 2019*; *Onorato et al., 2020*).

Our findings provide strong empirical evidence that narrow spiking interneurons are the main carriers of gamma rhythmic activity in nonhuman primate prefrontal cortex during cue and outcomes processing (*Whittington et al., 2000*; *Hasenstaub et al., 2005*; *Bartos et al., 2007*; *Hasenstaub et al., 2016*; *Chen et al., 2017*; *Shin and Moore, 2019*). This conclusion resonates well with rodent studies that document how interneurons in infra-/peri-limbic and cingulate cortex engage in gamma synchrony (*Fujisawa and Buzsáki, 2011*; *Cho et al., 2015*).

The second major implication of the gamma synchronous N3 *e-type* neurons is that gamma band synchrony was associated with task epochs in which neural circuits realize a circuit function that can be considered to be 'area specific'. In LPFC, the gamma increase was triggered by the color-cue onset of two peripherally presented stimuli that instructed covertly shifting attention. Our circuit model (*Figure 9A*) illustrates that cue related gamma was restricted to periods when object values were similar, and the animal still learned which object is most reward predictive. The control of learning what is relevant during cognitively demanding tasks is a key function of the LPFC, suggesting that gamma activity emerges when this key function is called upon (*Miller and Cohen, 2001*; *Szczepanski and Knight, 2014*; *Cho et al., 2020*). A similar scenario holds for the ACC whose central function is often considered to monitor and evaluate task performance and detect when outcomes should trigger a change in behavioral strategies (*Shenhav et al., 2013*; *Heilbronner and Hayden, 2016*; *Alexander and Brown, 2019*; *Fouragnan et al., 2019*). In ACC, the gamma increase was triggered by an unexpected, rewarded outcome (high RPE). Thus, the N3 *e-type* specific gamma band signature occurred specifically in those trials with conflicting stimulus values requiring behavioral control to reduce the prediction errors through future performance (*Figure 9A*). Considering this ACC finding together with the LPFC finding suggests that gamma activity of N3 *e-type* neurons indexes a key function of these brain areas, supporting recent causal evidence from rodent optogenetics (*Cho et al., 2020*).

Consistent with the proposed importance of interneurons for area-specific key functions prior studies have documented the functional importance of inhibition in these circuits. Blocking inhibition with GABA antagonists like bicuculline not only renders fast spiking interneurons nonselective during working memory tasks but abolishes the spatial tuning of regular spiking (excitatory) cells during working memory tasks in monkeys (*Sawaguchi et al., 1989*; *Rao et al., 2000*), disturbs accuracy in attention tasks (*Paine et al., 2011*) and reduces set shifting flexibility by enhancing perseveration (*Enomoto et al., 2011*). Similarly, abnormally enhancing GABAa levels via muscimol impairs working memory and set shifting behavior (*Rich and Shapiro, 2007*; *Urban et al., 2014*) and can result in either maladaptive impulsive behaviors (*Paine et al., 2015*), and when applied in anterior cingulate cortex to perseveration (*Amiez et al., 2006*). Thus, altered medial and lateral prefrontal cortex inhibition is closely linked to an inability to adjust attentional strategies given unexpected outcomes. This evidence supports our studies suggestion of the importance of inhibitory neuron involvement in resolving uncertainties during adaptive behaviors.

Taken together, our interneuron-specific findings in primate LPFC and ACC stress the importance of interneurons to influence circuit activity beyond a mere balancing of excitation. Multiple theoretical accounts have stressed that some types of interneurons 'control information flow' (*Fishell and Kepecs, 2020*), by imposing important filters for synaptic inputs to an area and gain-control the output from that area (*Akam and Kullmann, 2010*; *Kepecs and Fishell, 2014*; *Womelsdorf et al., 2014b*; *Roux and Buzsáki, 2015*; *Cardin, 2018*). Testing these important circuit functions of interneurons has so far been largely limited to studies using molecular tools. Our study addresses this limitation by characterizing putative interneurons, delineating their suppressive effects on the circuit and highlighting their functional activation during reversal learning. The observed interneuron-specific, gamma synchronous coding of choice probabilities and prediction errors lends strong support to study cell-type-specific circuit mechanisms of higher cognitive functions.

## Materials and methods

All animal care and experimental protocols were approved by the York University Council on Animal Care (ethics protocol 2015–15 R2) and were in accordance with the Canadian Council on Animal Care guidelines.

### Electrophysiological recording

Data was collected from two male rhesus macaques (*Macaca mulatta*) from the anterior cingulate cortex and lateral prefrontal cortex as described in full in *Oemisch et al., 2019*. Extracellular recordings were made with tungsten electrodes (impedance 1.2–2.2 MOhm, FHC, Bowdoinham, ME) through rectangular recording chambers implanted over the right hemisphere. Electrodes were lowered daily through guide tubes using software-controlled precision micro-drives (NAN Instruments Ltd., Israel). Wideband local field potential (LFP) data was recorded with a multi-channel acquisition system (Digital Lynx SX, Neuralynx) with a 32 kHz sampling rate. Spiking activity was obtained following a 300–8000 Hz passband filter and further amplification and digitization at a 32 kHz sampling rate. Sorting and isolation of single unit activity was performed offline with Plexon Offline Sorter, based on the first two principal components of the spike waveforms and the temporal stability of isolated neurons. Only well-isolated neurons were considered for analysis (*Ardid et al., 2015*). Experiments were performed in a custom-made sound attenuating isolation chamber. Monkeys sat in a custom-made primate chair viewing visual stimuli on a computer monitor (60 Hz refresh rate, distance of 57 cm) and performing a feature-based attention task for liquid reward delivered by a custom-made valve system in *Oemisch et al., 2019*.

### Anatomical reconstruction of recording locations

Recording locations were identified using MRI images obtained following initial chamber placement. During MR scanning, we placed a grid marking the chamber center and peripheral positions as well as a diluted iodine solution inside the chamber for visualization. This allowed the referencing of target regions to the chamber center in the resulting MRI images. The positioning of electrodes was estimated daily using the MRI images and audible profiles of spiking activity. The relative coarseness of the MRI images did not allow us to differentiate the specific layer of recording locations in lateral prefrontal and anterior cingulate cortices.

### Task paradigm

The task (*Figure 1*) required centrally fixating a dot and covertly attending one of two peripherally presented stimuli (5° eccentricity) dependent on color-reward associations. Stimuli were 2.0° radius wide block sine gratings with rounded-off edges, moving within a circular aperture at 0.8 °/s and a spatial frequency of 1.2 (cycles/°). Color-reward associations were reversed without cue after 30 trials or until a learning criterion was reached, which makes this task a color-based reversal learning task.

Each trial began with the appearance of a gray central fixation point, which the monkey had to fixate. After 0.5–0.9 s, two black/white gratings appeared to the left and right of the central fixation point. Following another 0.4 s the two stimulus gratings either changed color to green and red (monkey K: cyan and yellow), or they started moving in opposite directions up and down, followed after 0.5–0.9 s by the onset of the second stimulus feature that had not been presented so far, for example if after 0.4s the grating stimuli changed color then after another 0.5–0.9 s they started moving in opposite directions. After 0.4–1 s either the red and green stimulus dimmed simultaneously for 0.3 s or they dimmed separated by 0.55 s, whereby either the red or green stimulus could dim first. The dimming of the rewarded stimulus represented the GO cue to make a saccade to one of two response targets displayed above and below the central fixation point. The dimming of the no-rewarded stimulus thus represented a NO-GO cue triggering the withholding of a response and waiting until the rewarded stimulus dimmed. The monkeys had to keep central fixation until this dimming event occurred.

A saccadic response following the dimming was rewarded if it was made to the response target that corresponded to the (up- or down-ward) movement direction of the stimulus with the color that was associated with reward in the current block of trials, for example if the red stimulus was the currently rewarded target and was moving upward, a saccade had to be made to the upper response target at the time the red stimulus dimmed. A saccadic response was not rewarded if it was made to

the response target that corresponded to the movement direction of the stimulus with the non-reward associated color. Hence, a correct response to a given stimulus must match the motion direction of that stimulus as well as the timing of the dimming of that stimulus. This design ensures the animal could not anticipate the time of dimming of the current target stimulus (which could occur before, after, or at the same time as the second stimulus), and thus needed to attend continuously until the 'Go-signal' (dimming) of that stimulus occurred. If dimming of the target stimulus occurred after dimming of the second/distractor stimulus, the animal had to ignore dimming of the second stimulus and wait for dimming of the target stimulus. A correct response was followed by 0.33 ml of water reward.

The color-reward association remained constant for 30 to a maximum of 100 trials. Performance of 90% rewarded trials (calculated as running average over the last 12 trials) automatically induced a block change. The block change was un-cued, requiring monkeys to use the trial's reward outcome to learn when the color-reward association was reversed. Reward was delivered deterministically.

In contrast to color, other stimulus features (motion direction and stimulus location) were only randomly related to reward outcome – they were pseudo-randomly assigned on every trial. This task ensured that behavior was guided by attention to one of two colors, which was evident in monkeys choosing the stimulus with the same color following correct trials with 89.5% probability (88.7%/90.3% for monkey H/K), which was significantly different from chance (t-test, both p<0.0001).

Monkeys performed the task at 83/86% (monkey's H/K) accuracy (excluding fixation break errors). The 17/14% of errors were composed on average to 50/50% of erroneous responding to the dimming of the distractor when it dimmed before the target and 34/37% of erroneous responding at the time when target and distractor dimmed simultaneously but the monkey chose the distractor direction, and 16/13% of error were responses when the target dimmed before any distractor dimming and the choice was erroneously made in the direction of the distractor.

## Behavioral analysis of the animal's learning status

To characterize the reversal learning status of the animals, we determined the trial during a block when the monkey showed consistent above chance choices of the rewarded stimulus using the expectation maximization algorithm and state–space framework introduced by *Smith et al., 2004*, and successfully applied to reversal learning in our previous work (*Balcarras et al., 2016*; *Hassani et al., 2017*; *Oemisch et al., 2019*). This framework entails a state equation that describes the internal learning process as a hidden Markov or latent process and is updated with each trial. The learning state process estimates the probability of a correct (rewarded) choice in each trial and thus provides the learning curve of subjects. The algorithm estimates learning from the perspective of an ideal observer that takes into account all trial outcomes of subjects' choices in a block of trials to estimate the probability that the single trial outcome is reward or no reward. This probability is then used to calculate the confidence range of observing a rewarded response. We identified a 'Learning Trial' as the earliest trial in a block at which the lower confidence bound of the probability for a correct response exceeded the p=0.5 chance level.

## Reinforcement learning modeling to estimate choice probability and expected value of color

The color reversal task required monkeys to learn from trial outcomes when the color reward association reversed to the alternate color. This color-based reversal learning is well accounted for by an attention augmented Rescorla Wagner reinforcement learning model ('*attention-augmented RL*') that we previously tested against multiple competing models (*Balcarras et al., 2016*; *Hassani et al., 2017*; *Oemisch et al., 2019*). Here, we use this model to estimate the trial-by-trial fluctuations of the expected value for the rewarded color, the choice probability p(choice) of the animal's stimulus selection and the positive reward prediction error (RPE, 'R-V', see *Equation 1*, *below*). P(choice) increased and RPE decreased with learning similar to the increase in the probability of the animal to make rewarded choices (*Figure 2—figure supplement 3*). They were highly anticorrelated (r = −0.928) (*Figure 2—figure supplement 3A*).

The attention augmented RL is a standard Q Learning model with an added decay constant that reduces the value of those features that are part of the non-chosen (i.e. non-attended) stimulus on a

given trial. On each trial $t$ this model updates the value $V$ for features $i$ of the chosen stimulus according to

$$V_{i,t+1} = V_{i,t} + \eta\left(R_t - V_{i,t}\right) \tag{1}$$

where $R$ denotes the trial outcome (0=non-rewarded, 1=rewarded) and $\eta$ is the learning rate bound to [0 1]. For the same trial, the feature values $i$ of the non-chosen stimulus decay according to

$$V_{i,t+1} = (1-\omega)V_{i,t} \tag{2}$$

With ω denoting the decay parameter. Following these value updates, the next choice $C_{t+1}$ is made by a softmax rule according to the sum of values that belongs to each stimulus. We indicate the stimulus by the index j and the set of feature values that belong to it by set s$_j$, (for instance, color x, location y, direction z):

$$P(C_{t+1} = j) = \frac{\exp\left(\beta\sum_{i \in s_j}V_{i,t}\right)}{\sum_j \exp\left(\beta\sum_{i \in s_j}V_{i,t}\right)} \tag{3}$$

*Equation 4* defines the choice probability, or p(choice), that is used for the neuronal analysis of this manuscript (*Sutton and Barto, 2018*). P(choice) increases with trials since reversal (*Figure 2— figure supplement 3D*), indicating a reduction in the uncertainty of the choice the more information is gathered about the value of the stimuli.

We optimized the model by minimizing the negative log likelihood over all trials using up to 20 iterations of the simplex optimization method to initialize the subsequent call to fmincon matlab function, which constructs derivative information. We used an 80/20% (training/test dataset) cross-validation procedure repeated for n = 50 times to quantify how well the model predicted the data. Each of the cross-validations optimized the model parameters on the training dataset. We then quantified the log-likelihood of the independent test dataset given the training datasets optimal parameter values. The cross-validation results were compared across multiple models in a previous study (*Oemisch et al., 2019*). Here, we used the best-fitting model based on this prior work.

## Waveform analysis

We initially analyzed 750 single units and excluded 24 units that showed double troughs or those that had overall less than 50 spike number. We then analyzed 726 highly isolated cells in ACC (397 cells), and PFC (149 cells area 8, and 180 cells dLPFC). We trough-aligned all action potentials (AP) and normalized them to the range of −1(trough) to 1 (peak). APs were then interpolated from their original time-step of 1/32000 s to a new time step of 1/320000 s. To characterize AP waveforms, we initially computed three different measures of Trough to Peaks (T2P) and Time for Repolarization (T4R) and Hyperpolarization Rate (HR) according to *Equation 4-6*:

$$T2P = \left(t_{trough} - t_{peak}\right) \tag{4}$$

$$T4R = \left(t_{0.75xpeak} - t_{peak}\right) \tag{5}$$

$$HR = \frac{1}{t_{Vpeak} - t_{V0.63xpeak}} \tag{6}$$

where t$_{peak}$ is time of the most positive value (peak) of the spike waveform, t$_{trough}$ is time of the most negative value of the spike waveform, $t_{0.75xpeak}$ is the time of spike waveform after the peak with a voltage equal to 75% of the peak and $t_{V0.63xpeak}$ is the time of the spike waveform before the peak with a voltage value equal to 63% of the peak (*Figure 1—figure supplement 2A,B*). We performed Hartigan's dip test was to test the unimodality hypothesis of distributions (P<0.05). HR and T2P were highly correlated (r=-.76). We chose HR as it was able to reject the Hartigan's dip test null hypothesis of distribution unimodality (P=0.01). We then used HR and T4R to characterize waveform

dynamics. T4R interval likely describes dynamics of the waveform in a period that calcium activated potassium channels are activated and most voltage-gated potassium channels are closed. While, HR reflects a time interval that most of sodium channels are closed and potassium channels have greater contribution to the dynamics of the waveform (*Bean, 2007*).

Both T4R and HR and their first component of the PCA were fitted with a bi-modal Gaussian distribution. We applied Akaike's and Bayesian information criteria for the two vs one Gaussian fits to select the best fit to the waveform measures.

### Data analysis

Analysis of spiking and local field potential activity was done with custom MATLAB code (Mathworks, Natick, MA), utilizing functions from the open-source Fieldtrip toolbox (http://www.ru.nl/fcdonders/fieldtrip/).

For all statistical tests that were performed on time-series, we used permutation randomization test and multiple comparisons with both primary and secondary alpha level of 0.05, unless the type of multiple comparison correction is explicitly mentioned.

### Spike-triggered multiunit modulation

We used spike-triggered multiunit analysis to estimate whether its spiking increased or decreased concomitantly with the surrounding neural activity – measured on a different electrode located ~200-450 µm from the electrode measuring the spiking activity. To compute the relative multi-unit activity (MUA) of the signal before and after spike occurrences, we used the Wide-Band signal and bandpass filtered the signal to a frequency range of [800 3000] Hz. The signal was then rectified to positive values. For each single unit, we extracted a period of [-50 50] ms around each spike aligned to the spike trough and estimated the power time-course of the signal using a sliding median filter window (window length=5 ms) over the extracted signal every 0.5 ms. For a given single unit, we computed the Z-transformation of each spike-aligned median filtered peak-amplitude by subtracting its mean and dividing by its standard deviation. This step normalized the MUA around the spike times. We then computed the average Z-transformed MUA across all spikes for each single unit. To compare the post spike MUA to pre-spike MUA, we computed the spike triggered MUA modulation ratio (SMUM) according to equation $SMUM = \frac{MUA_{post} - MUA_{pre}}{MUA_{pre}}$. Pre-spike MUA was the mean in a period of 10 ms before the spike and the Post-spike MUA was the mean in a period of 10ms after the spike.

For comparison of spike-triggered MUA modulation of broad vs narrow spiking neurons, we used the Wilcoxon test on the computed ratio, under the null hypothesis that there is no difference of MUA strength before and after the spike occurrence for narrow vs broad spiking neurons. We also performed the test on each individual group compare with population.

We also tested whether spike-triggered MUA modulation differed varies with the distance of the electrode tip that measured the spike providing neuron and the electrode that measured the MUA, but found no distance dependency (Wilcoxon test, n.s.).

### Analysis of firing statistics

To analyze firing statistics of cells, we followed procedures described in by *Ardid et al., 2015*, and for each neuron we computed the mean firing rate (FR), Fano factor (FF, mean of variance over mean of the spike count in consecutive time windows of 100 ms), the coefficient of variation (CV, standard deviation over mean of the inter-spike intervals, *Figure 1—figure supplement 2C*), and a measure of local variability of spike trains called the local variation (LV, *Figure 1—figure supplement 2D*). LV measures the regularity/burstiness of spike trains. It is proportional to the square of the difference divided by sum of two consecutive inter-spike intervals (*Shinomoto et al., 2009*).

### Cell clustering technique

We followed procedures described in *Ardid et al., 2015*, with minor adjustments to test whether neurons fall into different clusters according to the dynamics of their waveform dynamic measures and their firing statistics. For main clustering analysis, we used the K-Means clustering algorithm MATLAB/GNU Octave open-source code, freely available in public Git repository https://bitbucket.org/sardid/clusteringanalysis. We used the K-Means clustering algorithm to characterize subclasses of cells within the dataset upon the Euclidian distances of neuronal measures. We initially used three

measures of the waveform: Hyperpolarization Rate, Time for repolarization, and their first component of PCA. For the firing statistic measures we used local variation, coefficient of variance, Fano factor, and firing rate. The k-Means clustering algorithm is sensitive to duplicated and uninformative measures. We set a criterion of. 9 of Spearmans' correlation coefficient to exclude measures that were highly correlated (1st PCA was excluded). To reduce the biases upon on variable magnitudes, we z-score transformed each measure and normalized it to a range of [0 1]. We then computed the percent of variance explained by each measure from overall variance in our data. The measures were sorted based on their explaining variance of the overall variance within data. To disregard uninformative measures, a cut-off criterion of 90% were set to the cumulated sorted variance explained across measures. The Fano Factor was excluded based on this criterion from the k-Means clustering (*Figure 4—figure supplement 2A*).

## Determining cluster numbers

We used a set of statistical indices to determine a range of number of clusters that best explains our data. These indices evaluate the quality of the k-means clustering (*Ardid et al., 2015*): Rand, Mirkin, Hubert, Silhouette, Davies-Bouldin, Calinski-Harabasz, Hartigan, Homogeneity and Separation indexes (*Figure 4—figure supplement 1A*). We then run 50 replicates of k-means clustering for k = 1–40 number of clusters. For each k, we chose the best replicate based on the minimum squared Euclidian distances of all cluster elements from their respective centroids. While validity measures were improved by increasing number of clusters, the benefit was slowed down for number of clusters more than 5, suggesting a range of at least 5–15 clusters that could be accountable for our dataset. We then used a meta-clustering algorithm to determine the most appropriate number of clusters: n = 500 realizations of the k-means (from k = 5 to k = 15) were run. For each k and n, 50 replicates of the clustering were run and the best replicate were selected. For each k and across n, we computed the probability that different pairs of elements belonged to the same cluster. To identify reliable from spurious clusters, we used a probability threshold (p>=0.9) and considered only reliable clusters with at least five neurons to remove those composed of outliers. From the diagonal matrix of pairing cells into the same clusters using the defined criterion (p>=0.9), clustering with 8 number of classes reached the highest number of cells grouped together (100%, *Figure 4—figure supplement 1B*). The final clustering was then visualized with a dendrogram based on squared Euclidean distances between the cluster centroids. We validated finally determined number of clusters using Akaike's and Bayesian criteria which showed the smallest value for k = 8 (AIC: [−17712,−17735, −18476,−11114] and BIC: [−1.7437,−1.7368, −1.8109,−1.0747], for k = [6,7, 8,9]).

## Validation of the identified cell classes

We used dataset randomization (n = 200 realizations) as in *Ardid et al., 2015*, to validate our meta-clustering analysis by computing two validity measures. First, In each realization, each of eight clusters were associated to the closest cell class in *Figure 4A,B*. From all realizations and for each cell class, the difference between the mean of all clusters that were associated to the same cell class with respect to the mean of all clusters that were not associated to that cell class is computed versus when the clusters were randomly assigned to the cell classes (*Figure 2—figure supplement 4C*). Second, we validated the reliability of cell class assignment using n = 200 realizations of a randomization procedure that calculated the proportion of consistently assigned cells to a class compared to other cells assigned to that class. The proportion of class-matching cells with respect to control was systematically higher than class-matching when using a bootstrap procedure with random assignment of class labels (*Figure 2—figure supplement 4D*). We further validated the meta-clustering results for each monkey separately. We validated the results, analogous to what is describe above. First, validation according to the distances of clusters for each monkey (*Figure 4—figure supplement 2E*). Second, validation according to the percent number of cells matches for each monkey (*Figure 4—figure supplement 2F*).

## Correlation of local variation with burst index

The Local Variation (LV) measured how regular neighboring spike trains are, leading to higher values when neurons fire short interspike interval (ISIs) spikes (bursts) intermittent with pauses. We quantified how the LV correlated with the likelihood of neurons to show burst spikes. We calculated the

burst proportion as: number of ISIs < 5 ms divided by number of ISIs < 100 ms similar to *Constantinidis et al., 2002*. To control for effect of firing rate on the measure, we normalized it by the firing rate that would have been expected for a Poisson distribution of ISIs.

We used burst-index computed for neurons and grouped neurons in PFC and ACC into two subgroups, high burst proportion and low burst proportion (Log(BI)>0 and Log(BI)<0 respectively). We computed the proportion of neurons in each group that showed significant correlation with RPE (in ACC) and Choice Probability (in PFC). In PFC, 25% of high BI neurons and 27.5% of low BI neurons were significantly correlated with Choice probability. In ACC, however, 47% of high BI neurons and 35.2% of low BI neurons were significantly correlated with RPE. Chi-square test failed to show significant differences between two groups (low vs high BI) for proportion of significantly correlated cells with RPE (in ACC, p=0.15), and with Choice Probability (in PFC, p=0.75). The correlation of LV and BI is for all neurons is shown in *Figure 1—figure supplement 2F*.

## Spike-LFP synchronization analysis

Adaptive Spike Removal method was used on wide-band signal to remove artifactual spike current leakage to LFP (details in *Banaie Boroujeni et al., 2020a*). We then used the fieldtrip toolbox on the spike removed data to compute the Fourier analysis of the local field potential (LFP). Spike removed signals were resampled with 1000 Hz sampling rate. For each frequency number, Fourier transform was performed on five complete frequency cycles using an adaptive window around each spike (two and a half cycles before and after the spike). We then computed the pairwise phase consistency (PPC) to measure spike-LFP synchronization.

To determine at which frequency-band single neurons showed reliable spike-LFP PPC, a permutation test was adapted and used to construct a permutation distribution of spike-LFP PPC under the null hypothesis of no significant statistical dependencies of spike-LFP phase locking were preserved between spike phases and across frequencies. Then, each bands of significant frequencies were identified and for each band the sum of PPC value (which is unbiased by number of spikes) was computed. We then determined the significance based on PPC band-mass.

To determine whether the spectrum of spike-LFP synchronization measure (PPC) contains peaks that are statistically significant, we used four criteria similar to *Ardid et al., 2015*. These criteria ensure to indicate reliable frequencies that show phase-consistent spiking. First, detected peaks had to be Rayleigh test significant (p<0.05), to reject the homogeneity hypothesis of the phase distribution. Second, each peak had to have PPC value greater than 0.005. Third, each peak had to have peak prominence of at least 0.0025 from its neighboring minima to disregard locally noisy and possibly spurious PPC peaks. Fourth, detected peaks had to have PPC value greater than 25% of PPC range.

## Statistical analysis on the class-specific PPC peak distribution

To determine whether clusters show significant proportion of PPC peaks in a specific frequency band, 1000 samples with the same size to each class was selected from the population of neurons. For each sample, we computed the mean to construct a distribution of sample means under the null hypothesis that no class show proportion of PPC peak in frequency bands different than the population of samples. The distribution of peak proportion for each class was then compared with identified 95% confidence interval of the population of samples. This procedure was done separately for classes of neurons in PFC and ACC (*Figure 6* and *Figure 6—figure supplement 1*).

## Analysis of the firing onset-responses to the color onsets and error/reward outcome onsets

For each neuron, the spike density was computed using a gaussian window of 600 ms (std 50 ms) around the Cue onsets, Error outcome onsets and Reward onsets across trials. We then performed the z-score transformation of event onset aligned mean response of each cell over trials, by subtracting the pre-onset mean of spike density divided by its standard deviation (a time window of [−500 ms 0 ms] prior to the event onsets). To investigate class-specific event response, we used a permutation approach and randomly selected 1000 samples with a class size same as each class. We then constructed a distribution of mean samples under the null hypothesis that no class show event response different than sample population. Cell classes that showed significantly different response

than the population were then identified in a duration that they show response more extreme than two standard deviation from the population of samples. We performed these tests separately for classes in area PFC and ACC and event onsets: Color-Cue, Motion-Cue, Error outcome, and reward outcome.

For Broad vs Narrow spiking cell comparison of event onset response, we randomly shuffled the label of neurons and constructed a distribution of 1000 times randomly sampled difference of mean of Narrow and Broad spiking cells. We then computed 95% CI of the population samples and computed the most extreme 5% of time courses from the 95% CI under the null hypothesis that Broad and Narrow population of neurons do not show significant mean difference responses to the event onsets.

## Analysis of effect size of the firing onset-responses to the cue onsets and error/reward outcome event onsets

For effect size analysis of cell class-specific response to each of the onsets, we computed the mean difference of each cell class from each of 1000 randomly labeled samples divided by their pooled standard deviation to compute Cohen's d for each randomly selected sample. At the end, we averaged over the 1000 unsigned Cohen's d computed for each cell class. The procedure was done separately for ACC and LPFC classes and for Cue onsets and Error/Reward outcome event onsets. (*Supplementary file 1*).

## Analysis of time-resolved spike-LFP coherence under different behavioral conditions

To analyze the spike-LFP phase synchronization of neurons for the trials with 50% lowest and the 50% highest reward prediction error (RPE) for ACC neurons, and for the trials with 50% lowest and 50% highest choice probability (p(choice)) for LPFC neurons we computed time-resolved spike-LFP pairwise phase consistency. First, we divided trials into two groups of high and low RPE and p (choice) values (trials were assigned based on their median value for each experimental session). Then, for each neuron, RPE, and p(choice)condition we extracted spikes and their phase synchronization to the LFP in different frequencies (4–80 Hz, 1 Hz resolution) by applying Fourier transform on a hanning-tapered LFP signal (±2.5 frequency cycles around each spike). Then we computed the PPC for moving windows of ±350 ms every 50 ms around the outcome onset (for RPE) and around color onset (for p(choice)). We included only neurons with at least 50 spikes across trials, using on average 44 (SE 2) trials. To control for spike number, we repeated the procedure 500 times with a random subsample of 50 spikes of a neuron for each window before computing the PPC. For each neuron, behavioral condition, and window we calculated the average PPC over the random subsamples.

## Statistical analysis of time-resolved spike-LFP coherence for putative interneurons and broad spiking neurons

Statistics on the time-resolved coherence was computed in two steps. In the first step, we tested for each post-event time window the null hypothesis that *N3-type* neurons and broad spiking neurons showed similar spike-LFP synchronization strength after the event onset compared to the time windows prior to the event. To test this, we first normalized the time resolved coherence for each neuron to the baseline coherence (−850 ms to 0 ms) before reward or attention-cue onset (in ACC and PFC, respectively). We then randomly selected 1000 sample of neurons from the population with the same size as neurons in class N3 and broad cells under the null hypothesis that N3 class and broad spiking neurons do not show different synchronization pattern triggered by event onset compared with population. For each sample, we extracted the 95% CIs, and over the population of samples we extracted the most extreme 5% of the previously extracted CIs and set the final 95% multiple comparison corrected confidence intervals. We then found the average of normalized PPC values for N3 class and broad spiking neurons in a time period and frequency domain that were more extreme than the defined confidence intervals. The area of significance then was shown by black contours. In the second step, we asked whether N3 class neurons show different average synchrony strength over a time window of (0 ms 500 ms) aligned windows to the attention-cue onset (in PFC and for high and low Choice Probability conditions) and to the reward onset (in ACC and for high and low Reward Prediction Error conditions). We randomly selected 1000 sampgles, with the same size as

N3 class, from broad spiking neurons and computed their average pre-onset normalized synchrony in the defined post-onset period. We then constructed the most 5% extreme values of 95% confidence intervals defined over 1000 samples and across frequencies under the null hypothesis that N3 class cells do not show different synchrony strength from broad spiking cells in the post-onset time period and across different frequencies. We set the confidence interval levels and selected frequency bands more extreme than the CIs as significantly different (multiple comparison adjusted alpha level = 0.05, *Figure 7* and *8*).

### Analysis of spike-LFP synchronization controlled for event evoked LFP

This analysis controls that the synchronization results are not confounded by event evoked LFP signals. First, we extracted the LFP aligned to the color cue and the reward onset on each individual trial and averaged it in a −0.5 to 1 s window around the onset of the color cue and reward onset, respectively. We then removed the average event evoked LFP from individual trials. We then repeated the above-described synchronization and statistical analysis on the event evoked LFP subtracted trials. Subtraction of event-evoked LFPs did not change the results (*Figure 7—figure supplement 4*).

### Statistical analysis of functional spike-LFP gamma synchronization for neuron types

We analyzed how distribution of PPC values for each e-type is different from the other e-type in high and low RPE/p(choice) conditions. For each area, we extracted average PPC value for each neuron and conditions in frequency range 35–45 Hz. We used Kruskal Wallis test to see whether neuron types show different synchronization patterns. Lastly, we performed multiple comparison (Tukey-Kramer corrected) to see whether any of the classes is different from the others. These analyses were done separately for each area and each behavioral condition. No significant differences were observed between more certain conditions (high p(choice) and low RPE). Consistent with time resolved results, only N3 class showed stronger gamma synchrony in low p(choice) condition in LPFC, and high RPE condition in ACC (*Figure 7—figure supplement 1*; *Figure 7—figure supplement 3*).

### Analysis of narrow vs. broad and cell class-specific firing correlations with reversal learning

To investigate whether firing rate of cells correlate with the learning state, we performed correlation analysis between firing rate of single neurons and model parameters: probability of chosen stimulus (choice probability, $p_{(choice)}$), and positive Reward Prediction Error ($RPE_{pos}$). For the correlation analysis, we excluded neurons that had less than 30 trials of neural activity. For each neuron, the event onset response was normalized to the mean of all trials' pre-onset firing (in a period of −0.5 s to the event onset) and was divided by the standard deviation of all in that period. We then computed for each neuron the Spearman correlation coefficient between $p_{(choice)}$ values and then normalized firing rate in a moving window ±200 ms with sliding increments of 25 ms relative to the Color-Cue onset. We used the same procedure for the reward-onset mean of normalized firing rate and $RPE_{pos}$ values. To test whether narrow and broad spiking neurons correlate their firing rate differently to model values, we randomly shuffled cell labels and constructed a distribution of 1000 differences of the mean correlations of randomly assigned neurons to the broad and narrow groups under the null hypothesis that there is no difference in correlations depending on the spike waveform group. We then computed the most extreme 5% of the sample difference of means through their time course and identified the 95% confidence interval to test our null hypothesis. We also tested whether cells of different cell classes showed different correlations of firing rate and p(choice) or $RPE_{pos}$. using the Kruskal-Wallis test considering cell class as the grouping variable. To test which class shows correlations different than the population mean, we randomly shuffled cell class labels 1000 times and computed the mean difference between each randomly labeled cell class and the population. We then constructed a distribution of mean difference samples under the null hypothesis that no class shows a mean correlation different from the population mean. We then computed the top 5% of samples and identified 95% confidence interval. Classes that showed a mean difference of correlation to the population more extreme than the identified CI were marked as significant. All mentioned

procedures were performed separately for neurons in area ACC and PFC and for both, $p_{(choice)}$ or $RPE_{pos}$ values. In addition to the correlations of firing rate and p(choice), and firing rate and $RPE_{pos}$, we also calculated the time resolved correlation of neurons firing rate with number of trials since reversal. We found that B-type and N-type neurons in LPFC and in ACC did not change their firing differently as a function of the raw trial count since reversal. The lack of correlation with trial number was true for the color cue period and the reward period of the task (data not shown).

## Training classifiers for predicting cell classes from their correlations with learning variables

We used a machine learning approach to test how accurately cells can be labeled to a cell class based on their functional properties. For training classifiers, we used correlation of cells firing rate and RPE/p(choice) separately for areas LPFC and ACC. We test whether functional correlation of cells activity in a class allows to reliably classify them into the true class label (from the k-means clustering) or in alternate classes. We used multiclass Support Vector Machine (SVM) with one to one comparison of identified cell-classes with 10 folds of cross validation. A vector of correlation values (each element representing one neuron) was used along with a vector of cluster labels (from our clustering results) to train the SVM. The classifier used a Gaussian radial basis function kernel with a scaling factor of 1. For each classifier, only classes were considered that contained $\geq 5$ cells and each unique cluster was present in all folds. As classes N1 and N2 did not meet the criteria, we excluded them from the classifier and instead randomly distributed them to other classes (weighted by the size of classes) as an internal noise factor. For each learning measure (RPE and p(choice)) and for each area (LPFC and ACC), we subsampled each cluster with a size equal to the half of the minimum size of clusters ensuring an equal cell number from clusters in each subsample. We constructed the confusion matrix as the ratio of outcome matrix to the total count across all 1000 subsamples test and performed a binomial test (FDR-corrected $p < 0.05$) to find cells of the confusion matrix that are significantly greater than the chance level (chance level here was defined by one divide by the number of classes). Prediction of classifiers on correlation of LPFC rate and RPE, and ACC rate and p(choice) were closed to the chance level (not shown). However, in ACC N3 class was predictable with an accuracy of 0.34 from its correlation with RPE, and in LPFC, N3 class was predictable with an accuracy of 0.31 from its correlation with p(choice) (*Figure 5—figure supplement 3*).

## Analysis of the information coding cells for the rule identity and target location

To determine what proportion of neurons relative to the Color-Cue onsets as well as Reward/Error outcome onsets systematically carry information about the rule identity (Red vs. Green), or target location (Left vs. Right), we considered neurons we had at least 20 trials for each condition. We used a moving window of $\pm 200$ ms with sliding increments of 25 ms relative to the Cue-onset or Error/Reward outcome onsets. For each window, we performed the nonparametric rank sum test between the two of conditions under the null hypothesis that neurons do not fire preferentially different to a specific color or location (*Figure 2—figure supplement 2*). For Narrow and Broad spiking neurons, we computed the proportion of neurons that showed statistically significant firing rate ($p < 0.05$) to each condition. We then randomly shuffled the proportion amounts of significantly different firing neurons over the time course and computed 95% CI under the null hypothesis that each group of neurons do not show proportionally different number of neurons compared to the pre-onset population of proportion values.

## Analysis of cell class firing statistics measures

For each of firing statistic measures (firing rate, local variation, and coefficient of variance), we performed nonparametric Kruskal-Wallis test with cell class as grouping variable to test for a main effect of cell class on each firing statistics. We then performed rank sum multiple comparison for pairwise comparison of cell class differences ($p < 0.05$).

## Analysis of PPC strength for learning correlated cells vs non-correlated cells

We grouped our neurons based on their waveform (Narrow vs Broad) and then further grouped them into subgroups of those that their firing after the onset were significantly correlated with learning values and those that were not (p(choice) x Firing Rate after Cue-onset in PFC, and RPEpos x Firing rate after Reward-onset in ACC). For each waveform-grouped neuron, we randomly shuffled their labels and computed the difference of PPC peak proportions between neurons that their firing rate were significantly correlated with learning state and those that were not significantly correlated. We constructed a distribution of 1000 randomly selected samples of difference of proportions of PPC peaks under the null hypothesis that for each waveform grouped neurons there is no significant difference in the proportion PPC peaks for neurons that their firing rate were significantly correlated with learning values and those that were not significantly correlated. We then identified the most extreme 5% of the peak proportion difference and computed 95% CI over the population of samples.

## Acknowledgements

The authors thank Mariann Oemisch and Ali Hassani for help with the study. This work was supported by the National Institute Of Mental Health of the National Institutes of Health under Award Number R01MH123687 and a grant from the Canadian Institutes of Health Research (MOP 102482). The content is solely the responsibility of the authors and does not necessarily represent the official views of the National Institutes of Health or. the Canadian Institutes of Health Research.

## Additional information

### Funding

| Funder | Grant reference number | Author |
| --- | --- | --- |
| National Institute of Mental Health | MH123687 | Thilo Womelsdorf |
| Canadian Institutes of Health Research | MOP 102482 | Thilo Womelsdorf |

The funders had no role in study design, data collection and interpretation, or the decision to submit the work for publication.

### Author contributions

Kianoush Banaie Boroujeni, Conceptualization, Resources, Data curation, Software, Formal analysis, Validation, Investigation, Visualization, Methodology, Writing - original draft, Writing - review and editing; Paul Tiesinga, Conceptualization, Software, Formal analysis, Validation, Investigation, Visualization, Methodology, Writing - original draft, Writing - review and editing; Thilo Womelsdorf, Conceptualization, Resources, Data curation, Software, Formal analysis, Supervision, Funding acquisition, Validation, Investigation, Visualization, Methodology, Writing - original draft, Project administration, Writing - review and editing

### Author ORCIDs

Kianoush Banaie Boroujeni (iD) https://orcid.org/0000-0002-2323-0648
Thilo Womelsdorf (iD) https://orcid.org/0000-0001-6921-4187

### Ethics

Animal experimentation: All animal care and experimental protocols were approved by the York University Council on Animal Care (ethics protocol 2015-15-R2) and were in accordance with the Canadian Council on Animal Care guidelines.

Decision letter and Author response
Decision letter https://doi.org/10.7554/eLife.69111.sa1
Author response https://doi.org/10.7554/eLife.69111.sa2

# Additional files

## Supplementary files

• Supplementary file 1. Cohen's d effect sizes for firing rate modulation of each of eight *e-types* during the trial epochs Feature-1, Feature-2, and Reward for lateral prefrontal cortex (PFC) and anterior cingulate cortex (ACC).

• Source data 1. Correlation Analysis.

• Source data 2. Cluster Analysis.

• Source data 3. Synchronization Analysis.

• Transparent reporting form

## Data availability

Source neural data and matlab scripts for reproducing the main figures with the data are included in the manuscript as supporting files Source Data 1, 2, and 3.

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

# Appendix 1

## 1. Overview of circuit modeling

We constructed circuit motifs to account for our experimental observation that gamma synchronization characterized cue and reward onset triggered activity when choice probabilities were low (near ~0.5) and reward prediction errors relatively high. These circuit motifs provide a proof-of-concept that the empirical observations can follow from biologically plausible motifs. These circuits motifs also provide predictions which can be tested in future studies.

One circuit motif is comprised of two populations of excitatory cells (E1 and E2) and one population of interneurons (I). This 'E-E-I' motif (*Figure 9A*, *Figure 9—figure supplement 1*) was constructed to test the gamma to beta synchronization switch that the N3 *e-type* interneuron population in LPFC showed in the empirical analysis. The second circuit motif is comprised of two populations of inhibitory neurons (I1 and I2) and only one population of excitatory neurons (E). This 'E-I-I' motif (*Figure 9B*, *Figure 9—figure supplement 2*) was constructed to test the theta to gamma switch that the N3 e-type interneuron population in ACC showed empirically.

## 2. E-E-I circuit motif realizing the switch from gamma to beta frequency synchronization

### 2.1 E-E-I Network architecture

We simulated a simple *E-E-I* model with two excitatory populations recurrently connected with one inhibitory population that is conceived of reflecting the interneurons of the N3 *e-type* (*Figure 9—figure supplement 1B*, *Figure 9A*). Each population was represented by a two variables, a firing rate r modeled after the work of Hahn and colleagues (*Hahn et al., 2020*), and a synaptic variable s modeled as in *Keeley et al., 2017*. The full description of the model is given below. Both E populations are reciprocally connected to the I population. We assume that the E cells receive input representing the aggregate values of the objects. We model the situation that the value of object one increases by increasing the drive to the E1 population, whereas concomitantly we reduce the drive to E2, such that their sum remains the same.

### 2.2 Model equations for the E-E-I circuit model

The activity of each population is represented by two vectors $r = (r_{E1}, r_{E2}, r_I)$, representing the firing rate and $s = (s_{E1}, s_{E2}, s_I)$, representing the synaptic inputs. They satisfy the following coupled differential equations

$$\tau \frac{dr}{dt} = -r + \alpha G(Ws + I) + I_{noise}$$

And

$$\tau_{syn} \frac{ds}{dt} = -s + \gamma F(r)(1 - s)$$

Where $\tau = (\tau_{E1}, \tau_{E2}, \tau_I) = (1.5385, 1.5385, 1.5385)$ is the firing rate time scale, $\tau_{syn} = (\tau_{syn,E1}, \tau_{syn,E2}, \tau_{syn,I}) = (2.3077, 2.3077, 15.3846)$ is the synaptic time scale, $\alpha = (\alpha_{E1}, \alpha_{E2}, \alpha_I) = (2.5, 2.5, 5)$ is a scaling variable to adjust the mean firing rate of each population, $\gamma = (\gamma_{E1}, \gamma_{E2}, \gamma_I) = (4, 4, 3)$ is the scale of synaptic onset rate, $I = (I_{E1}, I_{E2}, I_I)$ is the drive for each population, and W is a 3 by 3 connection strength matrix:

$$W = \begin{pmatrix} 2.0 & 0 & -2.6414 \\ 0 & 2.0 & -2.6414 \\ 3.0 & 3.0 & -0.1 \end{pmatrix}$$

We write $I_{E1} = I_0 + I_{max}x$ and $I_{E2} = I_0 + I_{max}(1 - x)$, where x varies between 0 and 1. Here $I_I = 0$, $I_0 = 0.8$, $I_{max} = 0.4$. The noise current $I_{noise}$ had a standard deviation of 0 for the simulations shown in this note. It can be used to induce transient oscillations when there is a stable fixed point with eigenvalues that have an imaginary part.

The firing rate response function is

$$G(x) = \frac{x}{1 - e^{-x}},$$

and the one for the synaptic inputs is

$$F(r) = \frac{1}{1 + \exp\left(\frac{\theta - r}{k}\right)}$$

Here $\theta = (\theta_{E1}, \theta_{E2}, \theta_{I,}) = (5, 5, 10)$ is the activation threshold for the synapse and the $k = (k_{E1}, k_{E2}, k_I) = (0.5, 0.5, 1.0)$ is the sharpness of the synaptic activation function.

## 2.3 Simulation results of E-E-I model

When the drive to E1 increases, the activity of population E1 increases whereas that of E2 decreases, with the level of I activity varying only moderately with E1 drive (*Figure 9—figure supplement 1A*). The circuit executes a soft version of the winner-take-all mechanism, the E population with the largest drive suppresses that of the one with the lower drive. We chose parameters such that the network displayed oscillations by first finding a Hopf bifurcation, using a continuation approach implemented with the software auto07 (*Doedel et al., 1991*). A Hopf bifurcation is signaled when the Jacobian at the fixed point has two complex conjugate eigenvalues of which the real part becomes positive at the bifurcation (*Strogatz, 1994*). For small amplitudes, the oscillation frequency is directly related to the imaginary part of the eigenvalues. Stable oscillations appear in the model with the frequency increasing from beta for low E1 drives to gamma when the E1 and E2 is similar (*Figure 9—figure supplement 1B*). The power of these oscillations follows more or less the mean activity of each population.

## 3.3. E-I-I circuit motif realizing the switch from gamma to theta frequency synchronization

### 3.1 E-I-I network architecture

We constructed a second model to account for the switch between theta and gamma synchronization (*Figure 9—figure supplement 2*, *Figure 9B*). This model has two types of interneurons (the I1 and I2 populations) and one E cell population (E), reciprocally connected. They form two PING-type motifs similarly to *Domhof and Tiesinga, 2021*, which focused on beta/gamma frequency switches (see *4. Discussion*). The first motif with I1 forming a fast circuit, generating gamma, the second one together with I2 forming a slow circuit for theta. Each motif can create its own oscillation, but when one circuit is dominant it takes over the other circuit and imposes its frequency. We assume that interneuron population I1 corresponds to PV neurons because they have a faster dynamics. We simulate the case of rewarded trials, which means that the RPE is low when the expected value is high, whereas when the RPE is high the expected value is low. We further assume that the value-associated drive to I1 is part of a disinhibitory circuit, that is, it is an inhibitory input to I1 that reflects the expected value. In other words, when RPE varies from low to high values, the drive to I1 varies from low to high.

### 3.2 Model equations for the E-I-I circuit model

The network is simulated using the same modeling framework as in 2.2 (*above*), but now there are two I populations, I1, I2, and only one E population, hence the vectors are changed in an obvious way: $r = (r_E, r_{I1}, r_{I2})$; and $s = (s_E, s_{I1}, s_{I2})$, $\tau = (\tau_E, \tau_{I1}, \tau_{I2}) = (1, 1, 5)$; $\tau_{syn} = (\tau_{syn,E}, \tau_{syn,I1}, \tau_{syn,I2}) = (1.5, 5, 45)$, $\alpha = (\alpha_E, \alpha_{I1}, \alpha_{I2}) = (2.5, 5, 5)$, $\gamma = (\gamma_E, \gamma_{I1}, \gamma_{I2}) = (4, 3, 3)$, and $I = (I_E, I_{I1}, I_{I2})$. Here $I_{I1} = I_{01} + I_{max1}x$ with $I_{01} = -3$ and $I_{max1} = 3$; $I_E = 0.71646$; $I_{I2} = -0.3$. The noise current $I_{noise}$ has a standard deviation of 0. W is the following 3 by 3 matrix:

$$W = \begin{pmatrix} 2.0 & -1.3207 & -1.3207 \\ 3.0 & -0.1 & 0 \\ 3.0 & 0 & -0.1 \end{pmatrix}$$

The response functions G and F are identical to those specified in model 1 (see 2.2), with for F the parameter values: $\theta = (\theta_E, \theta_{I1}, \theta_{I2}) = (5, 10, 10)$ and $k = (k_E, k_{I1}, k_{I2}) = (0.5, 1.0, 1.0)$

### 3.3 Simulation Results of E-I-I model

We again used auto07 to find Hopf bifurcations, from which we started the exploration of the network dynamics. When we increased the drive to I1 the firing rate of I1 increased (*Figure 9—figure supplement 2A*) and the oscillation frequency increased from around the theta band to gamma frequencies (*Figure 9—figure supplement 2B*).

## 4. Discussion of circuit motifs, relation to other models and experiment

The E-E-I motif provides a proof of principle for the link between diversity of input and oscillation frequency (see *Figure 9—figure supplement 1* and *Figure 9A*). We increased the drive to E1 and reduced it to E2 in such a way that the sum remained constant and studied the oscillation frequency of the I population. The situation with high drive to E1 and low drive to E2 (and vice versa) corresponds to a situation with diverse inputs which happens in a reversal block after learning of values is completed (in the 'steady state') and one object has high value and the other object a low value. In this regime oscillations are prominent in the beta frequency range (*Figure 9—figure supplement 1A*). But when the drive of the E1 and E2 populations is similar, indexing the situation of low p (choice), i.e. when it is near 0.5, the I population increased its oscillation frequency to the gamma range (*Figure 9—figure supplement 1*). Hence, in the model, competition between two similarly-valued objects that results in a low choice probability is indexed by gamma oscillations of the inhibitory cell population, while otherwise beta synchrony predominates. This result matches the core oscillatory signature we observed in the LPFC around the color cue onset. It suggests that the transient gamma increase of the N3 *e-type* might reflect the gating of diverse inputs as has been suggested by larger-scale modeling of similar circuit motifs (*Buia and Tiesinga, 2008*; *Sherfey et al., 2018*; *Sherfey et al., 2020*).

The second circuit that implemented a E-I-I model provides a proof of principle for the link between the increased activation of a 'fast' interneuron population (I1) and a switch from theta to gamma oscillations. Here, theta synchronous activity driven by the I2 neurons corresponds to low RPE trials (after learning of values is completed), in which a reward R is received and the value V of the chosen stimulus was relatively high (a high V and a large R, the RPE is computed as R - V) (see *Equation 2* in Materials and methods of main text) (*Watabe-Uchida et al., 2017*). In contrast, the gamma synchronous state that emerges with larger drive to the I1 neurons in the model correspond to high RPE trials, in which a reward R is received, but the value of the chosen stimulus was relatively low (low V). This circuit motif is plausible when one assumes that the I1 neuron population is disinhibited when the chosen stimulus value is low. Such a disinhibition can be achieved by lowering the drive to I2 cells, or by assuming a separate disinhibitory circuit involving other inhibitory cells. In the model simulation we only explored the former assumption. In summary, the E-I-I motif reproduces the switch of gamma to theta synchronization we observed during learning in ACC N3 *e-type* neurons. At the functional level, the circuit suggests that the emergence of gamma activity in this network indexes the detection of a discrepancy between the received reward (as one source of input) and the chosen stimulus value (as another source of input).

The oscillation frequency observed in these two models was not directly related to biophysical time scales, such as, synaptic or membrane time scales or rate constants for the opening and closing of ionic channels, as would be the case in models based on Hodgkin-Huxley type channels (*Tiesinga et al., 2001*), rather it was achieved by the product of the two effective time scales (firing rate and synaptic) in the model. Therefore, these models serve as a proof of principle, indicating how populations may be wired up to produce oscillations with different frequencies, but they can not make conclusive predictions regarding the dynamics of the underlying interneurons, that is, whether they are PV or SOM, or what type of spike patterns they produce. For this type of insight proper network models composed of biophysical models need to be constructed. Nevertheless, we think it is reasonable to identify faster interneuron populations with PV+ interneurons given prior modeling studies (see next paragraph), and thereby putatively link them to the N3 *e-type* (*see* also Discussion of the main text).

Similar reservations hold for the mechanism by which oscillations are generated, such as for instance ING versus PING (*Whittington et al., 2000*; *Tiesinga and Sejnowski, 2009*; *Tiesinga, 2012*). Model 1 is functionally a soft winner-take-all model, but the oscillations could emerge by way of an ING motif, potentially heterogeneously activated, when individual interneurons receive a different mix of inputs from E1 and E2. Previous simulations by us and others (*Wang and Buzsáki, 1996*; *White et al., 1998*; *Tiesinga and José, 2000*; *Tiesinga and Sejnowski, 2004*) show that this would be feasible. Model 2 is comprised of two competing E-I motifs, which our recent simulations indicate (*Domhof and Tiesinga, 2021*) could implement switches when one motif is more strongly activated than the other. Our simulations do not exclude the possibility that the I1 population synchronizes by the ING mechanism, but it would in our opinion represent a less parsimonious explanation.

The involvement of ING and PING mechanisms for beta and gamma oscillations are well-established. For theta oscillations other mechanisms have also been proposed, for instance by way of intrinsic membrane resonance (*Hutcheon and Yarom, 2000*) in the pyramidal cells (*Tiesinga et al., 2001*) activated by neuromodulatory tone or in a specific type of interneuron (*Rotstein et al., 2005*), which do need to be reciprocally connected to a fast interneuron for the theta oscillations to emerge. In other models slower synaptic time scales were instrumental (*White et al., 2000*). As resonance mechanisms were not explicitly modeled, our model simulations do not directly speak to whether the empirical findings rely on resonance properties. We can therefore not conclusively exclude them until a more comprehensive modeling study is conducted that not only takes into account synaptic time scales but also the intrinsic dynamics of all the involved neuron classes together with their task-dependent firing rate dynamics. A comprehensive review of cortical rhythms and their mechanisms can be found in *Wang, 2010*.

