## [Decision Letter]

**Acceptance summary:**

This paper will be of interest to system neuroscientists studying reinforcement learning, as well as neuroscientists in the field of brain rhythms. The work sheds new light on the specialization of individual cell types in the cortex of animals engaged in a challenging task. The authors combine many different techniques (single-cell recordings, clustering of cell types, behavioral modeling, spike-field coherence) in order to understand the differential contributions of subclasses of cell types to cortical computations during a reversal-learning task. The questions asked by the authors in this paper are interesting and their treatment is thorough, with many controls. As a result, this work is a valuable addition to the field.

**Decision letter after peer review:**

[Editors’ note: the authors submitted for reconsideration following the decision after peer review. What follows is the decision letter after the first round of review.]

Thank you for submitting your work entitled "Interneuron Specific Gamma Synchrony Indexes Cue Uncertainty and Prediction Errors in Prefrontal and Cingulate Cortex" for consideration by *eLife*. Your article has been reviewed by 3 peer reviewers, and the evaluation has been overseen by a Reviewing Editor and a Senior Editor. The reviewers have opted to remain anonymous.

We are sorry to say that, after consultation with the reviewers, we have decided that your work will not be considered further for publication by *eLife*. While we found the work overall potentially interesting, and the reviewers each found merit in particular elements, several major concerns were raised regarding key components (including the computational model and the focus on the N3 subtype). You will find these detailed in the reviews attached below.

In its current state, the paper tries many things at once (neuron subtype clustering, reward processing, computational modeling), which by itself is laudable, but as it stands it is coming short at tying these aspects together.

We are not inviting a revision because the amount of work we consider required is too extensive. However, given there was considerable interest in elements of the work, we want to point out you are free to decide to rework the current manuscript into a new version and submit this new manuscript to *eLife*. Note that if you choose to submit a new manuscript it would go through the regular process again (i.e., consideration by editors) and if selected to be sent out for review could potentially go back to the same reviewers and/or new ones.

*Reviewer #1:*

Boroujeni et al. recorded extracellular spikes from single neurons in brain areas LPFC and ACC in two awake behaving macaques that were performing a reward reversal learning task. They classified the recorded neurons into various subtypes, and investigated how neuronal activity in these different subtypes related to the variables of the behavioural task.

The paper's clear and primary strength is the classification of extracellularly recorded neurons into broad- and narrow-spiking neurons, and even further into subtypes of these two classes. While a split based purely on spike waveform shape into broad- and narrow-spiking is relatively common, the cluster-based classification into subtypes based on various additional parameters like spike variability is novel and potentially illuminating. The authors furthermore convincingly demonstrate that the recorded narrow-spiking neurons (often labelled "putative inhibitory interneurons") are indeed likely inhibitory in nature, by showing that the net effect of a spike in these cells on the surrounding population spike rate is negative. The analysis choices in this part of the paper were clear, well-motivated, and well-presented.

However, the bulk of the paper is taken up by the relationship between neuronal spiking and variables from the behavioural task, specifically choice probability (p(choice)) and reward prediction error (RPE). Here, the conclusions appear not backed up by the data, for several reasons.

First of all, the authors only present results for correlations with RPE in the reward window, and results for correlations with p(choice) in the stimulus windows. One of the main conclusions of the paper is that LPFC neurons code for p(choice) whereas ACC neurons code for RPE. However, correlations with RPE in the stimulus windows and p(choice) in the reward window are never shown. Furthermore, the authors demonstrate that, purely given the task structure, RPE and p(choice) are almost perfectly negatively correlated (r = -0.928, Figure S4). It is therefore very possible that the crucial split is not between p(choice) and RPE as the determinant of neural activity, but simply the time window in which these are analyzed.

Second, the authors present a "circuit architecture" that might account for the observed results. In the Results, this model is presented as though it were a computationally implemented biophysical neural circuit model that makes predictions that are in line with the observed data. I cannot find details of the implementation of such a model in the Methods, which makes the status of the predictions here unclear. It is not explained why two equally-valued objects would lead to gamma synchronization, whereas two objects of unequal value lead to beta synchronization (the key conclusion derived from the model). This appears to depend on total input strength, but it is hard to see why 0.5 + 0.5 (equal value, numbers provided by authors) would result in higher input than 0.8 + 0.2 (unequal value, again numbers from this paper, Figure 9). These choices, and others, appear arbitrary. In general, the description of the model in Results reads more like an interpretation/Discussion section than an outline of model-derived Results.

Third, the presented empirical evidence for narrow-spiking cells (or, more specifically, the N3-subtype) engaging preferentially in gamma-band synchronization, whereas broad-spiking cells engage preferentially in beta-band synchronization, is modest. Interneuron engagement in gamma rhythms is expected from the literature, of course, but in the present dataset this is less clear-cut. In particular, the spectral peaks in Figure 6C are quite similar between broad- and narrow-spiking, and labelling the former "beta" but the latter "gamma" requires a more thorough analysis than is now presented.

Fourth, there are some issues with reporting, where occasionally results are only reported for the narrow-spiking cells and not for the broad-spiking cells, or it is unclear whether a stated result holds for all or just a subset of cells, etc.

Finally, all results are shown aggregated over two animals, while it is important to know how the key results hold in the two animals separately.

I mention some additional recommendations here.

At the very least, correlation analyses for both p(choice) and RPE should be shown for all time windows, to allow a proper assessment. If the authors indeed wish to maintain the hard claim of a dissociation ACC<>RPE and LPFC<>p(choice) this should explicitly be tested by e.g. directly comparing the correlations with the two behavioural variables.

The model should be specified in much more detail. Specifically, the assumptions built into it should be clearly defined, and the quantitative predictions derived from it should be presented.

I understand that the data are not yet publicly released, as others from the same lab are still working on the same data (which is common in the field). However, I would urge the authors to make the source code for all reported analyses publicly available already, to greatly improve transparency and replicability. ("Upon reasonable request" is not sufficient for this goal.)

In general, the narrative could be streamlined a bit, as it currently stands the manuscript is hard to read.

*Reviewer #2:*

This paper studies the role of lateral prefrontal cortex (LPFC) and anterior cingulate cortex (ACC) in reversal learning. The authors suggest that LPFC plays a role in computing the probability that the animal will make a certain choice (termed choice probability), whereas ACC signals the reward prediction error. Interestingly, narrow spiking cells (putatively inhibitory neurons also known as fast spiking units) had a higher correlation with these task-relevant parameters, compared to broad spiking cells (putatively excitatory neurons also known as regular spiking units).

Next, the authors define electrophysiological cell types (termed e-types), based on spike waveform and firing patterns. The narrow spiking cells are subdivided into 3 subclasses, termed N1, N2 and N3. Notably, the same subclass of narrow spiking cells, N3, had a correlation with choice probability in LPFC and a correlation with reward prediction error in ACC. Neither of the other narrow spiking subtypes had a significant correlation with either parameter in either area.

In the final part of the paper, the authors examine the phase-locking behavior of these N3 cells to the local field potential (LFP). They find that in LPFC, N3 cells phase lock to gamma (35 – 45 Hz) during the initial learning stage shortly after rule reversal, but as learning progresses and performance reaches a new plateau, their phase locking switches to the beta-band (15 – 30 Hz). Perhaps most remarkably, the N3 cells in ACC showed a similar reversal learning stage dependent phase locking behavior; to elaborate, they phase-locked to gamma only when the reward prediction error was high (i.e., shortly after rule reversal).

These results are generally well supported by rigorous statistics and sophisticated analyses. However, there are several weaknesses. First, while the claim that LPFC encoded choice probability is well supported, the claim that ACC encodes reward prediction error is not as well substantiated. As seen in Figure 3, percent neurons showing significantly correlation between their firing rate and reward prediction error is not very different between LPFC and ACC, and quite similar between broad spiking and narrow spiking units within ACC.

Second, the authors build a reinforcement learning model to calculate "Choice Probability", which quantifies the probability that the animal will select the rewarded stimulus. According to this definition, choice probability should dip upon reversal, and rise to a new plateau after several trials. However, this metric is fairly unintuitive, not to mention in conflict with existing nomenclature (e.g., Nienborg, Cohen and Cumming 2012). It would be helpful to have an accompanying plot of how the firing rate and phase locking behavior of each neuronal type changes as a function of trials after reversal.

Third, the extent to which choice probability encoding neurons and reward prediction error encoding neurons in each area falls into a specific e-type is not shown.

Undoubtedly, it is noteworthy and remarkable that N3 is the only e-type that shows a positive correlation with choice probability in lateral prefrontal cortex and a positive correlation with reward prediction error in ACC (Figure 5). But do all choice probability encoding neurons in LPFC and reward prediction error encoding neurons in ACC fall into the N3 e-type?

Further, the task-dependent phase locking behavior of e-types other than N3 are not shown. Given that N3 is the only NS e-type that shows a relationship with task-relevant parameters, I would expect the task learning dependent phase-locking behavior to also be unique to N3, but this result is not presented in this paper.

Finally, the conceptual model in Figure 9 captures the results presented in this paper and gives rise to testable predictions. It seems that some predictions of this model should be testable with the presented data. For example, the prediction that in LPFC, broad spiking cells fall into two functional categories, whereas N3 cells are more functionally homogeneous, would be an interesting prediction to test. Further, the prediction that in ACC, broad spiking cells encode reward whereas N3 cells encode reward prediction error is easily testable and would strengthen the conclusions of this paper.

The main finding of this paper, that a specific electrophysiological subclass of narrow spiking cells serve important roles in a reversal learning by preferentially phase-locking to gamma band LFP, would be of broader significance and impact if this finding could be generalized to other brain regions, behavioral tasks and model species. That said, there are already several papers in the literature that define e-types. Specifically, Markram et al. (2015) define 11 e-types; Gouwens et al. 2019 define 6 e-types that constitute narrow spiking cells (referred to as fast spiking cells in Gouwens et al). For sake of future efforts to study e-types and their functional roles, it would be important to reconcile these disparate definitions of e-types.

Moreover, there are at least two other papers showing that subclasses of narrow spiking neurons have different relationship with gamma (Shin and Moore 2019; Onorato et al., 2020). It would be very interesting and important to know whether the 3 narrow spiking e-types discussed in this paper match up with the subclasses in the two aforementioned papers.

In sum, this paper is a valuable addition to the reinforcement learning literature as well as neuronal cell types and neural oscillations literature. Some additional analyses could strengthen the conclusions of this paper. It is unclear how the e-types defined in this paper will tie into other neuronal categorizations in recent literature. This link to prior work will be important for broader significance.

Comments for the authors:

I. Comments on Figures

1. Figure 2 and Figure S6 shows the PSTH aligned to Feature 1 and Feature 2 based on the cue order (Motion first vs Color first). It would be highly relevant to also show the PSTH aligned to Feature 1, Feature 2 and Reward based on behavioral outcome (correct vs incorrect, and there are at least 3 different types of error outcomes; please see my comment III-2 in Comments on Methods below for elaboration).

In particular, PSTH aligned to reward conditioned on behavioral outcome is crucial for interpreting Figure 3.

2. Figures 2 and 3: The correlation between firing rate and Choice Probability / RPE is interesting, but not very intuitive. It would be helpful to have a plot of Choice Probability and Reward Prediction Error as a function of trials since reversal, as well as the firing rate for each cell type and brain area as a function of trials since reversal. This way we can see whether LPFC NS firing rate after color cue onset tracks Choice Probability, and whether ACC NS firing rate after reward tracks RPE.

3. Figure 4B firing rate unit is missing both the figure and in the main text.

Figure 4C rastergram firing rate seems massively different from the average firing rate in 4B? e.g., for Figure 4C rastergram for N1, there seems to be ~5 spikes per 100ms, which would be ~50Hz, but the average firing rate for N1 is 4Hz?

Also, please discuss why the narrow spiking firing rate is so low (assuming the firing rate unit was Hz, mean firing rate is <2Hz for N2 and N3). Narrow spiking firing rates have typically been reported to be ~10Hz in vivo.

4. Figure 5: It is remarkable that N3 is the only e-type that shows a positive correlation with choice probability in LPFC and a positive correlation with reward prediction error in ACC. To what extent do choice probability encoding neurons and reward prediction error encoding neurons in each area fall into a specific e-type? I would like to know whether a neuron's e-type is predictable from task-dependent functional properties of the neuron.

5. Figure 6C: suggest plotting N3 in the same plot as Broad Spiking and Narrow Spiking units such that the magnitude can be compared more easily.

In addition, please clarify what the y-axis of Figure6c means (Peak densities of spike-LFP synchronization (PPC)). Is this simply the average PPC spectra? Or normalized for each unit in some way? I would recommend plotting the former, such that it is possible to compare which e-types have the best locking properties to which frequency band.

6. Figure 7 and 8: It's very interesting that initially after reversal, N3 locks to gamma but later, as performance reaches a new plateau, N3 locks to beta. If you plot trial since reversal on the x-axis, and plot the peak of PPC spectra (averaged across N3 cells) on the y-axis, do you see a gradual change in peak frequency or is it more of a step function change after each reversal? Relatedly, if you plot the histogram of PPC spectra peak frequency across N3 cells, is it a bimodal distribution (one peak in beta and another peak in gamma) or is it unimodal?

7. It would be interesting to know the behavior-dependent phase locking of other e-types as well. I suggest adding Figure 7 and 8 C and F for all e-types as a supplemental figure.

8. Were LPFC and ACC recorded simultaneously? If so, it would be very interesting to see if inter-area coherence mimics the changes in PPC. For example, does the gamma band coherence go up in the first few trials after reversal, followed by an increase in beta band coherence as behavioral performance plateaus?

9. Figure 9 outlays a really nice hypothesis that gives rise to testable predictions. Some of these predictions are testable within the data presented in this paper. I think it would significantly strengthen this paper if some of these predictions could be tested:

Figure 9 hypothesizes that in LPFC, Broad Spiking neurons should encode Value predictions; e.g., red-selective neurons that, after learning, fire more when red is being rewarded compared to when green is being rewarded. These Value-predictive neurons should fire similarly during learning, and is perhaps even predictive of the animal's choice on a trial-by-trial basis (e.g., on trials that red-selective neurons fired more during learning, the animal saccades according to the red stimulus). In contrast, N3 neurons should show no such Value-predictive behavior. Is there evidence of such prediction in the data?

Relatedly, Figure 9 hypothesizes that in ACC, Broad Spiking neurons encode reward, whereas N3 encode RPE. According to this prediction, N3 activity should be higher for "surprise correct" trials shortly after reversal, and go down as performance plateaus, whereas Broad Spiking neurons should be excited by reward the same amount regardless of whether it is shortly after reversal or after behavioral performance has reached plateau. Is this seen in data? I think this would be made clear if the PSTH aligned to reward were plotted, as suggested in Comment 1.

II. Comments on Main Text

1. "We next asked whether the narrow spiking, putative interneurons that encode p(choice) in LPFC and RPE in ACC are from the same electrophysiological cell type, or e-type (Markram et al., 2015)."

There are ~11 e-types described in Markram et al., 2015. Further, Gouwens…Koch 2019 NN describes ~6 sub-e-types of Fast Spiking cells. I recommend the authors to speculate on how previously reported e-types match up with the e-types described in this paper.

2. "Prior studies have suggested that interneurons have unique relationships to oscillatory activity (Cardin et al., 2009; Vinck et al., 2013; Voloh and Womelsdorf, 2018; Womelsdorf et al., 2014a),"

I suggest adding Chen…Zhang 2017 Neuron to this list of references.

3. Discussion section: There are at least two other papers showing that subclasses of narrow spiking neurons have different relationship with gamma (Shin and Moore 2019 Neuron; Onorato…Vinck 2020 Neuron). It would be an interesting addition to the Discussion section to speculate on whether the 3 narrow spiking e-types discussed in this paper match up with the subclasses in the two aforementioned papers.

III. Comments on Methods

1. In general, the Method section is not consistent about referring to relevant figures for the analyses being described. It would really help the reader if the analyses that went into each figure were clarified: e.g., "Statistical Analysis of time resolved spike-LFP coherence for putative interneurons and broad spiking neurons (Figure 7, 8)"

2. Task design: "Color-reward associations were reversed without cue after 30 trials or until a learning criterion was reached, which makes this task a color-based reversal learning task. "

It seems that a strategy that a monkey might employ would be to count the number of trials after reversal to anticipate when the next reversal would happen, which would rely on a different mental strategy than reversal learning tasks where the reversal points are not predictable. Is there any behavioral evidence that would discount the possibility that the monkeys are counting?

"Hence, a correct response to a given stimulus must match the motion direction of that stimulus as well as the timing of the dimming of that stimulus."

In this task, there appears to be one way to be correct, but several distinct ways of being incorrect. First, the monkey could be incorrect in both the timing and the saccade direction. Second, the monkey could be correct with the timing but incorrect with the direction. Third, the monkey could be correct with the direction but incorrect with the timing. The third outcome could be further subdivided into premature response versus late response. The reason why a monkey might make each mistake is different. Only the first scenario supports the possibility that the monkey thought the other color was being rewarded, e.g., shortly after reversal. It would be interesting to know the proportion of each error type as a function of trials since reversal. Furthermore, I would expect the negative reward prediction error to be most prominent in the first type of error. Hence, it would make sense to me if only the first type of error was considered when calculating choice probability and reward prediction error.

3. "Here, we use this model to estimate the trial-by-trial fluctuations of the expected value (EV) for the rewarded color and the choice probability (CP) of the animal's stimulus selection. EV and CP increase with learning similar to the increase in the probability of the animal to make rewarded choices, causing all three variables to correlate (Figure 4E, F)."

Figure 4 does not have E-F panels.

4. Behavioral analysis: I could not find a formal definition of Choice Probability and Reward Prediction Error anywhere. I assume Equation 4 defines Choice Probability, while Rt-Vt defines RPE? I suggest making these definitions clear in the Methods, as well as the main text and the figure legend.

Choice Probability is abbreviated in at least three different ways throughout the manuscript (e.g., p(choice), CP, CHP). Please be consistent.

Note on terminology: Choice Probability commonly refers to the relationship between the activity of individual sensory neurons and the animal's behavioral choice (see Nienborg, Cohen and Cumming 2012 ARN). The duplicate terminology may be confusing for some readers. I suggest using a different term (e.g., Probability of Choice).

5. "We then quantified the log-likelihood of the independent test dataset given the training datasets optimal parameter values."

Where is this result plotted? What is the model performance in predicting test dataset?

6. Waveform analysis: It would help to add a diagram of T2P, T4R and HR in Figure 4.

Relatedly, trough comes before the peak in extracellular spike waveforms (as apparent in Figure 4C) – T2P should be (t_peak_-t_trough_) in order to be a positive value?

7. "LV is a measure of regularity/burstiness of spike train and is proportional to the square of the difference divided by sum of two consecutive interspike intervals (Shinomoto et al., 2009)."

This sentence should go in the main text. The reason being; the way LV is described in the main text makes it sound like LV and CV measure the same things: "regular or variable interspike intervals (local variability 'LV'), or more or less variable firing relative to their mean interspike interval (coefficient of variation 'CV')."

8. Given how central the clustering analysis in Figure 4A is to the rest of the paper, the exact parameters that went into this analysis (HR, T4R, LV, CV, FR) should be made clear in the main text.

In addition, this clustering analysis is key to the reproducibility of e-types in other datasets. The authors have stated that "All data and code is available upon reasonable request." However, in my opinion, at least the code for the e-type clustering analysis should be made publicly available.

9. "Correlation of local variation with burst index"

Burst index is defined here, but not plotted in any figures. I suggest adding a plot depicting the relationship between local variation and burst index would be informative.

10. "First, we divided trials into two groups of high and low RPE and CHP values (trials were assigned based on their median value for each neuron)."

I understood RPE and Choice Probability to be values unique to each trial, rather than to each neuron? If so, the median value should be specific to each behavior session, not to each neuron? Please clarify.

11. "We included only neurons with at least 50 spikes per time window."

Does this sentence mean 50 spikes per time window per trial? For a 700ms time window, this would mean that the neuron would have to be firing at ~70Hz in order to be included in this analysis! If this sentence means 50 spike per time window across trials, please clarify. In this case, please also clarify the range of trial number that went into this analysis.

*Reviewer #3:*

In this work, Boroujeni et al. investigated the role of different cellular subtypes in the lateral prefrontal cortex (LFPC) and anterior cingulate cortex (ACC) of the rhesus macaque as the animals performed an attention demanding reversal-learning task. The authors use an attention-augmented reinforcement learning model to track the trial-by-trial values of key decision-making variables which were then correlated against the neural activity. The cellular population was separated into broad and narrow spiking neurons using features computed from the extracellularly recorded waveforms. The authors find that the activity of the narrow spiking cells in the LFPC is correlated with the choice probability, whereas the activity of narrow spiking cells in the ACC is correlated with reward prediction errors. Interestingly, the authors find that further splitting the population of broad and narrow spiking cells into subtypes revealed that both the choice probability in LPFC and the reward prediction error in the ACC were encoded by a specific subtype of putative interneuron. The authors show that the spike-field phase synchronization of this putative interneuron subgroup is also modulated by choice probability in the LFPC and reward prediction error in the ACC, mirroring the result from their single-unit correlation analysis. The authors use these results to propose a biologically plausible circuit model of how learning in such a task might be implemented through interneuron specific synchronization.

While many of the results in the paper seem robust, some of the conclusions drawn by the authors rest on analyses and methods that require further validation and controls.

1. The clustering of the cell population into 5 broad-spiking and 3 narrow-spiking subtypes is perhaps one of the most critical results that requires further validation since a lot the conclusions in the paper rely on the outcome of this analysis. The validation that the authors include in the paper (Figure S5C, S5D) address concerns regarding the clustering quality, but it's still unclear how meaningful this separation into these 8 clusters actually is. The clustering is also performed on the pooled data across both animals, but the authors should have also shown what the clustering looks like when performed independently on the population from each animal, and if there is a meaningful correspondence between the sets of clusters recovered in the two populations.

2. Most of the follow-up analysis focuses on the comparison between one specific interneuron subtype (N3) and all broad -spiking cells. I imagine that the reason for this is two-fold: (1) the N3 subtype is the only one that showed a significant modulatory effect on the multi-unit activity (Figure 4D), and (2) it seems to be special in the sense that the activity of the N3 cells is significantly correlated with choice probability in LPFC in addition to reward prediction error in ACC. While the reasons for showing key results only for the N3-type can be appreciated, the authors should have included additional control analysis to demonstrate that their results are indeed specific to the N3 subtype. For example, in Figure 7 and 8, the authors show a comparison of the spike-LFP phase synchronization between N3 and broad spiking cells, but no further characterization of subtypes within the broad spike cells or the other narrow spiking types (i.e. N1, N2).

3. The authors show that the spike-field phase synchronization of the N3 subgroup is also modulated by choice probability in the LFPC (Figure 7) and reward prediction error in the ACC (Figure 8), mirroring the result from their single-unit correlation analysis (Figures 2 and 3). Unlike their firing rate analysis however, they do not show anatomical specialization in these analyses, even though the model they propose in Figure 9 clearly shows that they hypothesize this to be the case. It would be very interesting to show the analysis performed in Figure 7 for the ACC N3 population, and likewise, the analysis performed in Figure 8 for the LPFC N3 population.

4. Behavior

a. In Figure 1C, I imagine that the proportion of rewarded choices at reversal (t=0, not shown) is equal to one minus the asymptotic performance? So around 0.1?

b. If the stimulus-reward pairings are fully deterministic, why does the monkey require so many trials (on average 7 I believe it was) to reach asymptotic performance again?

c. Related to the previous question, is there any change in this acquisition time over the course of a session (as they experience more and more reversals)?

d. Can you show some example fits of the reinforcement learning model? For example, the choice probability and expected value as a function of the trial number around a reversal.

5. Single Units

a. The authors correlate the neural activity with model-derived variables, like the probability of choice, and prediction error. The distributions of these variables, however (as indicated in Figure S4b, and S4C) are very skewed, and it seems like most of the variability comes from the few trials (around 10) that it takes to reach asymptotic performance after a reversal. It would be interesting to know what this correlation represents. Are the cells truly tracking small changes in the P(choice) and PE or does this reflect more of a discrete switch? Maybe the authors could show some scatters, firing rate vs. P(choice), of some example cells. How well can p(choice) and PE be decoded from the neural population?

6. Electrophysiology/Clustering

It seems that a lot of the results in the paper rely on clustering analysis. The authors have been cautious in their approach (i.e., validating the results), but given that a lot depends on the reliability of these results, I think it would be wise to add a few more control analyses. I am not sure how feasible these are, but worth considering nonetheless:

a. Another way of validating the clustering is to do it across animals. From what I understood, the clustering (for e-type) is done using data from both animals. How well would a clustering model fit within animals, predict the clustering across animals?

7. Spike field coherence

a. Can the authors comment on the effect of ERPs?

b. Simply controlling for the number of spikes between conditions is not necessarily sufficient. If you have a cell that responds to one condition but does not respond to another condition, the spikes for condition 1 are going to be much more clustered in time than for condition 2. Therefore the underlying LFP is not sampled in the same way between the two conditions.

c. Is it possible to show that the spike-field coherence results are also anatomically specific? Does the synchrony of cells in the ACC and LPFC mirror the single-unit results, i.e. reward prediction error in ACC but not LPFC and choice probability in LPFC but not ACC?

[Editors’ note: further revisions were suggested prior to acceptance, as described below.]

Thank you for submitting your article "Interneuron Specific Gamma Synchronization Indexes Cue Uncertainty and Prediction Errors in Lateral Prefrontal and Anterior Cingulate Cortex" for consideration by *eLife*. Your article has been reviewed by 2 peer reviewers, and the evaluation has been overseen by a Reviewing Editor and Michael Frank as the Senior Editor. The reviewers have opted to remain anonymous.

Essential revisions:

We believe that the manuscript has improved substantially since the initial submission, and appreciate that you did quite a lot of work to address the concerns raised previously. Reviewers and editors agreed the new analysis makes a much stronger case, and that this work will make a valuable addition to the field. Reviewers raised a few remaining issues, please find these below. We ask you to address these and invite you to submit a revised version of your manuscript at your earliest convenience.

*Reviewer #1:*

This paper studies the role of lateral prefrontal cortex (LPFC) and anterior cingulate cortex (ACC) in reversal learning. The authors suggest that LPFC plays a role in computing the probability that the animal will make a certain choice (termed choice probability), whereas ACC signals the reward prediction error. Interestingly, narrow spiking cells (putatively inhibitory neurons also known as fast spiking units) had a higher correlation with these task-relevant parameters, compared to broad spiking cells (putatively excitatory neurons also known as regular spiking units).

Next, the authors define electrophysiological cell types (termed e-types), based on spike waveform and firing patterns. The narrow spiking cells are subdivided into 3 subclasses, termed N1, N2 and N3. Notably, the same subclass of narrow spiking cells, N3, had a correlation with choice probability in LPFC and a correlation with reward prediction error in ACC. Neither of the other narrow spiking subtypes had a significant correlation with either parameter in either area.

In the final part of the paper, the authors examine the phase-locking behavior of these N3 cells to the local field potential (LFP). They find that in LPFC, N3 cells phase lock to gamma (35 – 45 Hz) during the initial learning stage shortly after rule reversal, but as learning progresses and performance reaches a new plateau, their phase locking switches to the beta-band (15 – 30 Hz). Perhaps most remarkably, the N3 cells in ACC showed a similar reversal learning stage dependent phase locking behavior; to elaborate, they phase-locked to gamma only when the reward prediction error was high (i.e., shortly after rule reversal).

The main finding of this paper, that a specific electrophysiological subclass of narrow spiking cells serve important roles in a reversal learning by preferentially phase-locking to gamma band LFP, would be of broader significance and impact if this finding could be generalized to other brain regions, behavioral tasks and model species. This paper cites several precedents in the literature that define e-types. Specifically, Markram et al. (2015) define 11 e-types; Gouwens et al. 2019 define 6 e-types that constitute narrow spiking cells (referred to as fast spiking cells in Gouwens et al). For sake of future efforts to study e-types and their functional roles, it would be important to reconcile these disparate definitions of e-types.

Moreover, as mentioned in the Discussion section of this paper, there are several other papers showing that subclasses of narrow spiking neurons have different relationship with gamma (Shin and Moore 2019; Onorato et al., 2020). It would be very interesting and important to know whether the 3 narrow spiking e-types discussed in this paper match up with the subclasses in the aforementioned papers.

In sum, this paper is a valuable addition to the reinforcement learning literature as well as neuronal cell types and neural oscillations literature. However, it is unclear how the e-types defined in this paper will tie into other neuronal categorizations in recent literature. This link to prior work will be important for broader significance.

Comments for the authors:

This paper has made significant improvements from the previous version. Most importantly, the implementation details of the circuit simulation are clarified. The vast majority of my prior concerns have been addressed. I have only a few suggestions remaining.

1. Given that reward prediction error analysis is critical to the thesis of this paper, I am still of the opinion that it would be important to include the PSTH aligned to the reward, for narrow spiking and broad spiking neurons (as in Figure 2) as well as for important e-types (as in Figure S3).

2. The added classifier analysis or predicting cell classes from their correlations with learning variables is very interesting. However, I am not clear on exactly what was used to train the SVM. The way I currently understand this analysis is that in LPFC, correlation between firing rate and p(choice) was calculated for each neuron – and this one-dimensional vector, the size of which is (Number of neurons)X1, was used to train the SVM. Is this the case? Please clarify.

3. Figure S5 E and F: it is hard to see a trend in these plots. I suggest either making the dots transparent; or plotting the data as a 2D-histogram. This way it would be possible to discern where the data is the densest.

4. In Methods, the numbering in the equations are not unique (there's two Equation 2 and two Equation 3). Please correct.

5. The following sentences in Supplementary Online Information needs to be corrected as indicated:

"These circuit motifs are provided to provide a proof-of-concept that the observations can follows from biologically plausible motifs. These circuits motifs also provide predictions which can be tested in future studies."

*Reviewer #2:*

In this work, Boroujeni et al. investigated the role of different cellular subtypes in the lateral prefrontal cortex (LFPC) and anterior cingulate cortex (ACC) of the rhesus macaque as the animals performed an attention-demanding reversal-learning task. The authors use an attention-augmented reinforcement learning model to track the trial-by-trial values of key decision-making variables which were then correlated against the neural activity. The cellular population was separated into broad and narrow spiking neurons using features computed from the extracellularly recorded waveforms. The authors find that the activity of the narrow spiking cells in the LFPC is correlated with the choice probability, whereas the activity of narrow spiking cells in the ACC is correlated with reward prediction errors. Interestingly, the authors find that further splitting the population of broad and narrow spiking cells into subtypes revealed that both the choice probability in LPFC and the reward prediction error in the ACC were encoded by a specific subtype of putative interneuron. The authors show that the spike-field phase synchronization of this putative interneuron subgroup is also modulated by choice probability in the LFPC and reward prediction error in the ACC, mirroring the result from their single-unit correlation analysis. The authors use these results to propose a biologically plausible circuit model of how learning in such a task might be implemented through interneuron-specific synchronization.

The analysis is thorough and the authors present a nice narrative of the results, even though in some cases my interpretation of the data is a little more mixed than what is written in the paper. For example, the authors are eager to point out that their results are "interneuron specific" and yet the data that they show suggests otherwise. Take the spike-LFP synchronization results shown in Figure S15, where it seems that the modulation of pairwise phase consistency with p(choice) could also be present for the B1 cluster of cells in addition to the N3 group (no stats shown). The same could be true for the B2 type in the ACC, which seems to show differential effects for high and low RPE.

Are these real effects or are these anomalies that are biased by a few outliers? In either case, please clarify.

Thank you for including the new supplementary figures; I can really appreciate the additional amount of work that must have gone into preparing the new controls for the second submission of the paper. The addition of the example model fittings (Figure S5) and the correlation of the firing rate from the two example cells with the RPE and p(choice) (Figure S10) are very nice. I would recommend to the authors to move the two examples in Figure S10 to one of the main figures. In the first submission, the focus of the paper was predominantly on the N3 subtype and its specialized functional properties in ACC and LPFC. The new figures however (specifically Figure S15) show that the story is a little more mixed than originally presented. B1 for example in LPFC shows differential effects for high and low P(choice) and B2 in ACC shows differential effects for high and low RPE. In any case, the new figures provide a much more complete story and I feel made the paper stronger.

---

## [Author Response]

[Editors’ note: the authors resubmitted a revised version of the paper for consideration. What follows is the authors’ response to the first round of review.]Reviewer #1:Boroujeni et al. recorded extracellular spikes from single neurons in brain areas LPFC and ACC in two awake behaving macaques that were performing a reward reversal learning task. They classified the recorded neurons into various subtypes, and investigated how neuronal activity in these different subtypes related to the variables of the behavioural task.The paper's clear and primary strength is the classification of extracellularly recorded neurons into broad- and narrow-spiking neurons, and even further into subtypes of these two classes. While a split based purely on spike waveform shape into broad- and narrow-spiking is relatively common, the cluster-based classification into subtypes based on various additional parameters like spike variability is novel and potentially illuminating. The authors furthermore convincingly demonstrate that the recorded narrow-spiking neurons (often labelled "putative inhibitory interneurons") are indeed likely inhibitory in nature, by showing that the net effect of a spike in these cells on the surrounding population spike rate is negative. The analysis choices in this part of the paper were clear, well-motivated, and well-presented.However, the bulk of the paper is taken up by the relationship between neuronal spiking and variables from the behavioural task, specifically choice probability (p(choice)) and reward prediction error (RPE). Here, the conclusions appear not backed up by the data, for several reasons.First of all, the authors only present results for correlations with RPE in the reward window, and results for correlations with p(choice) in the stimulus windows. One of the main conclusions of the paper is that LPFC neurons code for p(choice) whereas ACC neurons code for RPE. However, correlations with RPE in the stimulus windows and p(choice) in the reward window are never shown. Furthermore, the authors demonstrate that, purely given the task structure, RPE and p(choice) are almost perfectly negatively correlated (r = -0.928, Figure S4). It is therefore very possible that the crucial split is not between p(choice) and RPE as the determinant of neural activity, but simply the time window in which these are analyzed.

We believe there is a misunderstanding regarding some aspects of our data that we aim to address in three ways.

Firstly, conceptually the fact that RPE and p(choice) are anti-correlated (at r = -0.96, revised Figure S 5) does not change the interpretation of our findings during the cue and reward period. We report highly specific effects at the moment in the trial when p(Choice) and RPE are computed: A reward prediction error is by definition occurring after reward is processed and compared with an expected value, and not when a cue is processed. When a cue is processed the expected value of the cued information is reactivated and translated into a choice probability. We therefore show for LPFC and ACC and for broad and narrow spiking neuron types that after a cue p(choice) is correlated in narrow spiking neurons in PFC but not in ACC or for broad spiking neurons. Similarly, the reward prediction error (RPE) is computed when the outcome is received and during that time the RPE correlates with ACC narrow spiking neurons but not with broad spiking neurons and not in LPFC.

To address the reviewer’s concern whether the cue triggered correlation of p(choice) is not a correlation with RPE (and vice versa), we adjusted the interpretation in the text. In the revised text we more clearly emphasize that the time of low choice probabilities and high prediction errors demarcate a time of enhanced uncertainty about the relevant stimulus color. According to this perspective the firing and gamma synchrony correlations of narrow spiking neurons in PFC and ACC are reflecting not p(choice) and RPE per se, but a period with uncertainty about cues and outcomes. In the revised abstract we write:

“One of these interneuron subclasses showed prominent firing rate modulations and (35-45 Hz) gamma synchronous spiking during periods of uncertainty in both, lateral prefrontal cortex (LPFC) and in anterior cingulate cortex (ACC). […] Computational modeling this interneuron-specific gamma band activity in simple circuit motifs suggests it could reflect a soft winner-take-all gating of information having high degrees of uncertainty.” (abstract)

We hope that these changes address conceptually the reviewer’s concern.

Secondly, we also want to point out a methodological aspect. Applying partial correlations of firing rate and p(choice) and RPE with variables that are so highly correlated leads to ambiguous results. We therefore refrain from attempting to perform partial correlations. What seems to matter is the clear interpretation about which latent decision variable (after cue onset) and learning variable (after reward onset) should emerge at what time point during the trial (cue period versus reward period).

Thirdly, experimentally dissociating choice probability and reward prediction errors might be interesting. In our experiment the correlation of p(choice) and RPE cannot be dissociated cleanly (as in most reinforcement learning tasks). This has, however, also not been the focus of our manuscript.

Second, the authors present a "circuit architecture" that might account for the observed results. In the Results, this model is presented as though it were a computationally implemented biophysical neural circuit model that makes predictions that are in line with the observed data. I cannot find details of the implementation of such a model in the Methods, which makes the status of the predictions here unclear. It is not explained why two equally-valued objects would lead to gamma synchronization, whereas two objects of unequal value lead to beta synchronization (the key conclusion derived from the model). This appears to depend on total input strength, but it is hard to see why 0.5 + 0.5 (equal value, numbers provided by authors) would result in higher input than 0.8 + 0.2 (unequal value, again numbers from this paper, Figure 9). These choices, and others, appear arbitrary. In general, the description of the model in Results reads more like an interpretation/Discussion section than an outline of model-derived Results.

We apologize for the confusion about the computational circuit explanation. We adjusted this in the revised manuscript and provide a detailed description of the circuit and their implementation in firing rate models in a new Supplementary Online Information, new Suppl. Figures S17 and S18 showing the simulation results and in a simplified Figure 9.

The Suppl. Online Information now provides the implementation details and a discussion on what other architectures could underlie the empirically observed cell – type specific gamma synchronization effects. It has these four separate sections:

1. Overview of circuit modeling

2. E-E-I circuit motif realizing the switch from gamma to beta frequency synchronization

3. E-I-I circuit motif realizing the switch from gamma to theta frequency synchronization

4. Discussion of circuit motifs, relation to other models and experiment

We also clarify in the Results section of the main text that the circuit motifs are conceptualized in order to understand the possible circuit function of the observed gamma synchronization of the N3 interneuron class. We kindly refer to the Results section entitled “Circuits model of interneuron-specific switches of gamma to beta or theta synchronization”. In that section we state that

“… oscillatory activity signatures might inform us about the possible circuit motifs underlying uncertainty-related related computations. These computations are formally described in the reinforcement learning framework allowing us to propose a linkage of specific computations to oscillatory activity signatures and their putative circuits as proposed in the Dynamic Circuits Motif framework (Womelsdorf et al., 2014).”.

We then go on to explicitly refer to the Supplementary Online Information that contains detailed descriptions of the model details and the simulation results:

“To show the feasibility of this approach we devised two circuit models that reproduces the gamma band activity signatures in LPFC and ACC using populations of inhibitory cells modeled to correspond to N3 e-type cells (for modeling details, see Suppl. Online Information)”

The revised text also explicitly states that

“… [t]he described circuits provide proofs-of-concept that the synchronization patterns we observed in the N3 e-type interneurons in ACC and LPFC during periods of uncertain values and outcomes can originate from biologically realistic circuits.”

We believe the addition of the circuit models enhances the impact of the paper by explicitly pointing to possible circuit function of the observed interneuron specific gamma activity.

Third, the presented empirical evidence for narrow-spiking cells (or, more specifically, the N3-subtype) engaging preferentially in gamma-band synchronization, whereas broad-spiking cells engage preferentially in beta-band synchronization, is modest. Interneuron engagement in gamma rhythms is expected from the literature, of course, but in the present dataset this is less clear-cut. In particular, the spectral peaks in Figure 6C are quite similar between broad- and narrow-spiking, and labelling the former "beta" but the latter "gamma" requires a more thorough analysis than is now presented.

Our highly specific gamma synchronization effects of the N3 e-type neurons are not described in Figure 6 to which the reviewer refers to. We adjusted the revised text to make this more clear, added new analyses results and describe the statistics more explicitly to convey our specific findings:

1. We explicitly report that both, narrow and broad spiking neurons show beta spike-LFP peaks (Figure 6, Suppl. Figure S4). This analysis was task-epoch independent. Please also note that we describe these results in the section entitled “Narrow spiking neurons synchronize to theta, beta and gamma band network rhythms.”, including links to relevant literature. This shows we do not claim a simple beta – gamma dissociation.

2. Among the individual cell classes, the spike-LFP synchrony showed significant ~40 Hz gamma peaks in ACC for classes N2 and N3. The other cell classes in ACC and all of the cell classes in LPFC did not show 40Hz gamma in a task-epoch independent way. These results (for all cell classes in each area) are provided explicitly in Suppl. Figure S12.

We consider the results from 1) and 2) not essential for our manuscript’s main message. They are provided to more comprehensively provide information about the cell classes, as the clustering procedure did not take into account synchrony. As another reviewer pointed out, the more information we provide about the cell classes the easier it will be in the future to identify possible subclasses and compare results from different studies.

3. We provide statistical evidence that gamma spike-LFP synchronization emerges transiently after a cue only for the N3 class in LPFC during low choice probability trials, and only for the N3 class in ACC during trials with low reward prediction error. The statistics for this cell class specific transient gamma increase is provided in Figure 7C (for LPFC) and in Figure 8F (for ACC). The specificity of these findings is now documented in the new Suppl. Figures S13 and S14 (please also see below).

4. In response to the reviewer we now provide for each of the cell sub-classes in a new Suppl. Figure S15 the time-frequency spike-LFP synchronization cue-aligned for low and high p(Choice) and reward-aligned for high and low RPE trials for PFC and ACC, respectively. These figures are noisy, because of the typically low number of spikes, but they do show that class N3 has some 3545 Hz synchrony in each brain area that other cell types do not show. We revised the text to describe these new results.

Fourth, there are some issues with reporting, where occasionally results are only reported for the narrow-spiking cells and not for the broad-spiking cells, or it is unclear whether a stated result holds for all or just a subset of cells, etc.

Thank you for pointing this out. We added missing information and carefully checked that the revised text reports all main results also for the broad spiking cells. The newly added results did not change the conclusions from the study.

Finally, all results are shown aggregated over two animals, while it is important to know how the key results hold in the two animals separately.

To address this comment we added the results from the key analyses for each monkey separately as supplementary results. In summary, both animals show similar result patterns in their response to color onset and reward onset, their firing rate correlation, and for both animals we verified the cell clustering separately:

“New Supplementary Figure S6 shows the main firing rate results for each of the two monkeys separately. For both monkeys narrow spiking neurons have stronger event-related firing and correlation with choice probabilities and reward prediction errors. This supports our main findings.”

We added new result panels in Supplementary Figure S8E and F that show monkey specific validation of cell clusters using two established methods. Both methods provided results showing that the cell classes were reliable in each monkey. This was true also for the main interneuron class N3 that carries the main results of our manuscript.

I mention some additional recommendations here.At the very least, correlation analyses for both p(choice) and RPE should be shown for all time windows, to allow a proper assessment. If the authors indeed wish to maintain the hard claim of a dissociation ACC<>RPE and LPFC<>p(choice) this should explicitly be tested by e.g. directly comparing the correlations with the two behavioural variables.

Please also see our reply to the comment starting “However, the bulk of the paper…” above. We added the results of the suggested correlation analyses. We indeed do see only an ACC<>RPE and not an LPFC<>RPE effect, and we do find only a LPFC<>p(choice) and not an ACC<>p(choice) effect.

RPE and p(choice) are highly negatively correlated at r =-0.928 (shown in Suppl. Figure S5B), so that a third variable (firing rate) that correlates positively with one variable will negatively correlate with the other variable. However, choice probability is most meaningfully tested after the color cue onset (the color values are needed to compute p(Choice) and RPE is most meaningfully correlated after reward onset (when the value of the chosen stimulus is compared to the experienced reward).

We adjusted the text to make this more explicit and refer directly in the main text to the definitions of p(choice) (formalized in Equation 4, Methods) and of RPE (formally described in Equation 2, Methods).

In response to another comment, we also added results of the synchronization analysis for both ACC and LPFC for low and high p(choice) and low and high RPE (shown in new Suppl. Figure S13 and S14).

The model should be specified in much more detail. Specifically, the assumptions built into it should be clearly defined, and the quantitative predictions derived from it should be presented.

We agree. A substantially revised Results sections summarizes the circuit motifs. The circuit hypotheses shown in Figure 9 is simplified and streamlined. We added the details about the circuit motifs that reproduce the gamma synchronization effects in LPFC and in ACC and describe the assumptions explicitly. We now also show stimulation results in a new Suppl. Online Information and Suppl. Figures S17 and S18.

The circuit models are not provided to generate quantitative predictions. We specify this now explicitly in the revised Results and write e.g.:

“To show the feasibility of this approach we devised two circuit models that reproduces the gamma band activity signatures in LPFC and ACC using populations of inhibitory cells modeled to correspond to N3 e-type cells (for modeling details, see Suppl. Online Information) …”

The intended value of the circuit models is described in the revised text to …

“… provide proofs-of-concept that the synchronization patterns we observed in the N3 e-type interneurons in ACC and LPFC during periods of uncertain values and outcomes can originate from biologically realistic circuits.”

The model results also support our main interpretation that the gamma activity might indicate the gating of competing information (in LPFC) and the detection of mismatches of experienced reward and the expected value of the chosen stimulus.

Please also see our second response to reviewer #1 above.

I understand that the data are not yet publicly released, as others from the same lab are still working on the same data (which is common in the field). However, I would urge the authors to make the source code for all reported analyses publicly available already, to greatly improve transparency and replicability. ("Upon reasonable request" is not sufficient for this goal.)

We fully agree. We mentioned the open-source code used for clustering in the method section.The source-code for the adaptive spike removal is now added (https://github.com/banaiek/ASR).

We also will add a new GitHub link for the complete spike-triggered MUA analysis (with example scripts) upon publication of this paper.

All analyses code will also be linked and made available on the lab website via http://accl.psy.vanderbilt.edu/resources/code/

In general, the narrative could be streamlined a bit, as it currently stands the manuscript is hard to read.

We attempted to streamline the narrative (starting with an improved abstract). The revised manuscript has all changes highlighted in red font.

Thank you for the many helpful constructive comments.

Reviewer #2:

This paper studies the role of lateral prefrontal cortex (LPFC) and anterior cingulate cortex (ACC) in reversal learning. The authors suggest that LPFC plays a role in computing the probability that the animal will make a certain choice (termed choice probability), whereas ACC signals the reward prediction error. Interestingly, narrow spiking cells (putatively inhibitory neurons also known as fast spiking units) had a higher correlation with these task-relevant parameters, compared to broad spiking cells (putatively excitatory neurons also known as regular spiking units).Next, the authors define electrophysiological cell types (termed e-types), based on spike waveform and firing patterns. The narrow spiking cells are subdivided into 3 subclasses, termed N1, N2 and N3. Notably, the same subclass of narrow spiking cells, N3, had a correlation with choice probability in LPFC and a correlation with reward prediction error in ACC. Neither of the other narrow spiking subtypes had a significant correlation with either parameter in either area.In the final part of the paper, the authors examine the phase-locking behavior of these N3 cells to the local field potential (LFP). They find that in LPFC, N3 cells phase lock to gamma (35 – 45 Hz) during the initial learning stage shortly after rule reversal, but as learning progresses and performance reaches a new plateau, their phase locking switches to the beta-band (15 – 30 Hz). Perhaps most remarkably, the N3 cells in ACC showed a similar reversal learning stage dependent phase locking behavior; to elaborate, they phase-locked to gamma only when the reward prediction error was high (i.e., shortly after rule reversal).These results are generally well supported by rigorous statistics and sophisticated analyses. However, there are several weaknesses.First, while the claim that LPFC encoded choice probability is well supported, the claim that ACC encodes reward prediction error is not as well substantiated. As seen in Figure 3, percent neurons showing significantly correlation between their firing rate and reward prediction error is not very different between LPFC and ACC, and quite similar between broad spiking and narrow spiking units within ACC.

We did not intend to convey that RPE is not encoded in LPFC, or only encoded in ACC.

We adjusted the text to clarify this and write about Figure 3 that “…23% of LPFC and 35% of ACC…” neurons show sign. rate<>RPE correlations, reporting that the proportions are significantly different from zero. This corresponds well to our previous work that comprehensively surveyed how different types of RPEs are encoded in LPFC and in ACC with LPFC showing RPE encoding slightly less likely and slightly later than ACC when considering both, single isolated neurons as in this study, and multiunit activities (Oemisch et al., 2019).

What we found is that the positive correlation of RPE and firing rate had a stronger effect size in narrow spiking ACC neurons (Figure 3D) and is driven by a significant correlation of the N3 subclass (Figure 5D).

No other narrow or broad spiking neuron class has an average significant positive correlation strength.

We adjusted the text at various place to convey this more clearly in the revised manuscript. E.g., we write

“…time-resolved analysis of the strength of the average correlations revealed a significant positive firing x RPE correlation in the 0.2-0.6 s after reward onset for ACC N-type neurons, which was absent in LPFC (ACC, n=43 N-type neurons, randomization test p<0.05; LPFC: n=31 N-type neurons, no time bin with sign.; Figure 3C,D).”

We confirmed that this result was evident in each of the monkeys when tested separately (and added this result to the revised manuscript).

Second, the authors build a reinforcement learning model to calculate "Choice Probability", which quantifies the probability that the animal will select the rewarded stimulus. According to this definition, choice probability should dip upon reversal, and rise to a new plateau after several trials. However, this metric is fairly unintuitive, not to mention in conflict with existing nomenclature (e.g., Nienborg, Cohen and Cumming 2012). It would be helpful to have an accompanying plot of how the firing rate and phase locking behavior of each neuronal type changes as a function of trials after reversal.

We did not explicitly describe the p(choice) in the original manuscript and apologize for this oversight. In the revised text we followed the reviewer’s suggestion and added the progression of choice probabilities since trials after reversal in an extended Supplementary Figure S5. Choice probabilities positively correlate at r=0.27 with the trial number since reversal (as expected).

We also adjusted the text to more explicitly convey how the reinforcement learning literature uses a softmax (or Boltzman) equation to translate stimulus values into choice probabilities. We adjusted the methods section explicitly and write:

“Equation 4 defines the choice probability, or p(choice), that is used for the neuronal analysis of this manuscript (Sutton and Barto, 2018). P(choice) increases with trials since reversal (Supplementary Figure S5A,E), indicating a reduction in the uncertainty of the choice the more information is gathered about the value of the stimuli.”

We refrain from adding a rate or phase locking analysis as a function of trials since reversal because this would require trial by trial estimates of phase synchronization in short time windows around cue or reward onset. The isolated neurons fire few spikes in these epochs (Figure 4B), rendering the estimate of phase consistency noisy. We believe the trial-by-trial analysis will be more relevant when analyzing larger datasets that do not separate neurons into sub-classes with few and low firing neurons (given the small proportion of interneurons in the population).

Third, the extent to which choice probability encoding neurons and reward prediction error encoding neurons in each area falls into a specific e-type is not shown.Undoubtedly, it is noteworthy and remarkable that N3 is the only e-type that shows a positive correlation with choice probability in lateral prefrontal cortex and a positive correlation with reward prediction error in ACC (Figure 5). But do all choice probability encoding neurons in LPFC and reward prediction error encoding neurons in ACC fall into the N3 e-type?

The key finding is that the N3 e-type shows the strongest correlations with p(choice) (in LFPC) and RPE (in ACC). It does not entail that neurons in other e-type classes would not significantly encode p(choice) and RPE. In each class a small fraction of neurons individually shows significant correlations.

We clarify this in the text and added a new classification analysis to address the reviewers question.

First, we adjusted the text at various places to more explicitly refer to the “proportion of significance” and to the “strength / effect size” of the correlation when describing the results. E.g., we write

“… on average 23% of LPFC and 35% of ACC neurons showed significant firing rate correlations with RPE … ”, while “… analysis of the strength of the average correlations revealed …”

Second, to quantitatively convey the statistical nature of the main findings we added an analysis that predicted the cell class label of an individual cell based on its correlation of firing rate and p(choice) in LPFC and RPE in ACC. The confusion matrix results shown in the newly added Supplementary Figure S11 reveals that p(Choice) allowed to significantly predict a neurons’ class for various e-types and not only for the N3 e-type (for which the correlations were strongest as shown in other analyses). We added to the main text that in

“… LPFC, a linear classifier trained on multiclass p(choice) values was able to label N3 e-type from their p(choice) value with and accuracy of 31% (Suppl. Figure S11A).” and in

“…ACC a linear classifier trained on multiclass RPE values was able to label N3 e-type from their RPE value with and accuracy of 34% (Suppl. Figure S11B).”

Further, the task-dependent phase locking behavior of e-types other than N3 are not shown. Given that N3 is the only NS e-type that shows a relationship with task-relevant parameters, I would expect the task learning dependent phase-locking behavior to also be unique to N3, but this result is not presented in this paper.

We added the phase locking for all cell types around cue onset for low and high p(choice) in LPFC and around reward onset for low and high RPE in ACC as new Suppl. Figure 15. The plots show that the N3 e-type is the only narrow spiking e-type with a gamma response after cue (in LPFC) / after reward (in ACC) in the period of enhanced reward uncertainty (low p(choice) and high RPE).

Additionally, as requested, we now provide for each cell type the gamma spike-LFP phase locking for low and high p(choice) in LPFC, and low and high RPE in ACC in the new Suppl. Figure’s S13E-H and S14EH.

Notably, gamma synchrony is evident most clearly for the N3 e-type in LPFC during low p(choice) (Suppl. Figure S13E) and the N3 e-type in ACC during high RPE trials (Suppl. Figure S14H).

These results support our original conclusions. The revised text is adjusted to refer to these new results.

Finally, the conceptual model in Figure 9 captures the results presented in this paper and gives rise to testable predictions. It seems that some predictions of this model should be testable with the presented data. For example, the prediction that in LPFC, broad spiking cells fall into two functional categories, whereas N3 cells are more functionally homogeneous, would be an interesting prediction to test. Further, the prediction that in ACC, broad spiking cells encode reward whereas N3 cells encode reward prediction error is easily testable and would strengthen the conclusions of this paper.

In response to editorial and other comments we simplified the scope of the circuit modeling. We devised these models to account for the observe gamma synchronization of the N3 interneuron class and inform us about possible circuit functions of this gamma synchronization. Testing predictions of the model’s broad spiking neurons was not the intention and would exceed the scope of this manuscript.

In the revised manuscript we make this explicit by writing, for example, in the Results section:

“The described circuits provide proofs-of-concept that the synchronization patterns we observed in the N3 e-type interneurons in ACC and LPFC during periods of uncertain values and outcomes can originate from biologically realistic circuits. The results justify future studies testing detailed predictions that can be derived from these circuit motifs.”

The main finding of this paper, that a specific electrophysiological subclass of narrow spiking cells serve important roles in a reversal learning by preferentially phase-locking to gamma band LFP, would be of broader significance and impact if this finding could be generalized to other brain regions, behavioral tasks and model species. That said, there are already several papers in the literature that define e-types. Specifically, Markram et al. (2015) define 11 e-types; Gouwens et al. 2019 define 6 e-types that constitute narrow spiking cells (referred to as fast spiking cells in Gouwens et al). For sake of future efforts to study e-types and their functional roles, it would be important to reconcile these disparate definitions of e-types.

We agree. But we particularly want to highlight that our findings in two brain areas in the monkey is already an important, newly achieved milestone. There is no other paper to our knowledge that characterizes specific interneuron functions in the nonhuman primate in ACC and LPFC and succeeds to find a functional similarity for the same electrophysiological cell type.

To address the reviewer’s suggestions, we added a more explicit discussion that is aimed at linking the different e-typing approaches from in-vitro and from in-vivo studies. We write in the revised Discussion section:

“The first implication of our findings is that narrow spiking neurons can be reliably subdivided in three subtypes based on their electrophysiological firing profiles. […] These results illustrate that our three interneuron e-types will encompass further subclasses that future studies should aim to distinguish in order to narrow the gap between the in-vivo e-types that we and others report in the monkey, and the in-vitro e-types in the rodents that are more easily mapped onto specific molecular, morphological and genetic make-ups (Markram et al., 2015; Gouwens et al., 2019).”

Moreover, there are at least two other papers showing that subclasses of narrow spiking neurons have different relationship with gamma (Shin and Moore 2019; Onorato et al., 2020). It would be very interesting and important to know whether the 3 narrow spiking e-types discussed in this paper match up with the subclasses in the two aforementioned papers.

We agree these are very important references and added them at different places in the revised Discussion section.

Please see the quote in the previous reply. Additionally, when discussing the importance to link in the future e-types to morphological and molecular types in mice we refer to the Onorato study (of which the last author of this study is a co-author) and write:

“…As a caveat, this mapping of cell types between species might also reveal cell classes and unique cell class characteristics in nonhuman primate cortices that are not similarly evident in rodents as recently demonstrated in a cross-species study of non-fast spiking gamma rhythmic neurons in early visual cortex that were exclusively evident in the primate and not in mice (Onorato et al., 2020).”

When discussing when gamma synchrony has been observed in specific cell classes, e.g. we now write in the revised text:

“Such an intrinsic propensity for generating gamma rhythmic activity through, e.g. GABA_a_ergic time constant, is well described for PV+ interneurons (Wang and Buzsaki, 1996; Bartos et al., 2007; Womelsdorf et al., 2014b; Chen et al., 2017) and for some cells even for states with relatively low excitatory feedforward drive that might be more typical for prefrontal cortices than earlier visual cortices (Cardin et al., 2009; Vinck et al., 2013; Shin and Moore, 2019; Onorato et al., 2020).”

In sum, this paper is a valuable addition to the reinforcement learning literature as well as neuronal cell types and neural oscillations literature. Some additional analyses could strengthen the conclusions of this paper. It is unclear how the e-types defined in this paper will tie into other neuronal categorizations in recent literature. This link to prior work will be important for broader significance.

Thank you for the positive comment. We fully agree with the importance to link to other classification schemes and hope the revised discussion and the added details in the supplementary materials will support this goal.

Comments for the authors:

I. Comments on Figures1. Figure 2 and Figure S6 shows the PSTH aligned to Feature 1 and Feature 2 based on the cue order (Motion first vs Color first). It would be highly relevant to also show the PSTH aligned to Feature 1, Feature 2 and Reward based on behavioral outcome (correct vs incorrect, and there are at least 3 different types of error outcomes; please see my comment III-2 in Comments on Methods below for elaboration). In particular, PSTH aligned to reward conditioned on behavioral outcome is crucial for interpreting Figure 3.

We appreciate your point of view. But this manuscript is about encoding of positive reward prediction errors and choice probabilities during learning with multiple sub-conditions and result panels (with 17 supplementary figures). The proposed (hit vs miss) error type analysis would be a different manuscript and beyond the scope of this paper, or it would add complexity that the revisions aim to reduce.

2. Figures 2 and 3: The correlation between firing rate and Choice Probability / RPE is interesting, but not very intuitive. It would be helpful to have a plot of Choice Probability and Reward Prediction Error as a function of trials since reversal, as well as the firing rate for each cell type and brain area as a function of trials since reversal. This way we can see whether LPFC NS firing rate after color cue onset tracks Choice Probability, and whether ACC NS firing rate after reward tracks RPE.

We added example learning blocks showing the progression of choice probabilities and RPE (and the value of the chosen stimuli) over trials in an extended Suppl. Figure S5A. We show the overall change of choice probabilities and RPE since the first trial after reversal in Suppl. Figure S5E,F and report the correlations of these variables with the trial-since-reversal (which is r=0.13 for p(choice) and r= 0.23 for RPE and trial-since-reversal). We showed more examples in an earlier publication (Oemisch et al., 2019).

To provide ore insights into the firing rate changes with p(choice) and RPE we provide an example for LPFC and ACC in new Suppl. Figure S10 and write e.g.:

“The on-average positive correlation of firing rate and p(choice) was also evident in an example N3 e-type cell (Suppl. Figure S10A-C).”

Please note that the average correlation of the N3 cell classes firing rate with p(choice) is significant but weak: r=0.08 in LPFC (and with RPE in ACC it is r=0.09) so that trial x rate plots will be noisy. The key result is that the narrow spiking neurons do show on average an on-response to the cue and reward onset and that only one of these classes (N3) shows significant correlations (and gamma synchronization).

3. Figure 4B firing rate unit is missing both the figure and in the main text.Figure 4C rastergram firing rate seems massively different from the average firing rate in 4B? e.g., for Figure 4C rastergram for N1, there seems to be ~5 spikes per 100ms, which would be ~50Hz, but the average firing rate for N1 is 4Hz?Also, please discuss why the narrow spiking firing rate is so low (assuming the firing rate unit was Hz, mean firing rate is <2Hz for N2 and N3). Narrow spiking firing rates have typically been reported to be ~10Hz in vivo.

Thank you for pointing us to this. The firing rate axes is log(spikes/sec.), so the firing rates for N1, N2, and N3 are on average 20, 3.8, and 4.3 spikes/sec – similar to previous studies / datasets in monkey prefrontal cortex (e.g., Ardid et al., 2015).

We corrected this in Figure 4B and the main text. Figure 4C are example raster’s selected to convey primarily how regular/irregular the firing was (as captured in LV and CV). They might differ in mean rate to the average class specific firing rates.

4. Figure 5: It is remarkable that N3 is the only e-type that shows a positive correlation with choice probability in LPFC and a positive correlation with reward prediction error in ACC. To what extent do choice probability encoding neurons and reward prediction error encoding neurons in each area fall into a specific e-type? I would like to know whether a neuron's e-type is predictable from task-dependent functional properties of the neuron.

That is an interesting question. To address this, we trained a classifier on the correlation values of e-types for p(choice) and RPE, and then predicted the class labels of neurons in a one to all classification process with the trained classifier. Two N-type classes (N1 and N2) were not large enough to survive the sampling we implemented in the analysis. From the other 6 e-types in ACC, the N3 e-type and two other broad spiking classes were predictable significantly higher than the chance level. However, only N3 e-type was not significantly confused with classifiers trained on other e-types.

We added the results in Supplementary Figure S11 and adjusted the methods and results text. We summarize the results by writing in the revised text:

“In LPFC, a linear classifier trained on multiclass p(choice) values was able to label N3 e-type neurons from their p(choice) value with and accuracy of 31% (Suppl. Figure S11A).” and “In ACC a linear classifier trained on multiclass RPE values was able to label N3 e-type neurons from their RPE value with and accuracy of 34% (Suppl. Figure S11B).”

5. Figure 6C: suggest plotting N3 in the same plot as Broad Spiking and Narrow Spiking units such that the magnitude can be compared more easily.In addition, please clarify what the y-axis of Figure 6c means (Peak densities of spike-LFP synchronization (PPC)). Is this simply the average PPC spectra? Or normalized for each unit in some way? I would recommend plotting the former, such that it is possible to compare which e-types have the best locking properties to which frequency band.

The magnitudes are directly comparable and the axis limits identical: Figure 6C shows the proportion of neurons in that population with a reliable (significant) PPC peak at that frequency. We clarified this in the revised figure axis and legend. The method section describes the computation explicitly.

Similarly, revised Supplementary Figure S12 shows the detailed information about the likelihood for significant PPC peaks for each cell class in each brain area (with the same [ylim] to ease comparison).

The values are directly comparable between plots.

6. Figure 7 and 8: It's very interesting that initially after reversal, N3 locks to gamma but later, as performance reaches a new plateau, N3 locks to beta. If you plot trial since reversal on the x-axis, and plot the peak of PPC spectra (averaged across N3 cells) on the y-axis, do you see a gradual change in peak frequency or is it more of a step function change after each reversal? Relatedly, if you plot the histogram of PPC spectra peak frequency across N3 cells, is it a bimodal distribution (one peak in beta and another peak in gamma) or is it unimodal?

That is an interesting approach. We have tried it per your recommendation. However most of classes did not pass the minimum spike number criterion per time window per trial. The result is too noisy to be meaningful. We are actively working on a new experiment that increases the number of learning blocks per recorded cell to more than 30 and expect this new design will allow more fine-grained trial-by-trial predictions of learning (and to see for how many neurons this is a one shot or a gradual learning change).

For the revised circuit models (Figure 9 and Suppl. Figures S17 and S18) we do describe mechanisms for a gradual gamma-to-beta switch mechanism (for LPFC) and gamma-to-theta witch mechanism (for ACC). We also added an explicit discussion of possible alternative switch mechanisms in the Suppl. Online Information. In the models we propose and simulated the switch depends on the heterogeneity of the inputs to the network cells (we kept the overall excitatory drive constant in both tested models). This provides a starting point to test in the future more precise predictions about how rapid the switch occurs. Our manuscripts can provide the rationale for these more specific tests in the future.

7. It would be interesting to know the behavior-dependent phase locking of other e-types as well. I suggest adding Figure 7 and 8 C and F for all e-types as a supplemental figure.

We added as new Suppl. Figure S15 the suggested information with time frequency plots for low and high p(choice) in LPFC and low and high RPE in ACC for B1-5 and N1-N3.

8. Were LPFC and ACC recorded simultaneously? If so, it would be very interesting to see if inter-area coherence mimics the changes in PPC. For example, does the gamma band coherence go up in the first few trials after reversal, followed by an increase in beta band coherence as behavioral performance plateaus?

That is a really interesting point but unfortunately the number of data points recorded simultaneously is not sufficient for this analysis.

9. Figure 9 outlays a really nice hypothesis that gives rise to testable predictions. Some of these predictions are testable within the data presented in this paper. I think it would significantly strengthen this paper if some of these predictions could be tested:

The original model we proposed was a major issue for the editor and other reviewers as it was not detailed enough. We therefore adjusted the model section and now present a model that has all implementation details presented in a new Supplementary Online Information.

We devised and implemented the circuit model primarily to document how the enhanced gamma during learning can emerge in the prefrontal cortex circuits (where gamma during learning switches to beta after learning) and in the anterior cingulate circuits (where gamma during learning switches to theta after learning).

We adjusted Figure 9 to show only the core circuit motifs that could reproduce the observed gamma to beta switch and the gamma to theta switch. We implemented and simulated the circuits in a firing rate model and show the results in new Supplementary Figures S17 and S18, and provide the details in an extended Suppl. Online Information that has these four sections:

1. Overview of circuit modeling

2. E-E-I circuit motif realizing the switch from gamma to beta frequency synchronization

3. E-I-I circuit motif realizing the switch from gamma to theta frequency synchronization

4. Discussion of circuit motifs, relation to other models and experiment

With all these changes we hope to provide a useful interpretation to our gamma synchrony finding. Testing the perditions from these circuits need to await more quantitative and biophysically detailed modeling that is beyond the scope of this paper (we plan to provide this in a follow up modeling paper).

We summarize the purpose of the circuit modeling and the limitation of our approach in the Suppl. Online Information by writing:

"… these models serve as a proof of principle, indicating how populations may be wired up to produce oscillations with different frequencies, but they cannot make conclusive predictions regarding the dynamics of the underlying interneurons, i.e. whether they are PV or SOM, or what type of spike patterns they produce. […] Nevertheless, we think it is reasonable to identify faster interneuron populations with PV+ interneurons given prior modeling studies (see next paragraph), and thereby putatively link them to the N3 e-type (see also Discussion of the main text)."

Figure 9 hypothesizes that in LPFC, Broad Spiking neurons should encode Value predictions; e.g., red-selective neurons that, after learning, fire more when red is being rewarded compared to when green is being rewarded. These Value-predictive neurons should fire similarly during learning, and is perhaps even predictive of the animal's choice on a trial-by-trial basis (e.g., on trials that red-selective neurons fired more during learning, the animal saccades according to the red stimulus). In contrast, N3 neurons should show no such Value-predictive behavior. Is there evidence of such prediction in the data?Relatedly, Figure 9 hypothesizes that in ACC, Broad Spiking neurons encode reward, whereas N3 encode RPE. According to this prediction, N3 activity should be higher for "surprise correct" trials shortly after reversal, and go down as performance plateaus, whereas Broad Spiking neurons should be excited by reward the same amount regardless of whether it is shortly after reversal or after behavioral performance has reached plateau. Is this seen in data? I think this would be made clear if the PSTH aligned to reward were plotted, as suggested in Comment 1.

Please see our reply to the previous comment regarding the revised and more detailed circuit models. They were conceived to reproduce the oscillatory signatures rather than as a reinforcement learning network that learns values in pyramidal cell populations. We therefore refrain from more detailed analyses of value coding in different cell populations.

This paper is about cell specific coding of uncertain cues (low p(Choice)) and uncertain outcomes (high RPE) and not about the coding of value in ACC and LPFC. We plan on adding the proposes analyses in future work with a separate set of analyses.

II. Comments on Main Text

1. "We next asked whether the narrow spiking, putative interneurons that encode p(choice) in LPFC and RPE in ACC are from the same electrophysiological cell type, or e-type (Markram et al., 2015)."There are ~11 e-types described in Markram et al., 2015. Further, Gouwens…Koch 2019 NN describes ~6 sub-e-types of Fast Spiking cells. I recommend the authors to speculate on how previously reported e-types match up with the e-types described in this paper.

We appreciate this comment and added a discussion of the putative relationship of the in-vitro and the in-vivo e-types to the Discussion section. Please also see the reply to the reviewer 1 comment starting “At the very least, correlation analyses…” above.

2. "Prior studies have suggested that interneurons have unique relationships to oscillatory activity (Cardin et al., 2009; Vinck et al., 2013; Voloh and Womelsdorf, 2018; Womelsdorf et al., 2014a),"I suggest adding Chen…Zhang 2017 Neuron to this list of references.

Thank you for pointing us to this. We added Chen, G., Zhang, Y., Li, X., Zhao, X., Ye, Q., Lin, Y.,.… and Zhang, X. (2017). Distinct inhibitory circuits orchestrate cortical beta and gamma band oscillations. Neuron, 96(6), 1403-1418.

3. Discussion section: There are at least two other papers showing that subclasses of narrow spiking neurons have different relationship with gamma (Shin and Moore 2019 Neuron; Onorato…Vinck 2020 Neuron). It would be an interesting addition to the Discussion section to speculate on whether the 3 narrow spiking e-types discussed in this paper match up with the subclasses in the two aforementioned papers.

Thank you for pointing us to the papers, We added specific sentences in various Discussion sections. We outlined this in detail in reply to the comment starting “Moreover, there are at least two other papers…” above.

III. Comments on Methods

1. In general, the Method section is not consistent about referring to relevant figures for the analyses being described. It would really help the reader if the analyses that went into each figure were clarified: e.g., "Statistical Analysis of time resolved spike-LFP coherence for putative interneurons and broad spiking neurons (Figure 7, 8)"

We added explicitly the specific Figure numbers / Suppl. Figure numbers to the individual methods sections in the revised submission. We also changed the order of some methods sections to more fairly reflect the order in which they are applied in the Results section.

2. Task design: "Color-reward associations were reversed without cue after 30 trials or until a learning criterion was reached, which makes this task a color-based reversal learning task. "It seems that a strategy that a monkey might employ would be to count the number of trials after reversal to anticipate when the next reversal would happen, which would rely on a different mental strategy than reversal learning tasks where the reversal points are not predictable. Is there any behavioral evidence that would discount the possibility that the monkeys are counting?

Yes, in previous studies we compared quantitatively which of many different reinforcement learning (RL) models, Bayesian models and hybrid Bayes-RL best accounts for the behavior of the animals (Oemisch et al., 2019). The optimal Bayesian model would come closest to very fast changes of behavior after or at the time of reversal but it did not fit the monkeys’ behavior well. Monkeys are slower than predicted by optimal models and are slower than predicted from anticipating the trial of reversal (as suggested by the reviewer).

For estimating the RPE and (choice) we used the best-fitting (cross validated) RL model. We refer explicitly to this in the revised methods:

“This color-based reversal learning is well accounted for by an attention augmented Rescorla Wagner reinforcement learning model (‘attention-augmented RL’) that we previously tested against multiple competing models (Balcarras et al., 2016; Hassani et al., 2017; Oemisch et al., 2019). Here, we use this model …”.

"Hence, a correct response to a given stimulus must match the motion direction of that stimulus as well as the timing of the dimming of that stimulus."In this task, there appears to be one way to be correct, but several distinct ways of being incorrect. First, the monkey could be incorrect in both the timing and the saccade direction. Second, the monkey could be correct with the timing but incorrect with the direction. Third, the monkey could be correct with the direction but incorrect with the timing. The third outcome could be further subdivided into premature response versus late response. The reason why a monkey might make each mistake is different. Only the first scenario supports the possibility that the monkey thought the other color was being rewarded, e.g., shortly after reversal. It would be interesting to know the proportion of each error type as a function of trials since reversal. Furthermore, I would expect the negative reward prediction error to be most prominent in the first type of error. Hence, it would make sense to me if only the first type of error was considered when calculating choice probability and reward prediction error.

The reviewer is correct about the error types. We added error distribution information to the revised text (in the methods where other behavioral results were described already in order to not disrupt the flow of the result section). We write:

“Monkeys performed the task at 83 / 86 % (monkey’s H / K) accuracy (excluding fixation break errors). Of the 17/14 % of errors were composed on average to 50 / 50% of erroneous responding to the dimming of the distractor when it dimmed before the target and 34 / 37 % of erroneous responding at the time when target and distractor dimmed simultaneously but the monkey chose the distractor direction, and 16 / 13 % of error were responses when the target dimmed before any distractor dimming and the choice was erroneously made in the direction of the distractor.”

For computing the choice probabilities, we use all trials where the monkey made a choice rather than a fixation break or a no-response. In the model, the choice probability is not computed only based on color values, but on location and motion values too (location and motion direction of the stimuli are sources of uncertainty when making a choice). When monkeys make an unrewarded choice at the wrong time, to the wrong direction, or to the wrong stimulus, then this behavior still is a choice initiated by the animal. The choice probability estimate is not supposed to only include choices of one color versus another color. Rather, the choice probability estimate was used because it quantifies how certain the animal was when making the choice. (Analysis of the encoding of color-value estimates is not part of this manuscript but is a separate project).

Regarding RPE’s, this paper reports cell specific correlations with positive RPEs on correct trials (it is mentioned in the methods section). The errors that lead to negative RPEs are too rare in this design to allow a strong analysis in classes with low number of cells and low firing single neuron. A probabilistic reward schedule might be useful in the future to get more error trials. A separate analysis of negative versus positive prediction errors and how they distribute and carry different feature information is provided on a larger dataset in our previous work in Oemisch et al., 2019.

3. "Here, we use this model to estimate the trial-by-trial fluctuations of the expected value (EV) for the rewarded color and the choice probability (CP) of the animal's stimulus selection. EV and CP increase with learning similar to the increase in the probability of the animal to make rewarded choices, causing all three variables to correlate (Figure 4E, F)."Figure 4 does not have E-F panels.

We corrected the sentence and added the correct Figure reference (which is revised Suppl Figure S5B).

4. Behavioral analysis: I could not find a formal definition of Choice Probability and Reward Prediction Error anywhere. I assume Equation 4 defines Choice Probability, while Rt-Vt defines RPE? I suggest making these definitions clear in the Methods, as well as the main text and the figure legend.

We added the definition in the revised text and methods and write e.g.:

“RPE is calculated as the difference of received outcomes R and expected value V (see Methods).” and

“positive reward prediction error (RPE, ‘R-V’, see Equation 2, below).” and

“Equation 4 defines the choice probability, or p(choice), that is used for the neuronal analysis of this manuscript (Sutton and Barto, 2018). P(choice) increases with trials since reversal (Supplementary Figure S5D), indicating a reduction in the uncertainty of the choice the more information is gathered about the value of the stimuli.”

Choice Probability is abbreviated in at least three different ways throughout the manuscript (e.g., p(choice), CP, CHP). Please be consistent.

Yes, pardon. We corrected this and now use “p(choice)” consistently throughout the text.

Note on terminology: Choice Probability commonly refers to the relationship between the activity of individual sensory neurons and the animal's behavioral choice (see Nienborg, Cohen and Cumming 2012 ARN). The duplicate terminology may be confusing for some readers. I suggest using a different term (e.g., Probability of Choice).

We understand the concern. By using the term p(choice) more directly in the text and defining more explicitly how p(choice) was calculated in Equation 4 we hope this reduces ambiguity with other ways of calculating or using it.

5. "We then quantified the log-likelihood of the independent test dataset given the training datasets optimal parameter values."Where is this result plotted? What is the model performance in predicting test dataset?

We validated the model that we used here in previous work on the same dataset and do not want to repeat this. We added more details about this in the revised methods section:

“The cross-validation results were compared across multiple models in a previous study (Oemisch et al., 2019). Here, we used the best-fitting model based on this prior work.”

And we write at an early place in the text:

“This color-based reversal learning is well accounted for by an attention augmented Rescorla Wagner reinforcement learning model (‘attention-augmented RL’) that we previously tested against multiple competing models (Balcarras et al., 2016; Hassani et al., 2017; Oemisch et al., 2019). Here, we use this model to estimate …”

6. Waveform analysis: It would help to add a diagram of T2P, T4R and HR in Figure 4.Relatedly, trough comes before the peak in extracellular spike waveforms (as apparent in Figure 4C) – T2P should be (t_peak_-t_trough_) in order to be a positive value?

We added the diagrams in a new Suppl. Figure S2A-B and refer to it in the revised results. We corrected the mentioning of peak to trough to the correct trough to peak (T2P).

7. "LV is a measure of regularity/burstiness of spike train and is proportional to the square of the difference divided by sum of two consecutive interspike intervals (Shinomoto et al., 2009)."This sentence should go in the main text. The reason being; the way LV is described in the main text makes it sound like LV and CV measure the same things: "regular or variable interspike intervals (local variability 'LV'), or more or less variable firing relative to their mean interspike interval (coefficient of variation 'CV')."

We adjusted the revised text and now write:

“LV and CV are moderately correlated (r=0.26, Suppl. Figure S2E), with LV indexing the local similarity of adjacent interspike intervals, while CV is more reflective of the global variance of higher and lower firing periods (Shinomoto et al., 2009).”

8. Given how central the clustering analysis in Figure 4A is to the rest of the paper, the exact parameters that went into this analysis (HR, T4R, LV, CV, FR) should be made clear in the main text.In addition, this clustering analysis is key to the reproducibility of e-types in other datasets. The authors have stated that "All data and code is available upon reasonable request." However, in my opinion, at least the code for the e-type clustering analysis should be made publicly available.

The code for clustering analysis is publicly available (and already used in other publications) and it is now cited explicitly in the paper along with a clarification of the parameters and reference to a new Suppl. Figure illustrating them in example sketches. We write in the revised results:

“Prior studies have distinguished different narrow spiking e-types using the cells’ spike train pattern and spike waveform duration (Ardid et al., 2015; Dasilva et al., 2019; Trainito et al., 2019; Banaie Boroujeni et al., 2020c). We followed this approach using a cluster analysis to distinguish e-types based on spike waveform duration parameters (inferred hyperpolarization rate and time to 25% repolarization, Suppl. Figure S2A,B), on whether their spike trains showed regular or variable interspike intervals (local variability ‘LV’, Suppl. Figure S2D), or more or less variable firing relative to their mean interspike interval (coefficient of variation ‘CV’, Suppl. Figure S2C).”

9. "Correlation of local variation with burst index"Burst index is defined here, but not plotted in any figures. I suggest adding a plot depicting the relationship between local variation and burst index would be informative.

We added this in a new Supplementary Figure S2E and F and show the correlation of LV and CV (r=0.27) and LV and burst index (0.443).

10. "First, we divided trials into two groups of high and low RPE and CHP values (trials were assigned based on their median value for each neuron)."I understood RPE and Choice Probability to be values unique to each trial, rather than to each neuron? If so, the median value should be specific to each behavior session, not to each neuron? Please clarify.

The reviewer is correct, and we corrected it and clarified the sentence. Median of the p(choice) and RPE is calculated over values of each session (on trials), not for neurons.

11. "We included only neurons with at least 50 spikes per time window."Does this sentence mean 50 spikes per time window per trial? For a 700ms time window, this would mean that the neuron would have to be firing at ~70Hz in order to be included in this analysis! If this sentence means 50 spike per time window across trials, please clarify. In this case, please also clarify the range of trial number that went into this analysis.

It is ≥50 spikes across trials. We adjusted the text and also added the number of available trials. We write in the methods:

“We included only neurons with at least 50 spikes across trials, using on average 44 (SE= 2) trials per condition.”

Reviewer #3:

In this work, Boroujeni et al. investigated the role of different cellular subtypes in the lateral prefrontal cortex (LFPC) and anterior cingulate cortex (ACC) of the rhesus macaque as the animals performed an attention demanding reversal-learning task. The authors use an attention-augmented reinforcement learning model to track the trial-by-trial values of key decision-making variables which were then correlated against the neural activity. The cellular population was separated into broad and narrow spiking neurons using features computed from the extracellularly recorded waveforms. The authors find that the activity of the narrow spiking cells in the LFPC is correlated with the choice probability, whereas the activity of narrow spiking cells in the ACC is correlated with reward prediction errors. Interestingly, the authors find that further splitting the population of broad and narrow spiking cells into subtypes revealed that both the choice probability in LPFC and the reward prediction error in the ACC were encoded by a specific subtype of putative interneuron. The authors show that the spike-field phase synchronization of this putative interneuron subgroup is also modulated by choice probability in the LFPC and reward prediction error in the ACC, mirroring the result from their single-unit correlation analysis. The authors use these results to propose a biologically plausible circuit model of how learning in such a task might be implemented through interneuron specific synchronization.While many of the results in the paper seem robust, some of the conclusions drawn by the authors rest on analyses and methods that require further validation and controls.1. The clustering of the cell population into 5 broad-spiking and 3 narrow-spiking subtypes is perhaps one of the most critical results that requires further validation since a lot the conclusions in the paper rely on the outcome of this analysis. The validation that the authors include in the paper (Figure S5C, S5D) address concerns regarding the clustering quality, but it's still unclear how meaningful this separation into these 8 clusters actually is. The clustering is also performed on the pooled data across both animals, but the authors should have also shown what the clustering looks like when performed independently on the population from each animal, and if there is a meaningful correspondence between the sets of clusters recovered in the two populations.

We address the specific suggestions for further validation/recovering reliability and monkey separation of the clustering with added analysis and result figures. The additional validation of the cluster number and the cluster boundaries showed they are highly reliable also in each of the monkeys. We added new Suppl. Figure S7 and new Suppl. Figure S8 and adjusted the results text by writing:

“Cluster boundaries were highly reliable (Suppl. Figure S7). Moreover, assignment of a cell to its class was statistically consistent and also reliably evident for cells from each monkey independently (Suppl. Figure S8).”

We describe the methods in a new section “Determining cluster numbers” and write:

“We used a set of statistical indices to determine a range of number of clusters that best explains our data. […] We validated finally determined number of clusters using Akaike’s and Bayesian criteria which showed the smallest value for k=8 (AIC: [-17712, -17735, -18476, -11114] and BIC: [-1.7437, -1.7368, -1.8109, -1.0747], for k= [6,7, 8,9]).”

For the monkey specific analysis, we added to the revised methods:

“We further validated the meta-clustering results for each monkey separately. We validated the results, analogous to what is describe above. […] Second, validation according to the percent number of cells matches for each monkey (Suppl. Figure S8F).”

Overall, the clustering analysis results in our manuscript are highly consistent with a similar analysis performed in (at least) two different prior datasets in Ardid et al., 2015 and in DaSilva et al. 2019. We are currently working on a review paper that summarizes and compares the clustering based on the spike train parameters we use. Our reply to comment 7 of reviewer 2 also describes how the clustered e-types can be linked to e-types reported in recent in-vitro clustering results from mice visual cortex (Gouwens et al., 2019).

2. Most of the follow-up analysis focuses on the comparison between one specific interneuron subtype (N3) and all broad -spiking cells. I imagine that the reason for this is two-fold: (1) the N3 subtype is the only one that showed a significant modulatory effect on the multi-unit activity (Figure 4D), and (2) it seems to be special in the sense that the activity of the N3 cells is significantly correlated with choice probability in LPFC in addition to reward prediction error in ACC. While the reasons for showing key results only for the N3-type can be appreciated, the authors should have included additional control analysis to demonstrate that their results are indeed specific to the N3 subtype. For example, in Figure 7 and 8, the authors show a comparison of the spike-LFP phase synchronization between N3 and broad spiking cells, but no further characterization of subtypes within the broad spike cells or the other narrow spiking types (i.e. N1, N2).

We agree. That was an oversight on our part. We added multiple additional control analyses and results for other cell classes too. These added analyses support the special role of N3 cells in LPFC and in ACC. Perhaps most relevant are the spike-LFP results in the newly added Supplementary Figure’s S13, S14 and S15.

Suppl. Figure S15A shows the time frequency results for each cell class in LPFC around cue onset in the low and high p(choice) trials and reveals that the gamma increase to the feature cue onset is specific to the N3 class. Suppl. Figure S15B shows the time frequency results for each cell class in ACC around reward onset in the low and high RPE trials and reveals that the gamma increase to the reward onset is specific to the N3 class in the high RPE trials. These time frequency plots are noisy because many neurons fire only few spikes in the time windows of interest and there are only few cells in each class.

To more clearly illustrate the effect on gamma band spike-LFP synchronization and simplify the statistical analysis we extracted the average gamma band effect for all cell classes and sub-conditions, and summarize them directly in Figure ’s S13 and S14:

The specific gamma effect for cell classes in LPFC are shown in Figure S13: It shows that for low and high p(Choice) and low and high RPE, the N3 class sticks out by showing selectively higher gamma synchrony in the low p(Choice) condition (Suppl. Figure S13E).

In ACC it is similar. Figure S14 shows higher gamma for high RPE but not for low RPE or high or low p(choice) particularly for the N3 class (Suppl. Figure S14H).

We adjusted the revised methods sections to fairly describe these results

For LPFC:

“… N3 e-type neurons synchronizing significantly stronger to beta than broad spiking neuron types (Figure 7F) (p<0.05 randomization test, multiple comparison corrected). […] There was no difference in gamma synchrony of other cell classes in LPFC in the 0-0.7 s after reward onset in the high or low RPE trials, or around the (0.7 s) color onset in the high p(choice) trials (Suppl. Figure S13E-H, see Suppl. Figure S15A for time-frequency maps for all cell classes around cue onset).”

For ACC:

“… the N3 e-type neurons synchronized in a 35-42 Hz gamma band following the reward onset when RPE’s were high (i.e. when outcomes were unexpected), which was weaker and emerged later when RPEs were low, and which was absent in broad spiking neurons (Figure 8). […] E-type classes did not differ in their spike-LFP gamma in low RPE trials, or during the color cue period in high or low p(choice) trials (Suppl. Figure S14E-H, see Suppl. Figure S15B for time-frequency maps for all cell classes around reward onset).”

3. The authors show that the spike-field phase synchronization of the N3 subgroup is also modulated by choice probability in the LFPC (Figure 7) and reward prediction error in the ACC (Figure 8), mirroring the result from their single-unit correlation analysis (Figures 2 and 3). Unlike their firing rate analysis however, they do not show anatomical specialization in these analyses, even though the model they propose in Figure 9 clearly shows that they hypothesize this to be the case. It would be very interesting to show the analysis performed in Figure 7 for the ACC N3 population, and likewise, the analysis performed in Figure 8 for the LPFC N3 population.

We added the suggested analysis in new Figures which support the specificity of the observed effects:

Supplementary Figure S13A-E shows the reward onset aligned analysis for the LPFC broad spiking neurons and the N3 e-type in analogy to Figure 7 in the main text. There is no gamma spike-LFP synchrony for these neurons in the reward period at low or high RPE trials.

Supplementary Figure S14A-E shows the color onset aligned analysis for the ACC broad spiking neurons and the N3 e-type neurons in analogy to Figure 8 in the main text. There is no gamma spike-LFP synchrony for these neurons in the color cue period for low or high p(choice).

4. Behavior

a. In Figure 1C, I imagine that the proportion of rewarded choices at reversal (t=0, not shown) is equal to one minus the asymptotic performance? So around 0.1?

Thank you for catching this. The original plot was erroneously showing not proportion correct (but a probability estimate of correct choices). We corrected the panel to show “proportion correct choices” and also now show the last trials prior to the reversal.

b. If the stimulus-reward pairings are fully deterministic, why does the monkey require so many trials (on average 7 I believe it was) to reach asymptotic performance again?

We believe the animals learn sub-optimally slow in this task, because it is a difficult, attention demanding task. Their choice is made according to the motion direction of the stimulus with the rewarded color. When they do not get reward with their choice then this can be not only because of the wrong color of the stimulus, but also because of the wrong motion direction or even the wrong location they attended. This makes the task highly demanding and dependent on a good representation of the currently rewarded color. This task is very different to spatial reversal tasks or simple object-based reversal tasks where animals are allowed to look at the specific stimuli (In our task, animals do not make a saccade to the peripheral stimuli but to up- and down- ward presented response targets).

Please note that we tried fitting the behavior with optimal Bayesian models in previous work and confirmed that they failed to account for behavior, because they would learn too fast (Oemisch et al., 2019). Rather, we found that a reinforcement learning model with a selective decay (forgetting) of nonchosen (or: non-attended) stimulus values was outperforming other models consistently. We use this so-called attention-augmented reinforcement learning model to account.

c. Related to the previous question, is there any change in this acquisition time over the course of a session (as they experience more and more reversals)?

There is no systematic change in reversal acquisition time over the course of the session.

d. Can you show some example fits of the reinforcement learning model? For example, the choice probability and expected value as a function of the trial number around a reversal.

Thank you for this suggestion. We added some example blocks that show how the estimated RPE, the choice probability, and the value of the chosen stimulus varies trial-by-trial along with the correct/error outcome the monkey experiences. This is added in a revised Suppl. Figure S5. More extensive analysis of the model, including a comparison to alternative models that fit (and predict) the data worse, have been previously published in Oemisch et al., 2019.

We refer to the new figure panel by writing in the Results section:

“Choice probabilities (p(choice)) increase during reversal learning when reward prediction errors (RPEs) of outcomes decrease, evident in an anticorrelation of (p(choice)) and RPE of r=-0.928 in our task (Suppl. Figure S5A,B) with lower p(choice) (near ~0.5) and high RPE over multiple trials early in the reversal learning blocks when the animals adjusted to the newly reward color (Suppl. Figure S5E,F).”

5. Single Unitsa. The authors correlate the neural activity with model-derived variables, like the probability of choice, and prediction error. The distributions of these variables, however (as indicated in Figure S4b, and S4C) are very skewed, and it seems like most of the variability comes from the few trials (around 10) that it takes to reach asymptotic performance after a reversal. It would be interesting to know what this correlation represents. Are the cells truly tracking small changes in the P(choice) and PE or does this reflect more of a discrete switch? Maybe the authors could show some scatters, firing rate vs. P(choice), of some example cells. How well can p(choice) and PE be decoded from the neural population?

We cannot conclusively answer how gradual or discrete the correlation occurs in the current dataset given the few and low firing neurons and the rather few learning blocks during which they were recorded. The average correlation of the N3 cell classes firing rate with p(choice) were significant but weak: r=0.08 in LPFC and with RPE in ACC it was r=0.09.

To address the reviewer, we now visualize in a new Suppl Figure 10 two example cells of the N3 – e-type, showing the positive correlation of rate and RPE in ACC, and the positive correlation of rate and p(choice) in LPFC. We pooled all trials and sorted them based on RPE/P(choice) Value. We then plot the raster of trials sorted according to the RPE and p(choice) and the pre-onset normalized heatmap. A decoding analysis of rate and synchrony is not part (and beyond the scope) of this manuscript and will be comprehensively performed in a separate project.

6. Electrophysiology/ClusteringIt seems that a lot of the results in the paper rely on clustering analysis. The authors have been cautious in their approach (i.e., validating the results), but given that a lot depends on the reliability of these results, I think it would be wise to add a few more control analyses. I am not sure how feasible these are, but worth considering nonetheless:a. Another way of validating the clustering is to do it across animals. From what I understood, the clustering (for e-type) is done using data from both animals. How well would a clustering model fit within animals, predict the clustering across animals?

In the revised manuscript we provide additional validation analyses and separately validate the clustering for the individual monkeys.

Please see our reply to comment-1 of the reviewer for details and how we adjusted the text. For the assessment of the cluster quality for each monkey separately, please see newly added Suppl Figure S8E,F.

7. Spike field coherencea. Can the authors comment on the effect of ERPs?

We assume the reviewer suggests that the onset of the reward (or color cue) triggers a low frequency evoked response that biases how neurons phase synchronize to the LFP also at other frequencies.

To address this concern, we repeated the spike-LFP analysis and the statistical evaluation after removing the average cue- or reward onset- evoked LFP. The main p(Choice) and RPE effects in LPFC and ACC were unchanged.

We added this new result as a new Suppl. Figure 16, describe the methods, and summarize the result it in the revised main text:

“The spike-LFP synchronization results were unchanged when the average reward onset aligned LFP, or the color-cue aligned LFP was subtracted prior to the analysis, which controls for a possible influence of lower frequency evoked potentials (Suppl. Figure S16).”

b. Simply controlling for the number of spikes between conditions is not necessarily sufficient. If you have a cell that responds to one condition but does not respond to another condition, the spikes for condition 1 are going to be much more clustered in time than for condition 2. Therefore the underlying LFP is not sampled in the same way between the two conditions.

We have only few number of spikes for many cells and cell classes, so that controlling for the number of spikes seems to be a good control when quantifying and comparing spike-LFP phase consistency.

c. Is it possible to show that the spike-field coherence results are also anatomically specific? Does the synchrony of cells in the ACC and LPFC mirror the single-unit results, i.e. reward prediction error in ACC but not LPFC and choice probability in LPFC but not ACC?

We agree that this is an important point that we missed in the first submission – thank you for pointing us to this. We performed the analyses and added new supplementary figures (Suppl. Figures S13 and S14) and found that the effects are anatomically specific.

We kindly refer to the reply to comment 2 of the reviewer for details and how we adjusted the text.

[Editors’ note: what follows is the authors’ response to the second round of review.]

Reviewer #1:This paper has made significant improvements from the previous version. Most importantly, the implementation details of the circuit simulation are clarified. The vast majority of my prior concerns have been addressed. I have only a few suggestions remaining.1. Given that reward prediction error analysis is critical to the thesis of this paper, I am still of the opinion that it would be important to include the PSTH aligned to the reward, for narrow spiking and broad spiking neurons (as in Figure 2) as well as for important e-types (as in Figure S3).

To address the request we added to the revised manuscript information about the firing of the cell types during the reward period (new figure panels in Figure 3 and Figure5-supplement 1 (formerly Figure S9)).

First, we added the spike densities during the reward period in a revised Figure 3. And we revised the text to provide directly the reward related information by writing:

"First, we analyzed N- and B-type cell responses to the reward. In both LPFC and ACC areas, N- and B-type cells on average showed activation to the reward onset (p<0.05, randomization test, n=26 of 54 and 18 of 188 B- type cells with increases, respectively, and n=14 of 54 N- type and 5 of 188 B-type cells with decreased firing in LPFC, and n=30 of 50 N-type and 13 of 216 B- type cells with increases, respectively, and n=19 of 50 and 8 of 216 B-type cells with decreased firing in ACC). However the N- and B-type responses to the reward were not significantly different in ACC or LPFC (ns., randomization test, Figure 3A,B)."

We also added two figure panels to Figure5-supplement 1 (formerly Suppl. Figure S9) to show reward onset related activity and write in the revised figure legend:

"(G, H) Reward activated neurons of different e-types in LPFC and ACC. For LPFC the average normalized firing of each e-type to the reward onset (G) show moderately increased firing rate in most e-types. […] In ACC (H) the N2 e-type neurons showed stronger activation to the reward onset compared with other e-types (p<0.05, effect size values for B1, N2 are -0.311, -0.367 in LPFC and ACC respectively)."

We also added the effect sizes of these results to the Cohen's d effect size table which is now included in “Supplementary File 1”.

These results do not change any of the prior conclusions.

2. The added classifier analysis or predicting cell classes from their correlations with learning variables is very interesting. However, I am not clear on exactly what was used to train the SVM. The way I currently understand this analysis is that in LPFC, correlation between firing rate and p(choice) was calculated for each neuron – and this one-dimensional vector, the size of which is (Number of neurons)X1, was used to train the SVM. Is this the case? Please clarify.

The reviewer is correct. A vector of correlation values was used along with a vector of cluster label (from our clustering results) to train the SVM. So the input to the SVM was a 1D vector of correlation values with a length of Number of the neurons. We clarified it in the main text.

3. Figure S5 E and F: it is hard to see a trend in these plots. I suggest either making the dots transparent; or plotting the data as a 2D-histogram. This way it would be possible to discern where the data is the densest.

Thank you. We added two 2D histograms for RPE and P(choice) to Figure S5 and now say in the legend:

"(G, H) 2D histogram corresponding to E and F, respectively, showing the distribution of trials P(choice) (G) and RPE (H) and trial since reversal."

4. In Methods, the numbering in the equations are not unique (there's two Equation 2 and two Equation 3). Please correct.

Thank you for finding this issue. We corrected it.

5. The following sentences in Supplementary Online Information needs to be corrected as indicated:"These circuit motifs are provided to provide a proof-of-concept that the observations can follows from biologically plausible motifs. These circuits motifs also provide predictions which can be tested in future studies."

We corrected the sentence. Thank you for pointing us to this issue.

Reviewer #2:In this work, Boroujeni et al. investigated the role of different cellular subtypes in the lateral prefrontal cortex (LFPC) and anterior cingulate cortex (ACC) of the rhesus macaque as the animals performed an attention-demanding reversal-learning task. The authors use an attention-augmented reinforcement learning model to track the trial-by-trial values of key decision-making variables which were then correlated against the neural activity. The cellular population was separated into broad and narrow spiking neurons using features computed from the extracellularly recorded waveforms. The authors find that the activity of the narrow spiking cells in the LFPC is correlated with the choice probability, whereas the activity of narrow spiking cells in the ACC is correlated with reward prediction errors. Interestingly, the authors find that further splitting the population of broad and narrow spiking cells into subtypes revealed that both the choice probability in LPFC and the reward prediction error in the ACC were encoded by a specific subtype of putative interneuron. The authors show that the spike-field phase synchronization of this putative interneuron subgroup is also modulated by choice probability in the LFPC and reward prediction error in the ACC, mirroring the result from their single-unit correlation analysis. The authors use these results to propose a biologically plausible circuit model of how learning in such a task might be implemented through interneuron-specific synchronization.The analysis is thorough and the authors present a nice narrative of the results, even though in some cases my interpretation of the data is a little more mixed than what is written in the paper. For example, the authors are eager to point out that their results are "interneuron specific" and yet the data that they show suggests otherwise. Take the spike-LFP synchronization results shown in Figure S15, where it seems that the modulation of pairwise phase consistency with p(choice) could also be present for the B1 cluster of cells in addition to the N3 group (no stats shown). The same could be true for the B2 type in the ACC, which seems to show differential effects for high and low RPE.Are these real effects or are these anomalies that are biased by a few outliers? In either case, please clarify.

In response to the comment, we calculated (and added) the statistics for the time frequency synchronization results for all the cell classes. Statistically significant synchronization is shown as a black contour in the revised Figure7-supplement 2 (formerly figure S15).

The added statistics shows the synchronization effect in the 35-45 Hz gamma band was significant only for the N3 class. The B1 class in PFC did not show significant synchrony effects and thus show noisy outlier driven synchrony results. The B2 class in ACC shows a significant synchronization effect at a higher gamma band. We added this information explicitly in the main results text and write:

“Other e-type classes did not differ in their spike-LFP synchronization in this 35-45 Hz gamma band in low or high RPE trials with the exception of the B1 class in ACC that synchronized in high RPE trials at a higher >50Hz gamma band (Figure 7-supplement 3E-H, see Figure 7-supplement 2B for time-frequency maps for all cell classes around reward onset)”.

Thank you for including the new supplementary figures; I can really appreciate the additional amount of work that must have gone into preparing the new controls for the second submission of the paper. The addition of the example model fittings (Figure S5) and the correlation of the firing rate from the two example cells with the RPE and p(choice) (Figure S10) are very nice. I would recommend to the authors to move the two examples in Figure S10 to one of the main figures. In the first submission, the focus of the paper was predominantly on the N3 subtype and its specialized functional properties in ACC and LPFC. The new figures however (specifically Figure S15) show that the story is a little more mixed than originally presented. B1 for example in LPFC shows differential effects for high and low P(choice) and B2 in ACC shows differential effects for high and low RPE. In any case, the new figures provide a much more complete story and I feel made the paper stronger.

We very much appreciate the positive reception of the revision. We added to the main text that statistical testing (that we added about Figure7-supplement 2, formerly Figure S15) suggests the B2 class in ACC shows also a gamma effect at high RPE. The B1 class in PFC showed no significance. We opted to leave the example raster in the supplementary figure because it is difficult to discern a correlation finding in example a raster plot and the average results appear more convincing.